# BREGMAN GEOMETRY FOR STOCHASTIC ONLINE BILEVEL OPTIMIZATION

## ABSTRACT

We study *online bilevel optimization (OBO)* in the *stochastic* setting and ask whether geometry can eliminate the severe dependence on the condition number of the inner problem, $\kappa_g = \ell_{g,1}/\mu_g$. We introduce a family of *Bregman-based algorithms* and analyze both oracle and practical regimes. In the oracle setting, where exact hypergradients are available, generalized Bregman steps achieve sublinear bilevel local regret (i.e., $o(T)$) while *removing the cubic dependence on $\kappa_g$* incurred by Euclidean updates. In the practical stochastic setting, where hypergradients must be estimated, we design single-loop, sample-efficient algorithms that combine Bregman steps with time-smoothed hypergradient estimates. Our analysis shows that Bregman geometry again eliminates the $\kappa_g$-dependence and yields guarantees of sublinear bilevel local regret in this setting. It further reveals a broader insight: time smoothing, previously treated as a heuristic in deterministic OBO, naturally functions as a *variance-reduction mechanism* while keeping bias controlled, clarifying its role across both regimes. Finally, experiments on preconditioner learning and reinforcement learning support our theoretical findings across a variety of nonstationary loss sequences and large-scale, ill-conditioned datasets.

## 1 INTRODUCTION

Bilevel optimization in machine learning is widespread, with applications in *hyperparameter optimization* Pedregosa (2016), *learned optimizer training* Andrychowicz et al. (2016), and *reinforcement learning* Chakraborty et al. (2023). It addresses problems with a nested structure: the *outer* variable $\boldsymbol{\lambda} \in \mathcal{X} \subseteq \mathbb{R}^{d_1}$ is chosen by minimizing a composite outer objective $F = f + h$, while the *inner* variable $\boldsymbol{\beta} \in \mathbb{R}^{d_2}$ comes from minimizing an inner objective:

$$\boldsymbol{\lambda}^* \in \arg\min_{\boldsymbol{\lambda} \in \mathcal{X}} F(\boldsymbol{\lambda}), \qquad F(\boldsymbol{\lambda}) \triangleq f(\boldsymbol{\lambda}, \boldsymbol{\beta}^*(\boldsymbol{\lambda})) + h(\boldsymbol{\lambda}), \qquad \boldsymbol{\beta}^*(\boldsymbol{\lambda}) \in \arg\min_{\boldsymbol{\beta} \in \mathbb{R}^{d_2}} g(\boldsymbol{\lambda}, \boldsymbol{\beta}). \quad (1)$$

where $F$ is nonconvex and smooth, $h$ is convex and potentially non-smooth, and $g$ is smooth and $\mu_g$-strongly convex in $\boldsymbol{\beta}$. In many machine learning settings, $f$ and $g$ are expectations over data or environment randomness and are not available in closed form. With samples $(\xi, \zeta)$, the stochastic bilevel problem is

$$\boldsymbol{\lambda}^* \in \arg\min_{\boldsymbol{\lambda} \in \mathcal{X}} F(\boldsymbol{\lambda}) \triangleq \mathbb{E}_{\xi}\big[f(\boldsymbol{\lambda}, \boldsymbol{\beta}^*(\boldsymbol{\lambda}); \xi)\big] + h(\boldsymbol{\lambda}), \quad \boldsymbol{\beta}^*(\boldsymbol{\lambda}) \in \arg\min_{\boldsymbol{\beta} \in \mathbb{R}^{d_2}} \mathbb{E}_{\zeta}\big[g(\boldsymbol{\lambda}, \boldsymbol{\beta}; \zeta)\big]. \quad (2)$$

Gradient-based methods update $\boldsymbol{\lambda}$ using hypergradients of the composite outer objective $F(\boldsymbol{\lambda})$, computed either via implicit differentiation of the inner optimality condition Pedregosa (2016); Lorraine et al. (2020) or via iterative/truncated differentiation through an inner solver Maclaurin et al. (2015); Franceschi et al. (2017). Prior stochastic bilevel algorithms in the offline regime provide convergence and sample-complexity guarantees under noisy gradients Ghadimi & Wang (2018); Ji et al. (2021). Recent works study more-efficient stochastic hypergradient estimators, via momentum or variance reduction Khanduri et al. (2021); Chen et al. (2021); Hong et al. (2023).

Despite progress in the *offline* setting, the online bilevel optimization, e.g., $\forall t = 1, \ldots, T$ :

$$\boldsymbol{\lambda}_t^* \in \arg\min_{\boldsymbol{\lambda} \in \mathcal{X}} F_t(\boldsymbol{\lambda}) \triangleq \mathbb{E}_{\xi_t}\big[f_t(\boldsymbol{\lambda}, \boldsymbol{\beta}_t^*(\boldsymbol{\lambda}); \xi_t)\big] + h_t(\boldsymbol{\lambda}), \quad \boldsymbol{\beta}_t^*(\boldsymbol{\lambda}) \in \arg\min_{\boldsymbol{\beta} \in \mathbb{R}^{d_2}} \mathbb{E}_{\zeta_t}\big[g_t(\boldsymbol{\lambda}, \boldsymbol{\beta}; \zeta_t)\big] \quad (3)$$

remains underdeveloped. Existing methods are limited to *deterministic and Euclidean* Tarzanagh et al. (2024); Lin et al. (2024): they assume noiseless gradients and use Euclidean outer updates.

| Alg. | Bregman | Loop | Stoch. | Samples | $\kappa_g$ dep. |
|---|---|---|---|---|---|
| OAGD Tarzanagh et al. (2024) | ✗ | Double | ✗ | N/A | $p(\kappa_g)$ |
| SOBOW Lin et al. (2024) | ✗ | **Single** | ✗ | N/A | $p(\kappa_g)$ |
| **SOBO (ours)** | ✗ | **Single** | ✓ | **O(1)** | $\boldsymbol{\kappa_g^5}$ |
| **SOBBO (ours)** | ✓ | **Single** | ✓ | **O(1)** | **O(1)** |

Table 1: **Current Online Bilevel Optimizers.** *Bregman*: supports Bregman geometry. *Loop*: single vs. double loop. *Stoch.*: supports stochastic gradients. *Samples*: per-round gradients for stochastic algorithms. $\boldsymbol{\kappa_g}$ *dep.*: leading dependence on the inner condition number in bounds.

These methods achieve sublinear *bilevel local regret (BLR)*—a stationarity-based regret on the outer objective—but do not address two key challenges: (i) incorporating geometry-aware (Bregman) outer updates for ill-conditioned problems, and (ii) handling stochastic, nonstationary data. We next highlight how these gaps are significant in online bilevel optimization problems.

**Gap 1: Geometry.** Bregman geometry underlies many advances in online optimization, unifying adaptivity and proximal regularization while improving robustness to ill-conditioning (e.g., Adagrad Duchi et al. (2011), implicit online learning Kulis & Bartlett (2010)). In *online nonconvex single-level* settings, geometry-aware proximal (Bregman) updates are standard, and analyses based on stationarity-type criteria (gradient mapping/local regret) yield sublinear guarantees under reasonable smoothness and variation assumptions Hazan et al. (2017); Aydore et al. (2019); Hallak et al. (2021). In *online bilevel* problems, the challenge is related but distinct: ill-conditioning originates in the *inner* problem, where the condition number $\kappa_g$ controls the sensitivity of $\beta^*(\lambda)$ and, via the hypergradient, the stability of outer updates. When $\kappa_g$ is large, outer steps become fragile to noise and drift. Yet existing online bilevel formulations Tarzanagh et al. (2024); Lin et al. (2024) rely on Euclidean outer updates; to our knowledge, there are no geometry-aware outer steps even in deterministic settings. As a result, current methods exhibit strong dependence on $\kappa_g$—a critical liability in nonstationary regimes where ill-conditioning is ubiquitous.

**Gap 2: Stochasticity.** Large-scale learning is inherently stochastic: both inner and outer gradients are noisy, and hypergradients must be estimated. Deterministic formulations from Tarzanagh et al. (2024); Lin et al. (2024) cannot capture these dynamics because they require full-batch gradients. A stochastic formulation aligns the theory with practice, where mini-batch sampling is essential for scalability, computational efficiency, and robustness to noise. Moreover, a stochastic analysis provides additional theoretical insight: as we later show in Corollary 6.2, time-smoothing—previously used heuristically in deterministic OBO—emerges naturally as a *variance-reduction mechanism*, thereby clarifying its role across both deterministic and stochastic regimes.

**Our contributions.** We develop a unified framework for *online stochastic bilevel optimization (OSBO)* with guarantees in both oracle and stochastic settings:

1. **Bregman geometry improves $\kappa_g$ dependence.** *Bregman* steps achieve sublinear BLR ($o(T)$) and strictly better $\kappa_g$-dependence than Euclidean updates: in the oracle case they *remove the $\kappa_g^3$ dependence*; in the stochastic case they *eliminate the $\kappa_g^5$ dependence* (Table 1).

2. **Single-loop, sample-efficient stochastic algorithms.**:
   - **SOBO (Euclidean):** sublinear BLR but $\kappa_g^5$ dependence (Table 2).
   - **SOBBO (Bregman):** time-smoothed hypergradients + Bregman steps, *removing $\kappa_g^5$* while preserving sublinear BLR (Table 1).

3. **Time smoothing as variance reduction.** Stochastic analysis shows time smoothing is a *variance-reduction mechanism* for noisy hypergradients with controlled bias, see Table 2.

4. **Empirical validation.** On preconditioner learning and RL with nonstationary, ill-conditioned data, our stochastic methods outperform current online bilevel baselines.

The paper is structured as follows. Section 2 introduces notation, Bregman-based gradient steps, and bilevel local regret. Section 3 analyzes Bregman-based optimizers in an oracle setting. Section 4 addresses hypergradient estimation and presents the proposed algorithms. Section 5 provides regret analysis. Section 6 includes experimental results on preconditioner learning and reinforcement learning. Proofs and extensions to the deterministic setting are in the Appendix.

| Alg. | Upper bound on BLR |
|------|---------------------|
| OAGD Tarzanagh et al. (2024) | $O\!\left(\frac{T}{w} + H_{1,T} + H_{2,T}\right)$ |
| SOBOW Lin et al. (2024) | $O\!\left(\frac{T}{w} + V_{1,T} + H_{2,T}\right)$ |
| **SOBO (ours)** | $O\!\left(\frac{T\kappa_g^3}{w}\left(\kappa_g^2 + \sigma_f^2 + \kappa_g^2\sigma_g^2\right) + V_{1,T} + \kappa_g^2 H_{2,T}\right)$ |
| **SOBBO (ours)** | $O\!\left(\frac{T}{w}\left(\sigma_f^2 + \sigma_g^2\right) + V_{1,T} + H_{2,T}\right)$ |

**Table 2: Sublinear bilevel local regret.** Bounds are given for comparator sequences $V_{1,T}, H_{1,T}, H_{2,T}$. SOBBO eliminates the $\kappa_g^5$ dependence of SOBO and removes $\kappa_g$ from leading variance terms. Increasing the smoothing window $w$ reduces variance via the $\frac{T}{w}$ factor.

## 2 PRELIMINARIES

### 2.1 NOTATION AND ASSUMPTIONS

Let $\|\cdot\|$ denote the $\ell_2$ norm for vectors and the spectral norm for matrices, with $\langle \boldsymbol{\beta}_1, \boldsymbol{\beta}_2 \rangle$ denoting the inner product between $\boldsymbol{\beta}_1$ and $\boldsymbol{\beta}_2$. For a function $g_t(\boldsymbol{\lambda}, \boldsymbol{\beta}, \zeta)$, we denote the gradient as $\nabla g_t(\boldsymbol{\lambda}, \boldsymbol{\beta}, \zeta)$. Partial derivatives are denoted, for example with respect to $\boldsymbol{\lambda}$, as $\nabla_{\boldsymbol{\lambda}} g_t(\boldsymbol{\lambda}, \boldsymbol{\beta}, \zeta)$. We make the following assumptions that are standard in online Tarzanagh et al. (2024); Lin et al. (2024) and stochastic Ghadimi & Wang (2018); Ji et al. (2021); Huang et al. (2022b) bilevel optimization.

**Assumption A** (Smoothness of Objective Functions). *For each $t \in \{1, \ldots, T\}$, $\boldsymbol{\lambda} \in \mathcal{X}$, and $\boldsymbol{\beta} \in \mathbb{R}^{d_2}$, there exist $\ell_{f,0}, \ell_{f,1}, \ell_{g,1}, \ell_{g,2} > 0$ such that:*

  1. *$f_t(\boldsymbol{\lambda}, \boldsymbol{\beta}, \epsilon)$ is $\ell_{f,0}$-Lipschitz and $\nabla f_t$ is $\ell_{f,1}$-Lipschitz.*

  2. *$\nabla g_t(\boldsymbol{\lambda}, \boldsymbol{\beta}, \zeta)$ is $\ell_{g,1}$-Lipschitz, $\nabla^2_{\lambda\beta} g_t$ and $\nabla^2_{\beta\beta} g_t$ are $\ell_{g,2}$-Lipschitz.*

**Assumption B** (Strong Convexity of Lower-Level Objective). *For all $t$, $g_t(\boldsymbol{\lambda}, \cdot, \zeta)$ is $\mu_g$-strongly convex in $\boldsymbol{\beta}$ for every $\boldsymbol{\lambda} \in \mathcal{X}$.*

Assumptions A–B imply the inner condition number $\kappa_g := \ell_{g,1}/\mu_g \geq 1$.

**Assumption C** (Stochastic Gradients). *For all $t$ and $(\boldsymbol{\lambda}, \boldsymbol{\beta})$, unbiased stochastic estimators exist for the required first/second-order quantities (e.g. $\mathbb{E}[\widehat{\nabla}_\beta g_t] = \nabla_\beta g_t$), and analogously for $\widehat{\nabla} f_t$, $\widehat{\nabla}^2_{\beta\beta} g_t$, $\widehat{\nabla}^2_{\beta\lambda} g_t$. Their variances are bounded: $\mathbb{E}\|\widehat{\nabla}_\beta g_t - \nabla_\beta g_t\|^2 \leq \sigma_g^2$ and $\mathbb{E}\|\widehat{\nabla} f_t - \nabla f_t - B(\lambda, \beta)\|^2 \leq \sigma_f^2$.*

**Assumption D** (Bounded Decision Space). *$\mathcal{X} \subseteq \mathbb{R}^{d_1}$ is closed, convex, and bounded with diameter at most $S$, i.e., $\|\lambda_1 - \lambda_2\| \leq S$ for all $\lambda_1, \lambda_2 \in \mathcal{X}$.*

**Assumption E** (Bounded Objective). *For all $t$, $\sup_{\boldsymbol{\lambda} \in \mathcal{X}} |F_t(\lambda)| \leq Q$.*

**Assumption F** (Distance Generating Function). *For all $t$, $\phi_t : \mathcal{X} \to \mathbb{R}$ is continuously differentiable and $\rho$-strongly convex, so that $D_{\phi_t}$ is well-defined.*

Smoothness and strong convexity ensure the inner problem is well-conditioned; the stochastic oracle assumptions allow unbiased, bounded-variance access to gradients Khanduri et al. (2021); boundedness of $\mathcal{X}$ and $F_t$ prevent divergence Hazan et al. (2017); and the distance generating function specifies the geometry in which our regret is measured Huang et al. (2022a;b).

### 2.2 BREGMAN PROXIMAL GRADIENT

Introduced in Bregman (1967), Bregman divergences generalize the squared Euclidean distance. Given a continuously differentiable and $\rho$-strongly convex function $\phi(\boldsymbol{\lambda})$, the Bregman divergence is defined as $\mathcal{D}_\phi(\boldsymbol{\lambda}_2, \boldsymbol{\lambda}_1) := \phi(\boldsymbol{\lambda}_2) - \phi(\boldsymbol{\lambda}_1) - \langle \nabla\phi(\boldsymbol{\lambda}_1), \boldsymbol{\lambda}_2 - \boldsymbol{\lambda}_1 \rangle$. Given a Bregman divergence $\mathcal{D}_\phi(\cdot, \cdot)$, our proximal gradient step is

$$\boldsymbol{\lambda}^+ = \arg\min_{\boldsymbol{\lambda} \in \mathcal{X}} \left\{ \langle \boldsymbol{q}, \boldsymbol{\lambda} \rangle + h(\boldsymbol{\lambda}) + \frac{1}{\alpha} \mathcal{D}_\phi(\boldsymbol{\lambda}, \boldsymbol{u}) \right\}, \tag{4}$$

where $\phi(\boldsymbol{\lambda})$ is a continuously differentiable and $\rho$-strongly convex function, $h(\boldsymbol{\lambda})$ is a convex and potentially nonsmooth regularization term, $\alpha > 0$ is a step size, and $\boldsymbol{q}, \boldsymbol{u} \in \mathbb{R}^{d_1}$ are the estimate

of the gradient, and current reference point, respectively. Proximal gradient methods in offline bilevel optimization have been shown to improve convergence rates in the deterministic setting (e.g., Bio-BreD algorithm of Huang et al. (2022b)) and stochastic setting (e.g., SBio-BreD algorithm of Huang et al. (2022b)). Special cases of the gradient update in equation 4 include projected (stochastic) gradient descent ($\phi(\boldsymbol{\lambda}) = \frac{1}{2} \|\boldsymbol{\lambda}\|^2$, $\mathcal{X} \subseteq \mathbb{R}^{d_1}$, and $h(\boldsymbol{\lambda}) = 0$), as well as proximal (stochastic) gradient descent ($\phi(\boldsymbol{\lambda}) = \frac{1}{2} \|\boldsymbol{\lambda}\|^2$ and $\mathcal{X} = \mathbb{R}^{d_1}$). The aforementioned gradient step in equation 4 can be further extended to a time-varying distance generating function, e.g., $\phi_t(\boldsymbol{\lambda}) = \frac{1}{2}\boldsymbol{\lambda}^T \mathbf{H}_t \boldsymbol{\lambda}$ with an adaptive matrix $\mathbf{H}_t$, resulting in an adaptive proximal gradient method with similarities to Adagrad from Duchi et al. (2011) and Super-Adam of Huang et al. (2021b). The proximal gradient step of equation 4 has led to the introduction of a generalized projection from Ghadimi et al. (2016) defined for a step size $\alpha > 0$, $\boldsymbol{q} \in \mathbb{R}^{d_1}$, and $\boldsymbol{u} \in \mathcal{X}$ as $\mathcal{G}_{\mathcal{X}}(\boldsymbol{u}, \boldsymbol{q}, \alpha) := \frac{1}{\alpha}(\boldsymbol{u} - \boldsymbol{\lambda}^+)$. Here $\mathcal{G}_{\mathcal{X}}(\boldsymbol{\lambda}, \nabla f_t(\boldsymbol{\lambda}), \alpha)$ acts as a generalized gradient that simplifies to $\nabla f_t(\boldsymbol{\lambda})$ if $\mathcal{X} = \mathbb{R}^{d_1}$ and $h(\boldsymbol{\lambda}) = 0$.

### 2.3 Generalized Bilevel Local Regret

Bilevel local regret is a stationary metric for online bilevel optimization Tarzanagh et al. (2024); Lin et al. (2024) that extends the single-level local regret measure from Hazan et al. (2017). The work of Lin et al. (2024) in particular defines the bilevel local regret for a window length $w \geq 1$ and a sequence $\{\boldsymbol{\lambda}_t\}_{t=1}^T$ as $BLR_w(T) := \sum_{t=1}^T \|\nabla F_{t,w}(\boldsymbol{\lambda}_t)\|^2$ where for simplicity we have defined $F_{t,w}(\boldsymbol{\lambda}_t) := \frac{1}{w} \sum_{i=0}^{w-1} F_{t-i}(\boldsymbol{\lambda}_{t-i})$ as a time-smoothed outer level objective with $F_t = 0 \ \forall t \leq 0$. Note for the online stochastic bilevel formulation of equation 3, the bilevel local regret can be equivalently written as $BLR_w(T) := \sum_{t=1}^T \|\nabla F_{t,w}(\boldsymbol{\lambda}_t)\|^2 == \sum_{t=1}^T \left\| \left( \frac{1}{w} \sum_{i=0}^{w-1} \mathbb{E}_\epsilon \left[ \nabla_{\boldsymbol{\lambda}} f_{t-i}(\boldsymbol{\lambda}, \boldsymbol{\beta}_{t-i}^*(\boldsymbol{\lambda}_{t-i}), \epsilon) \right] \right) \right\|^2$. To analyze convergence benefits from the Bregman-based gradient step of equation 4 in online bilevel optimization algorithms, we introduce a new generalized projection based bilevel local regret as

$$BLR_w(T) := \sum_{t=1}^T \|\mathcal{G}_{\mathcal{X}}(\boldsymbol{\lambda}_t, \nabla F_{t,w}(\boldsymbol{\lambda}_t), \alpha)\|^2 \qquad (5)$$

where $w \geq 1$ is the window length, and $\{\boldsymbol{\lambda}_t\}_{t=1}^T$ is the sequence of iterative updates generated. Note that in the setting where $\mathcal{X} = \mathbb{R}^{d_1}$, $h(\boldsymbol{\lambda}) = 0$, and $\phi_t(\boldsymbol{\lambda}) = \phi(\boldsymbol{\lambda}) = \frac{1}{2} \|\boldsymbol{\lambda}\|^2$, our variation of local regret in equation 5 reduces to the regret measure of Lin et al. (2024). However, our definition offers an important generalization of bilevel local regret when an adaptive distance generating function $\phi_t(\boldsymbol{\lambda})$ or a non-zero regularization term $h(\boldsymbol{\lambda})$ is present.

Besbes et al. (2015) shows that in order to derive useful regret bounds of online algorithms in time-varying environments further regularity constraints must be imposed on the sequence, such as sublinear comparator sequences. Example comparator sequences include path variation Yang et al. (2016), function variation Besbes et al. (2015), or gradient variation Chiang et al. (2012). In online bilevel optimization one proposed sequence is the $p$-th order inner level path variation of optimal decisions from Tarzanagh et al. (2024), and is $H_{p,T} := \sum_{t=2}^T \sup_{\boldsymbol{\lambda} \in \mathcal{X}} \|\boldsymbol{\beta}_{t-1}^*(\boldsymbol{\lambda}) - \boldsymbol{\beta}_t^*(\boldsymbol{\lambda})\|^p$. A regularity metric on the $p$-th order variation of the evaluations of the outer level function across time is suggested by Lin et al. (2024) and is $V_{p,T} := \sum_{t=1}^T \sup_{\boldsymbol{\lambda} \in \mathcal{X}} |F_{t+1}(\boldsymbol{\lambda}) - F_t(\boldsymbol{\lambda})|^p$. Note the latter regularity metric, $V_{p,T}$, tracks how the optimal outer level variable, which is fixed for a given $t \in [1, T]$, can vary over time. For the online stochastic bilevel formulation of equation 3, the aforementioned comparator sequences can be equivalently written as $H_{p,T} := \sum_{t=2}^T \sup_{\boldsymbol{\lambda} \in \mathcal{X}} \|\boldsymbol{\beta}_{t-1}^*(\boldsymbol{\lambda}) - \boldsymbol{\beta}_t^*(\boldsymbol{\lambda})\|^p$ and $V_{p,T} := \sum_{t=1}^T \sup_{\boldsymbol{\lambda} \in \mathcal{X}} |\mathbb{E}_\epsilon [f_{t+1}(\boldsymbol{\lambda}, \boldsymbol{\beta}_{t+1}^*(\boldsymbol{\lambda}), \epsilon)] - \mathbb{E}_\epsilon [f_t(\boldsymbol{\lambda}, \boldsymbol{\beta}_t^*(\boldsymbol{\lambda}), \epsilon)]|^p$. We will be utilizing the regularity metrics of second-order inner-level path variation, $H_{2,T}$, and first-order variation of the evaluations of the outer level objective, $V_{1,T}$, and impose a sublinear constraint, that is $H_{2,T} = o(T)$ and $V_{1,T} = o(T)$.

## 3 Bregman Bilevel Optimization under Hypergradient Oracle

The hypergradient in the online setting has been formally derived using the chain rule followed by an implicit function theorem by Lin et al. (2024); Tarzanagh et al. (2024). Namely,

**Lemma 1.** *(Tarzanagh et al. (2024)) Under Assumptions A and B, we have $\forall \boldsymbol{\lambda} \in \mathcal{X}$*

$$\nabla F_t(\boldsymbol{\lambda}) = \nabla_{\boldsymbol{\lambda}} f_t(\boldsymbol{\lambda}, \boldsymbol{\beta}_t^*(\boldsymbol{\lambda})) + \nabla \boldsymbol{\beta}_t^*(\boldsymbol{\lambda}) \nabla_{\boldsymbol{\beta}} f_t(\boldsymbol{\lambda}, \boldsymbol{\beta}_t^*(\boldsymbol{\lambda}))$$

$$= \nabla_{\boldsymbol{\lambda}} f_t(\boldsymbol{\lambda}, \boldsymbol{\beta}_t^*(\boldsymbol{\lambda})) - \nabla_{\boldsymbol{\lambda},\boldsymbol{\beta}}^2 g_t(\boldsymbol{\lambda}, \boldsymbol{\beta}_t^*(\boldsymbol{\lambda})) \left( \nabla_{\boldsymbol{\beta},\boldsymbol{\beta}}^2 g_t(\boldsymbol{\lambda}, \boldsymbol{\beta}_t^*(\boldsymbol{\lambda})) \right)^{-1} \nabla_{\boldsymbol{\beta}} f(\boldsymbol{\lambda}, \boldsymbol{\beta}_t^*(\boldsymbol{\lambda})). \quad (6)$$

The above gradient decomposition is a common expansion in bilevel optimization that utilizes the smoothness and strong convexity assumptions A and B. The next Lemma provides an upper bound on the difference in the evaluated hypergradient $\|\nabla F_t(\boldsymbol{\lambda}_1) - \nabla F_t(\boldsymbol{\lambda}_2)\|$ in terms of the Lipschitz constant $\ell_{F,1}$.

**Lemma 2.** *(Lemma 3 in Tarzanagh et al. (2024)) Under assumptions A and B, it holds that, for all $\boldsymbol{\lambda}_1, \boldsymbol{\lambda}_2 \in \mathcal{X}$, $\|\nabla F_t(\boldsymbol{\lambda}_1) - \nabla F_t(\boldsymbol{\lambda}_2)\| \leq \ell_{F,1} \|\boldsymbol{\lambda}_1 - \boldsymbol{\lambda}_2\|$, where the constant $\ell_{F,1} = O(\kappa_g^3)$ is dependent on the condition number $\kappa_g$, strong convexity parameter $\mu_g$, and Lipschitz constants $\ell_{f,1}, \ell_{f,0}, \ell_{g,2}$, see Lemma equation 11 for full analytical form of $\ell_{F,1}$.*

In order to analyze the effect of generalized Bregman-based gradient steps in online bilevel optimization, we introduce the hypergradient oracle. This obviates the need for a choice of the hypergradient estimation and allows us to show the independent improvement of the rate of bilevel local regret.

**Definition 1.** *The hypergradient oracle is a function $\mathcal{O}(\boldsymbol{\lambda})$ that returns the true hypergradient $\mathcal{O}(\boldsymbol{\lambda}) : \boldsymbol{\lambda} \mapsto \nabla F_t(\boldsymbol{\lambda})$, where $\nabla F_t(\boldsymbol{\lambda}) = \nabla_{\boldsymbol{\lambda}} F_t(\boldsymbol{\lambda}, \boldsymbol{\beta}_t^*(\boldsymbol{\lambda})) + \nabla_{\boldsymbol{\lambda}} \boldsymbol{\beta}_t^*(\boldsymbol{\lambda}) \nabla_{\boldsymbol{\beta}} F_t(\boldsymbol{\lambda}, \boldsymbol{\beta}_t^*(\boldsymbol{\lambda}))$.*

The oracle has access to the true hypergradient at optimal inner level variables $\boldsymbol{\beta}_t^*(\boldsymbol{\lambda}) \, \forall \boldsymbol{\lambda} \in \mathcal{X}$. Algorithm 1 employs the hypergradient oracle, and, together with the Bregman-based step implemented as a subroutine in Algorithm 2, constitutes a special case of our general algorithm, to be introduced later. Sections 4 and 5 present this general algorithm and complementary regret analysis for the generalized Bregman-based gradient step in the practical setting which requires hypergradient estimation. With

---

**Algorithm 1** Bregman Optimizer

**Require:** Initial variable $\boldsymbol{\lambda}_1 \in \mathcal{X}$, step size $\alpha > 0$, Bregman reference function $\phi$
1: **for** $t = 1, \ldots, T$ **do**
2: $\quad \nabla F_t(\boldsymbol{\lambda}_t) \leftarrow \mathcal{O}(\boldsymbol{\lambda}_t) \triangleright$ Query oracle
3: $\quad \boldsymbol{u} \leftarrow \boldsymbol{\lambda}_t$
4: $\quad \boldsymbol{q} \leftarrow \nabla F_t(\boldsymbol{\lambda}_t)$
5: $\quad \boldsymbol{\lambda}_{t+1} \leftarrow$ Algorithm 2 for $\boldsymbol{u}, \boldsymbol{q}, \alpha, \phi$
6: **end for**
7: **return** $\boldsymbol{\lambda}_{T+1}$

---

**Algorithm 2** Generalized Gradient Step

**Require:** $\boldsymbol{u}, \boldsymbol{q}$, step size $\alpha$, reference function $\phi$
1: $\boldsymbol{\lambda}^+ \leftarrow \arg\min_{\boldsymbol{\lambda} \in \mathcal{X}} \{\langle \boldsymbol{q}, \boldsymbol{\lambda} \rangle + h(\boldsymbol{\lambda}) + \frac{1}{\alpha} \mathcal{D}_\phi(\boldsymbol{\lambda}, \boldsymbol{u})\}$
2: **return** $\boldsymbol{\lambda}^+$

---

the oracle, Theorem 3 shows that Algorithm 1 with generalized Bregman steps achieves regret $o(T)$, compared to $o(\kappa_g^3 T)$ for classical gradient descent ($\mathcal{X} = \mathbb{R}^{d_1}, h(\boldsymbol{\lambda}) = 0, \phi(\boldsymbol{\lambda}) = \frac{1}{2}\|\boldsymbol{\lambda}\|^2$). The improvement comes from eliminating the multiplicative $\kappa_g^3 > 1$ factor. For background on the role of condition numbers in bilevel optimization, see Huang et al. (2022b).

**Theorem 3** (Bregman Steps under Hypergradient Oracle). *Suppose $h(\boldsymbol{\lambda}) = 0$ and $\mathcal{X} = \mathbb{R}^{d_1}$. Assume Assumptions A–F hold and that the cumulative variation is sublinear:*

$$\sum_{t=1}^{T} \left( F_t(\boldsymbol{\lambda}_t) - F_{t+1}(\boldsymbol{\lambda}_{t+1}) \right) \leq o(T).$$

*If Algorithm 1 uses a $\rho$-strongly convex reference function $\phi(\boldsymbol{\lambda})$ defining the Bregman divergence $D_\phi(\boldsymbol{\lambda}, \boldsymbol{u})$ with $\rho = O(\ell_{F,1})$, then the bilevel local regret (with $w = 1$) satisfies*

$$\sum_{t=1}^{T} \left\| \mathcal{G}_{\mathcal{X}}(\boldsymbol{\lambda}_t, \nabla F_t(\boldsymbol{\lambda}_t), \alpha) \right\|^2 \leq o(T),$$

*i.e., a sublinear rate independent of the condition number $\kappa_g > 1$.*

*Remark.* The result indicates that generalized Bregman steps eliminate the condition-number dependence present in Euclidean updates.

**Corollary 3.1** (Classical Gradient Descent as Euclidean Bregman Step)**.** *Let $\phi(\boldsymbol{\lambda}) = \frac{1}{2}\|\boldsymbol{\lambda}\|^2$, so that $\mathcal{D}_\phi(\boldsymbol{\lambda}, \boldsymbol{u}) = \frac{1}{2}\|\boldsymbol{\lambda} - \boldsymbol{u}\|^2$ and $\rho = 1$. Then Algorithm 1 reduces to classical gradient descent and the bilevel local regret satisfies*

$$\sum_{t=1}^T \left\| \mathcal{G}_{\mathcal{X}}(\boldsymbol{\lambda}_t, \nabla F_t(\boldsymbol{\lambda}_t), \alpha) \right\|^2 = \sum_{t=1}^T \|\nabla F_t(\boldsymbol{\lambda}_t)\|^2 \le o(\ell_{F,1} T) = o(\kappa_g^3 T).$$

Theorem 3 shows the improvement a Bregman-based gradient step can have on the sublinear rate of bilevel local regret of Algorithm 1 in terms of the condition number $\kappa_g > 1$. Next, we extend our Bregman algorithm and analysis to the setting where hypergradient estimation is required.

## 4 BREGMAN BILEVEL OPTIMIZATION UNDER HYPERGRADIENT ESTIMATION

### 4.1 STOCHASTIC HYPERGRADIENT ESTIMATION

Following previous work on bilevel optimization Ghadimi & Wang (2018); Tarzanagh et al. (2024); Lin et al. (2024), the computational difficulty in obtaining $\boldsymbol{\beta}_t^*(\boldsymbol{\lambda})$ motivates the use of a surrogate $\boldsymbol{\beta}$ in the hypergradient expansion of equation 6 for a fixed $\boldsymbol{\lambda} \in \mathcal{X}$ and $\boldsymbol{\beta} \in \mathbb{R}^{d_2}$ as $\widetilde{\nabla} f_t(\boldsymbol{\lambda}, \boldsymbol{\beta}) := \nabla_{\boldsymbol{\lambda}} f_t(\boldsymbol{\lambda}, \boldsymbol{\beta}) - \nabla_{\boldsymbol{\lambda}, \boldsymbol{\beta}}^2 g_t(\boldsymbol{\lambda}, \boldsymbol{\beta}) \left( \nabla_{\boldsymbol{\beta}, \boldsymbol{\beta}}^2 g_t(\boldsymbol{\lambda}, \boldsymbol{\beta}) \right)^{-1} \nabla_{\boldsymbol{\beta}} f(\boldsymbol{\lambda}, \boldsymbol{\beta})$. Note by further considering the stochastic setting, the hypergradient is composed of first and second-order stochastic gradients, for which we have unbiased oracles with finite variances under Assumption C. However a stochastic estimator is still required for the inverse Hessian $\left( \nabla_{\boldsymbol{\beta}, \boldsymbol{\beta}}^2 g_t(\boldsymbol{\lambda}, \boldsymbol{\beta}, \zeta) \right)^{-1}$. A common stochastic estimator for the inverse Hessian has been proposed by Ghadimi & Wang (2018), and used in Khanduri et al. (2021) and Huang et al. (2022b). We use the aforementioned stochastic hypergradient estimate in this work and denote the estimate as $\widetilde{\nabla} f_t(\boldsymbol{\lambda}_t, \boldsymbol{\beta}_{t+1}, \mathcal{E}_t)$. Construction of the stochastic hypergradient estimate is included in Lemma 4.

**Lemma 4.** *(Algorithm 3 in Ghadimi & Wang (2018)) Suppose Assumptions A, B, and C. Then for an upper bound of $m$, learning rate $\tilde{\eta}$, and independent samples $\mathcal{E} = \{\epsilon, \zeta^0, \ldots, \zeta^{m-1}\}$, the stochastic gradient of $\widetilde{\nabla} f_t(\boldsymbol{\lambda}, \boldsymbol{\beta}, \mathcal{E})$ provides an estimate of $\widetilde{\nabla} f_t(\boldsymbol{\lambda}, \boldsymbol{\beta})$ and is constructed via Algorithm 5*

$$\widetilde{\nabla} f_t(\boldsymbol{\lambda}, \boldsymbol{\beta}, \mathcal{E}) := \nabla_{\boldsymbol{\lambda}} f_t(\boldsymbol{\lambda}, \boldsymbol{\beta}, \epsilon) - \nabla_{\boldsymbol{\lambda}, \boldsymbol{\beta}}^2 g_t(\boldsymbol{\lambda}, \boldsymbol{\beta}, \zeta^0)$$

$$\times \left[ \frac{m}{\tilde{\eta}} \prod_{j=1}^{\widetilde{m}} \left( I_{d_2} - \frac{1}{\tilde{\eta}} \nabla_{\boldsymbol{\beta}}^2 g_t(\boldsymbol{\lambda}, \boldsymbol{\beta}, \zeta^j) \right) \right] \nabla_{\boldsymbol{\beta}} f_t(\boldsymbol{\lambda}, \boldsymbol{\beta}, \epsilon), \tag{7}$$

*where $\widetilde{m} \sim \mathcal{U}(0, 1, \ldots, m - 1)$ and, for $m = 0$, $\prod_{j=1}^m (\cdot) = I_{d_2}$.*

The next Lemma from Khanduri et al. (2021) characterizes the bias of this stochastic estimate.

**Lemma 5.** *(Lemma B.1 in Khanduri et al. (2021)) Suppose Assumptions A,B, and C. For any $m \ge 1$ the gradient estimator of equation 7 satisfies the bias of $B(\boldsymbol{\lambda}, \boldsymbol{\beta}) := \left\| \widetilde{\nabla} f_t(\boldsymbol{\lambda}, \boldsymbol{\beta}) - \mathbb{E}_{\mathcal{E}} \left[ \widetilde{\nabla} f_t(\boldsymbol{\lambda}, \boldsymbol{\beta}, \mathcal{E}) \right] \right\| \le \ell_{f,1} \kappa_g \left( 1 - \frac{\mu_g}{\ell_{g,1}} \right)^m$*

### 4.2 SINGLE-LOOP EFFICIENCY WITH TIME-SMOOTHING

Due to the computational cost double-loop algorithms can incur, single-loop algorithms for bilevel optimization are often desired. However in the stochastic case, variability of stochastic gradients impose a difficulty for the construction of single-loop algorithms. One proposed solution commonly employed, see Khanduri et al. (2021) and Huang et al. (2021a), is the use of momentum techniques to achieve variance reduction. A similar methodology appears in the online deterministic setting of Lin et al. (2024) where the technique of time-smoothing is applied to average evaluated hypergradients and improve the rate of bilevel local regret.

Motivated by the success of momentum techniques in stochastic bilevel optimization and their technical similarity to time-smoothing, we employ time-smoothing from Lin et al. (2024) to efficiently average the evaluated stochastic hypergradients. In particular, we introduce time-smoothing

with the estimator $\widetilde{\nabla} f_{t,w}(\boldsymbol{\lambda}_t, \boldsymbol{\beta}_{t+1}, \mathcal{Z}_{t,w})$ defined for all $t \in [1, T]$, window size $w \geq 1$, and independent samples $\mathcal{Z}_{t,w} = \{\mathcal{E}_{t-i}\}_{i=0}^{w-1}$ where $\mathcal{E}_t = \{\epsilon_t, \zeta_t^0, \ldots, \zeta_t^{m-1}\}$ as $\widetilde{\nabla} f_{t,w}(\boldsymbol{\lambda}_t, \boldsymbol{\beta}_{t+1}, \mathcal{Z}_{t,w}) :=$ $\frac{1}{w} \sum_{i=0}^{w-1} \widetilde{\nabla} f_{t-i}(\boldsymbol{\lambda}_{t-i}, \boldsymbol{\beta}_{t+1-i}, \mathcal{E}_{t-i}), \quad f_t = 0 \ \forall t \leq 0.$

Our general algorithm is included in Algorithm 3 and states our novel Bregman bilevel optimizer that efficiently utilizes stochastic hypergradient estimation with time-smoothing techniques to solve the online stochastic bilevel optimization problem of equation 3. The special case of it, for $K = 1$, is an efficient single-loop algorithm. In the next section (Section 5), we further show how the above time-smoothing technique has the effect of variance reduction on the rate of regret.

---

**Algorithm 3** Stochastic Online Bregman Bilevel Optimizer

---

**Require:** Horizon $T$; inner steps $K \geq 1$; step sizes $\alpha, \eta > 0$; batch sizes $s, m$; Bregman reference $\phi$; window $w \geq 1$
1: Initialize $\boldsymbol{\beta}_1 \in \mathbb{R}^{d_2}$, $\boldsymbol{\lambda}_1 \in \mathcal{X}$
2: **for** $t = 1 \ T$ **do**
3:      $\boldsymbol{\omega}_t^0 \leftarrow \boldsymbol{\beta}_t$
4:      **for** $k = 1 \ K$ **do**
5:          Sample $s$ i.i.d. draws of $\zeta$; set $\bar{\zeta}_{t,k} \leftarrow \{\zeta_{t,i}^{k-1}\}_{i=1}^s$
6:          $\boldsymbol{\omega}_t^k \leftarrow \boldsymbol{\omega}_t^{k-1} - \eta \nabla_{\boldsymbol{\omega}} g_t(\boldsymbol{\lambda}_t, \boldsymbol{\omega}_t^{k-1}, \bar{\zeta}_{t,k})$
7:      **end for**
8:      $\boldsymbol{\beta}_{t+1} \leftarrow \boldsymbol{\omega}_t^K$
9:      $\widetilde{\nabla} f_t(\boldsymbol{\lambda}_t, \boldsymbol{\beta}_{t+1}, \mathcal{E}_t) \leftarrow \text{STOCHHYPERGRAD}(\boldsymbol{\lambda}_t, \boldsymbol{\beta}_{t+1}, \eta, m)$          $\triangleright$ Alg. 5
10:     Store $\widetilde{\nabla} f_t(\boldsymbol{\lambda}_t, \boldsymbol{\beta}_{t+1}, \mathcal{E}_t)$ in memory
11:     $\boldsymbol{q} \leftarrow \widetilde{\nabla} f_{t,\boldsymbol{w}}(\boldsymbol{\lambda}_t, \boldsymbol{\beta}_{t+1}, \mathcal{Z}_{t,w})$ via time-smoothing
12:     $\boldsymbol{u} \leftarrow \boldsymbol{\lambda}_t$
13:     $\boldsymbol{\lambda}_{t+1} \leftarrow \text{GENERALGRADSTEP}(\boldsymbol{u}, \boldsymbol{q}, \alpha, \phi)$          $\triangleright$ Alg. 2
14: **end for**
15: **return** $\boldsymbol{\lambda}_{T+1}, \boldsymbol{\beta}_{T+1}$

---

## 5 CONVERGENCE ANALYSIS

To analyze the bilevel local regret of Algorithm 3, we require a bound on the error introduced by stochastic hypergradient estimation. In the oracle setting (Theorem 3), this error is absent, but in the stochastic case it contributes an additional term to the regret. Lemma 16 (Appendix) provides such a bound: it decomposes the cumulative hypergradient error into contributions from (i) past bilevel local regret, (ii) time-smoothed hypergradient error, (iii) variations in the optimal inner solutions, and (iv) the variance $\sigma_g^2$ of stochastic inner gradients. Building on this decomposition, we now establish the main regret bound for Algorithm 3.

**Theorem 6.** *(Proof in Appendix: Theorem 17) Suppose Assumptions A-F. Let the inner step be* $\eta = \Omega(1/\mu_g)$ *such that* $\eta \leq \min\{\frac{2}{\ell_{g,1}+\mu_g}, \frac{1}{w}\}$, *outer step be* $\alpha \leq \min\{\frac{3\rho}{4\ell_{F,1}}, \frac{\rho\sqrt{(1-\nu)}}{\kappa_g^2\sqrt{72C\mu_g}}\}$, *and batch size for stochastic inverse Hessian approximation* $m = \log(w)/\log\left(1 - \frac{\mu_g}{\ell_{g,1}}\right) + 1$. *Then the bilevel local regret of the **single-loop** ($K = 1$) and **sample-efficient** ($s = O(1)$) Algorithm 3 satisfies*

$$BLR_w(T) \leq O\left(\frac{T\kappa_g^3}{w\rho}\left(1 + \frac{\kappa_g^2 + \sigma_f^2 + \kappa_g^2\sigma_g^2}{\rho}\right) + \frac{V_{1,T}}{\rho} + \frac{\kappa_g^2 H_{2,T}}{\rho^2}\right) \quad (8)$$

*with comparator sequences $V_{1,T}$ and $H_{2,T}$, $\sigma_g^2, \sigma_f^2$ are finite variances from Assumption C, and $\kappa_g > 1$ is the condition number of the inner level objective.*

The next corollary highlights the improvement with a Bregman-based gradient step to equation 4.

**Corollary 6.1** (Effect of Bregman Steps). *As in Theorem 12, selecting $\rho = O(\ell_{F,1}) = O(\kappa_g^3)$ implies the bilevel local regret of*

$$BLR_w(T) \leq O\left(\frac{T}{w}\left(1 + \sigma_f^2 + \sigma_g^2\right) + V_{1,T} + H_{2,T}\right) \quad (9)$$

*where with gradient descent ($\rho = 1$), the rate is increased by a constant factor of $\kappa_g^5$.*

The next corollary highlights how time-smoothing with window length $w \geq 1$ is variance reduction.

**Corollary 6.2** (Variance Reduction via Windowing). *Increasing $w$ reduces the variance terms in the regret bound, as evidenced by*

$$BLR_w(T) \leq O\left( \frac{T\kappa_g^3}{w\rho}\left(1 + \frac{\kappa_g^2 + \sigma_f^2 + \kappa_g^2\sigma_g^2}{\rho}\right) + \frac{V_{1,T}}{\rho} + \frac{\kappa_g^2 H_{2,T}}{\rho^2} \right) \tag{10}$$

*where larger $w$ leads to a lower contribution of variance terms $\sigma_f^2$ and $\sigma_g^2$ to the regret.*

The next corollary as in deterministic online bilevel optimization problems (Lin et al. (2024)) considers sublinear comparator sequences, e.g., $V_{1,T} = o(T)$ and $H_{2,T} = o(T)$. For a properly chosen window of $w = o(T)$ note the rate of regret is sublinear, i.e. $BLR_w(T) = o(T)$.

**Corollary 6.3** (Sublinear Regret with Sublinear Comparators). *If $w = o(T)$, and the comparator sequences satisfy $V_{1,T} = o(T)$ and $H_{2,T} = o(T)$, then the regret bound*

$$BLR_w(T) \leq O\left( \frac{T\kappa_g^3}{w\rho}\left(1 + \frac{\kappa_g^2 + \sigma_f^2 + \kappa_g^2\sigma_g^2}{\rho}\right) + \frac{V_{1,T}}{\rho} + \frac{\kappa_g^2 H_{2,T}}{\rho^2} \right) \tag{11}$$

*ensures that $BLR_w(T) = o(T)$, implying a sublinear regret rate.*

**Corollary 6.4** (Window-Free Sublinear Regret). *Run with $w = 1$ and select Bregman Divergence such that $\rho = T^\alpha$ for any $\alpha \in (0,1)$. Then*

$$BLR_w(T) \leq O\left( T^{1-\alpha}\left(1 + \sigma_f^2 + \sigma_g^2\right) + \frac{V_{1,T}}{T^\alpha} + \frac{H_{2,T}}{T^{2\alpha}} \right), \tag{12}$$

*so $BLR(T) = o(T)$ or equivalently achieves sublinear bilevel local regret without time-smoothing.*

# 6 EXPERIMENTS

## 6.1 PRECONDITIONER LEARNING

**Task.** Adaptive methods (e.g., AdaGrad Duchi et al. (2011)) use data–dependent preconditioners but require hand-crafted choices. We instead learn a diagonal preconditioner online via bilevel optimization. At round $t$, we set $P(\lambda) = \text{diag}(\lambda) \succ 0$ and couple: (i) an inner preconditioned proximal update $\beta_t^*(\lambda) = \arg\min_\beta \mathbb{E}_{\zeta_t}[L_{\text{tr},t}(\beta; \zeta_t) + \frac{\gamma}{2}\|\beta - \beta_{t-1}\|_{P(\lambda)^{-1}}^2]$; and (ii) an outer update $\lambda_t^* = \arg\min_{\lambda \in \mathcal{X}} \mathbb{E}_{\epsilon_t}[L_{\text{val},t}(\beta_t^*(\lambda); \epsilon_t)]$. This shapes the inner geometry so training transfer improves validation, with $\beta$ and $\lambda$ both adapting online. **Models.** We use the same bilevel structure (inner: preconditioned proximal objective; outer: validation loss) across: (i) *quadratic regression* with diagonal $P(\lambda)$; and (ii) *linear classification* with smoothed hinge/logistic loss and the same proximal term. **Datasets.** We evaluate on the GDSC drug–response dataset in a high-dimensional regime ($p \gg n$) for both regression and classification; to stress nonstationarity and conditioning, we also use an imbalanced variant with three feature-based cohorts (marked vertical lines) and standard train/validation streams. **Baselines.** We compare **SOBBO** to deterministic online bilevel methods **OBBO**, **SOBOW** Lin et al. (2024), **OAGD** Tarzanagh et al. (2024), and our stochastic online bilevel method **SOBO**. Deterministic baselines (**SOBOW** and **OAGD**) compute hypergradients from full batches; **SOBO** uses stochastic mini-batched hypergradients and Euclidean steps. **SOBBO** performs outer steps with the quadratic divergence, $\mathcal{D}_{\phi_t}(\lambda_2, \lambda_1) = \frac{1}{2}\|\lambda_2 - \lambda_1\|_{H_t}^2$, such that $\phi_t(\lambda) = \frac{1}{2}\lambda^T H_t \lambda$ with diagonal matrix $H_t$ updated adaptively, as in Adagrad, similar to Huang et al. (2022a).

**Results.** Figure 1 shows that **SOBBO** attains lower local regret relative to deterministic Euclidean baselines as well as our stochastic baseline with Euclidean gradient descent. The middle panel of Figure 1 shows that larger window parameters reduce both the regret incurred and the variance of stochastic hypergradient estimates as we theoretically show in Corollary 6.2, a large improvement from the setting of SOBBO when $w = 1$. For the task of preconditioner learning, Figure 2 illustrates the difference in the learned optimizers across algorithms and window sizes, and the resulting validation improvements. **SOBBO** achieves the best validation loss (0.7214), outperforming **OBBO** (0.7880), **OAGD** (1.0110), and **SOBOW** (1.5291).

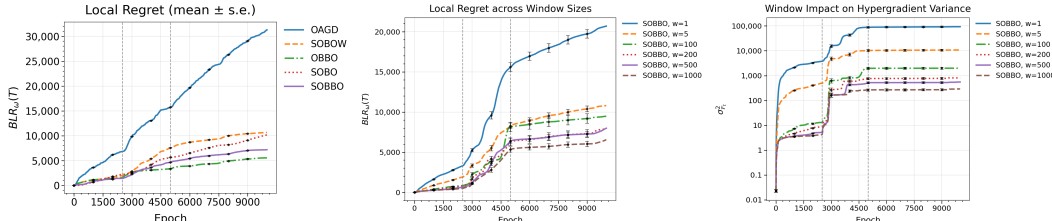

**Figure 1: Left:** Regret for deterministic and stochastic online bilevel optimizers; **SOBBO** attains smaller regret. **Middle:** Increasing window size reduces regret by stabilizing updates. **Right:** Larger window size reduces the variance of stochastic hypergradient estimates, as shown theoretically in Corollary 6.2.

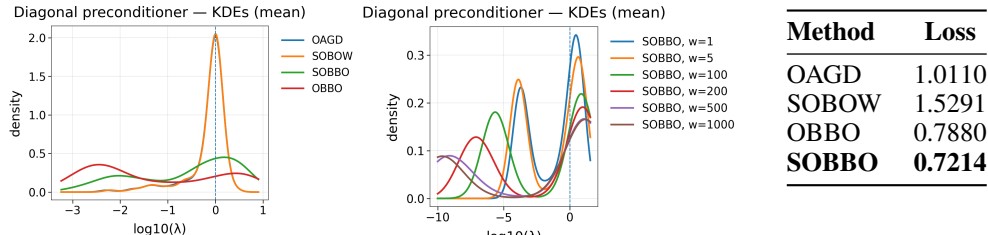

| Method | Loss |
|--------|--------|
| OAGD | 1.0110 |
| SOBOW | 1.5291 |
| OBBO | 0.7880 |
| **SOBBO** | **0.7214** |

**Figure 2: Left:** KDE of diagonal preconditioner entries ($d = 500$) across algorithms. **Middle:** KDE for different window sizes, showing the effect of time-smoothing. **Right:** Final validation loss across methods.

### 6.2 ACTOR–CRITIC REINFORCEMENT LEARNING

**Task.** Following the formulation of Prakash et al. (2025) we cast actor–critic as *online bilevel optimization* with actor (outer) $\theta$ and critic (inner) $\omega$: the actor solves $\min_\theta \mathbb{E}_{\epsilon_t}[f_t(\theta, \omega_t^*(\theta), \epsilon_t)] + r(\theta)$ while the critic solves $\omega_t^*(\theta) \in \arg\min_\omega \mathbb{E}_{\zeta_t}[g_t(\theta, \omega, \zeta_t)]$. **Models.** The actor $\pi_\theta$ is a 2-layer MLP (128–128), the critic $Q_\omega$ is a matching MLP with TD updates. **Datasets.** We consider a nonstationary Pendulum environment within Gymnasium with scheduled nonstationarity jumps occurring in the gravity and max torque, see left panel of Figure 3. **Baselines.** We compare our algorithm SOBBO, using the quadratic divergence $\mathcal{D}_{\phi_t}(\lambda_2, \lambda_1) = \frac{1}{2}||\lambda_2 - \lambda_1||_{H_t}^2$, against **SOBOW** and **OAGD** adapted to this RL setting, measuring bilevel local regret and the effect of window size.

**Results**: Figure 3 shows lower bilevel local regret of **SOBBO** relative to deterministic and stochastic baselines with increasing window parameter ($w = 50, 500, 5000$) further reducing regret.

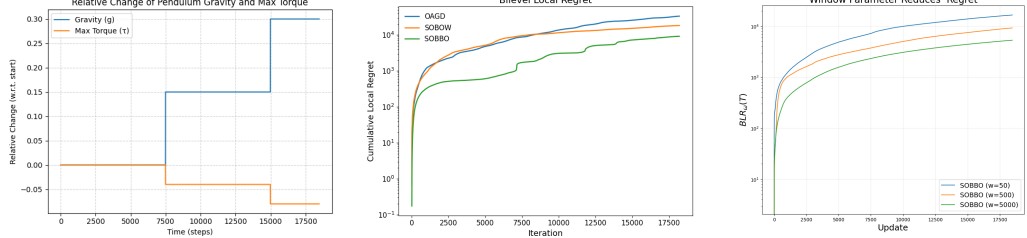

**Figure 3:** Bilevel RL experiments. **(a)** Relative changes in the Pendulum environment. **(b)** Bilevel local regret across algorithms; **SOBBO** achieves lowest. **(c)** Bilevel local regret over window sizes shows improved regret.

## 7 CONCLUSION

This work shows that Bregman geometry removes the dependence on the inner condition number in stochastic online bilevel optimization while attaining sublinear bilevel local regret. We develop single-loop, sample-efficient algorithms that couple generalized Bregman steps with time-smoothed hypergradient estimates, and our analysis identifies smoothing as an intrinsic variance-reduction mechanism that controls bias and unifies prior heuristics with theory. Empirical results corroborate these claims across ill-conditioned, large-scale datasets and nonstationary losses.

## 7.1 REPRODUCIBILITY STATEMENT

To ensure reproducibility, Sections 3 to 5 present the key algorithmic details of our Bregman-based bilevel optimizers. Section 6 documents the experimental setup, baselines, hyperparameters, and the open-source datasets used. Upon acceptance, we will release an open-source repository with implementations, configurations, and scripts to reproduce all experiments. All assumptions and complete proofs, including regret bounds in the stochastic setting (Appendix B) and the reduction to the deterministic setting (Appendix C), are provided in the appendices.

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

## A    PRELIMINARIES

**Lemma 7.** *(Lemma 12 in Tarzanagh et al. (2024)) For any set of vectors $\{\boldsymbol{\beta}_i\}_{i=1}^m$, it holds that*

$$\left\| \sum_{i=1}^m \boldsymbol{\beta}_i \right\|^2 \leq m \sum_{i=1}^m \|\boldsymbol{\beta}_i\|^2 \tag{13}$$

The following lemma provides progress bounds for gradient descent applied to a $\mu_g$-strongly convex and twice differentiable function $g(\boldsymbol{\beta})$.

**Lemma 8.** *Let $g(\boldsymbol{\omega})$ be a twice differentiable and $\mu_g$-strongly convex function with $\nabla g(\boldsymbol{\omega})$ satisfying $\ell_{g,1}$-Lipschitz continuity. Further assume $g(\boldsymbol{\omega})$ has a global minimizer $\widehat{\boldsymbol{\omega}}$ over the domain $\mathbb{R}^{d_2}$. Then under the gradient descent method of*

$$\boldsymbol{\omega}^k = \boldsymbol{\omega}^{k-1} - \eta \nabla g(\boldsymbol{\omega}^{k-1}),$$

*the following satisfies for $\eta \leq \frac{1}{\ell_{g,1}}$*

$$\left\| \boldsymbol{\omega}^k - \widehat{\boldsymbol{\omega}} \right\|^2 \leq (1 - \eta \mu_g) \left\| \boldsymbol{\omega}^{k-1} - \widehat{\boldsymbol{\omega}} \right\|^2,$$

The following two lemmas characterize useful properties known for the generalized projection $\mathcal{G}_{\mathcal{X}}(\boldsymbol{u}, \boldsymbol{q}, \alpha)$.

**Lemma 9.** *(Lemma 1 in Ghadimi et al. (2016)) Let $\boldsymbol{\lambda}^+$ be from equation 4. Then $\forall \boldsymbol{u} \in \mathcal{X}$, $\boldsymbol{q} \in \mathbb{R}^{d_1}$, and $\alpha > 0$ we have*

$$\langle \boldsymbol{q}, \mathcal{G}_{\mathcal{X}}(\boldsymbol{u}, \boldsymbol{q}, \alpha) \rangle \geq \rho \left\| \mathcal{G}_{\mathcal{X}}(\boldsymbol{u}, \boldsymbol{q}, \alpha) \right\|^2 + \frac{1}{\alpha} \left( h(\boldsymbol{\lambda}^+) - h(\boldsymbol{u}) \right) \tag{14}$$

*such that $\rho > 0$ is the strong convexity parameter of the distance generating function $\phi(\boldsymbol{\lambda})$.*

**Lemma 10.** *(Proposition 1 in Ghadimi et al. (2016)) Let $\mathcal{G}_{\mathcal{X}}(\boldsymbol{u}, \boldsymbol{q}, \alpha)$ be the generalized projection. Then $\forall \boldsymbol{q}_1, \boldsymbol{q}_2 \in \mathbb{R}^{d_1}$, $\forall \boldsymbol{u} \in \mathcal{X}$, $\forall \alpha > 0$, we have*

$$\left\| \mathcal{G}_{\mathcal{X}}(\boldsymbol{u}, \boldsymbol{q}_1, \alpha) - \mathcal{G}_{\mathcal{X}}(\boldsymbol{u}, \boldsymbol{q}_2, \alpha) \right\| \leq \frac{1}{\rho} \left\| \boldsymbol{q}_1 - \boldsymbol{q}_2 \right\|. \tag{15}$$

The next Lemma provides useful bounds on the hypergradient $\nabla F_t(\boldsymbol{\lambda})$, gradient estimate $\nabla f_t(\boldsymbol{\lambda}, \boldsymbol{\beta})$, and optimal inner level variables $\boldsymbol{\beta}_t^*(\boldsymbol{\lambda})$ in the deterministic online bilevel optimization problem.

**Lemma 11.** *(Lemma 3 in Tarzanagh et al. (2024)) Under assumptions A and B, it holds for all $t \in [1, T]$, $\boldsymbol{\lambda}_1, \boldsymbol{\lambda}_2 \in \mathcal{X}$, and $\boldsymbol{\beta} \in \mathbb{R}^{d_2}$ that*

$$\left\| \boldsymbol{\beta}_t^*(\boldsymbol{\lambda}_1) - \boldsymbol{\beta}_t^*(\boldsymbol{\lambda}_2) \right\| \leq \kappa_g \left\| \boldsymbol{\lambda}_1 - \boldsymbol{\lambda}_2 \right\|, \tag{16}$$

*where $\kappa_g := \frac{\ell_{g,1}}{\mu_g} = O(\kappa_g)$, the gradient estimator $\widetilde{\nabla} f_t(\boldsymbol{\lambda}, \boldsymbol{\beta})$ satisfies*

$$\left\| \widetilde{\nabla} f_t(\boldsymbol{\lambda}, \boldsymbol{\beta}) - \nabla F_t(\boldsymbol{\lambda}) \right\| \leq M_f \left\| \boldsymbol{\beta} - \boldsymbol{\beta}_t^*(\boldsymbol{\lambda}) \right\|, \tag{17}$$

*where $M_f := \ell_{f,1} + \ell_{f,1} \kappa_g + \frac{\ell_{f,0} \ell_{g,2}}{\mu_g} (1 + \kappa_g) = O(\kappa_g^2)$, and*

$$\left\| \nabla F_t(\boldsymbol{\lambda}_1) - \nabla F_t(\boldsymbol{\lambda}_2) \right\| \leq \ell_{F,1} \left\| \boldsymbol{\lambda}_1 - \boldsymbol{\lambda}_2 \right\|, \tag{18}$$

*where $\ell_{F,1} := \ell_{f,1}(1 + \kappa_g) + \frac{\ell_{f,0} \ell_{g,2}}{\mu_g}(1 + \kappa_g) + M_f \kappa_g = O(\kappa_g^3)$.*

## B    PROOF IN STOCHASTIC SETTING

The first theorem states the convergence result in the hypergradient oracle setting.

**Theorem 12.** *Suppose that $h(\boldsymbol{\lambda}) = 0$ and $\mathcal{X} = \mathbb{R}^{d_1}$. Additionally, let Assumptions A-F hold. If the cumulative difference of subsequent function evaluations satisfies the sublinearity condition:*

$$\sum_{t=1}^T \left( F_t(\boldsymbol{\lambda}_t) - F_{t+1}(\boldsymbol{\lambda}_{t+1}) \right) \leq o(T), \tag{19}$$

*then, by selecting a $\rho$-strongly convex reference function $\phi(\boldsymbol{\lambda})$ that uniquely defines the Bregman Divergence $D_\phi(\boldsymbol{\lambda}, \boldsymbol{u})$ such that $\rho = O(\ell_{F,1})$ in the generalized step of equation 4, the bilevel local regret (for $w = 1$) of Algorithm 1, using the generalized Bregman-based step from Algorithm 2, achieves a sublinear rate of $o(T)$, independent of the condition number $\kappa_g > 1$.*

*Proof.* We analyze the convergence of the Bregman optimizer in Algorithm 1, under the generalized Bregman-based gradient step of Algorithm 2. Note with Assumptions A and B, we apply Lemma 2 that says $F_t$ is $\ell_{F,1}$-smooth and implies that

$$F_t(\boldsymbol{\lambda}_{t+1}) - F_t(\boldsymbol{\lambda}_t) \leq \langle \nabla F_t(\boldsymbol{\lambda}_t), \boldsymbol{\lambda}_{t+1} - \boldsymbol{\lambda}_t \rangle + \frac{\ell_{F,1}}{2}\|\boldsymbol{\lambda}_{t+1} - \boldsymbol{\lambda}_t\|^2.$$

Substituting the generalized projection of $\mathcal{G}_{\mathcal{X}}(\boldsymbol{\lambda}_t, \nabla F_t(\boldsymbol{\lambda}_t), \alpha) := \frac{1}{\alpha}(\boldsymbol{\lambda}_t - \boldsymbol{\lambda}_{t+1})$ from the generalized Bregman-based gradient step of equation 4 gives us

$$F_t(\boldsymbol{\lambda}_{t+1}) - F_t(\boldsymbol{\lambda}_t) \leq \langle \nabla F_t(\boldsymbol{\lambda}_t), -\alpha\mathcal{G}_{\mathcal{X}}(\boldsymbol{\lambda}_t, \nabla F_t(\boldsymbol{\lambda}_t), \alpha) \rangle + \frac{\ell_{F,1}}{2}\|\alpha\mathcal{G}_{\mathcal{X}}(\boldsymbol{\lambda}_t, \nabla F_t(\boldsymbol{\lambda}_t), \alpha)\|^2. \quad (20)$$

Now applying Lemma 9, we obtain

$$\langle \nabla F_t(\boldsymbol{\lambda}_t), -\alpha\mathcal{G}_{\mathcal{X}}(\boldsymbol{\lambda}_t, \nabla F_t(\boldsymbol{\lambda}_t), \alpha) \rangle \leq -\rho\alpha\|\mathcal{G}_{\mathcal{X}}(\boldsymbol{\lambda}_t, \nabla F_t(\boldsymbol{\lambda}_t), \alpha)\|^2. \quad (21)$$

Substituting equation 21 into equation 20 and rearranging this inequality and telescoping we get

$$-\left(\rho\alpha - \frac{\ell_{F,1}\alpha^2}{2}\right)\sum_{t=1}^{T}\|\mathcal{G}_{\mathcal{X}}(\boldsymbol{\lambda}_t, \nabla F_t(\boldsymbol{\lambda}_t), \alpha)\|^2 \leq \sum_{t=1}^{T} F_t(\boldsymbol{\lambda}_t) - F_{t+1}(\boldsymbol{\lambda}_{t+1}).$$

Choosing $\alpha = \frac{1}{\ell_{F,1}}$ with our assumption on sublinear subsequent function evaluations, that is it holds that $\sum_{t=1}^{T} F_t(\boldsymbol{\lambda}_t) - F_{t+1}(\boldsymbol{\lambda}_{t+1}) \leq o(T)$, then we have the sublinear rate

$$\sum_{t=1}^{T}\|\mathcal{G}_{\mathcal{X}}(\boldsymbol{\lambda}_t, \nabla F_t(\boldsymbol{\lambda}_t), \alpha)\|^2 \leq \frac{\ell_{F,1}\sum_{t=1}^{T}(F_t(\boldsymbol{\lambda}_t) - F_{t+1}(\boldsymbol{\lambda}_{t+1}))}{(\rho - 1/2)} \leq o\left(\frac{\ell_{F,1}}{\rho}T\right).$$

Selecting the $\rho$-strongly convex function $\phi(\boldsymbol{\lambda})$ that specifies the Bregman Divergence $D_\phi(\boldsymbol{\lambda}, \boldsymbol{u})$ in equation 4 such that $\rho = O(\ell_{F,1}) = O(\kappa_g^3)$ implies that the bilevel local regret is sublinear with the rate of

$$\sum_{t=1}^{T}\|\mathcal{G}_{\mathcal{X}}(\boldsymbol{\lambda}_t, \nabla F_t(\boldsymbol{\lambda}_t), \alpha)\|^2 \leq o(T).$$

$\square$

Our first Lemma upper bounds the expected cumulative difference between the time-smoothed outer level objective $F_{t,w}(\boldsymbol{\lambda})$ evaluated at $\boldsymbol{\lambda}_t$ and $\boldsymbol{\lambda}_{t+1}$ in terms of the outer level objective upper bound $Q$ from Assumption E, window size $w$, and a comparator sequence on subsequent function evaluations $V_{1,T}$.

**Lemma 13.** *Suppose Assumption E. If Algorithm 3 is applied with window size $w \geq 1$ to generate the sequence $\{\boldsymbol{\lambda}_t\}_{t=1}^{T}$, then we have the upper bound in expectation of*

$$\sum_{t=1}^{T}(F_{t,w}(\boldsymbol{\lambda}_t) - F_{t,w}(\boldsymbol{\lambda}_{t+1})) \leq \frac{2TQ}{w} + V_{1,T}.$$

*Proof.* By definition in the stochastic setting, we have $F_t(\boldsymbol{\lambda}) \triangleq \mathbb{E}_\epsilon[f_t(\boldsymbol{\lambda}, \boldsymbol{\beta}_t^*(\boldsymbol{\lambda}), \epsilon)]$. Then it holds, with the linearity of expectation that

$$\sum_{t=1}^{T}(F_{t,w}(\boldsymbol{\lambda}_t) - F_{t,w}(\boldsymbol{\lambda}_{t+1})) = \sum_{t=1}^{T}\frac{1}{w}\sum_{i=0}^{w-1}(F_{t-i}(\boldsymbol{\lambda}_{t-i}) - F_{t-i}(\boldsymbol{\lambda}_{t+1-i}))$$

$$= \sum_{t=1}^{T}\frac{1}{w}\sum_{i=0}^{w-1}\left(\mathbb{E}_\epsilon\left[f_{t-i}(\boldsymbol{\lambda}_{t-i}, \boldsymbol{\beta}_{t-i}^*(\boldsymbol{\lambda}_{t-i}), \epsilon)\right] - \mathbb{E}_\epsilon\left[f_{t-i}(\boldsymbol{\lambda}_{t+1-i}, \boldsymbol{\beta}_{t-i}^*(\boldsymbol{\lambda}_{t+1-i}), \epsilon)\right]\right)$$

$$= \sum_{t=1}^{T}\frac{1}{w}\sum_{i=0}^{w-1}\mathbb{E}_\epsilon\left[f_{t-i}(\boldsymbol{\lambda}_{t-i}, \boldsymbol{\beta}_{t-i}^*(\boldsymbol{\lambda}_{t-i}), \epsilon) - f_{t-i}(\boldsymbol{\lambda}_{t+1-i}, \boldsymbol{\beta}_{t-i}^*(\boldsymbol{\lambda}_{t+1-i}), \epsilon)\right]$$

Which with the linearity of expectation is equivalent to

$$\sum_{t=1}^{T} \frac{1}{w} \sum_{i=0}^{w-1} \mathbb{E}_{\epsilon} \left[ f_{t-i} \left( \boldsymbol{\lambda}_{t-i}, \boldsymbol{\beta}_{t-i}^{*}(\boldsymbol{\lambda}_{t-i}), \epsilon \right) - f_{t-i} \left( \boldsymbol{\lambda}_{t+1-i}, \boldsymbol{\beta}_{t-i}^{*}(\boldsymbol{\lambda}_{t+1-i}), \epsilon \right) \right]$$

$$= \sum_{t=1}^{T} \frac{1}{w} \sum_{i=0}^{w-1} \mathbb{E}_{\epsilon} \left[ f_{t-i} \left( \boldsymbol{\lambda}_{t-i}, \boldsymbol{\beta}_{t-i}^{*}(\boldsymbol{\lambda}_{t-i}), \epsilon \right) - f_{t+1-i} \left( \boldsymbol{\lambda}_{t+1-i}, \boldsymbol{\beta}_{t+1-i}^{*}(\boldsymbol{\lambda}_{t+1-i}), \epsilon \right) \right] \quad (22)$$

$$+ \sum_{t=1}^{T} \frac{1}{w} \sum_{i=0}^{w-1} \mathbb{E}_{\epsilon} \left[ f_{t+1-i} \left( \boldsymbol{\lambda}_{t+1-i}, \boldsymbol{\beta}_{t+1-i}^{*}(\boldsymbol{\lambda}_{t+1-i}), \epsilon \right) - f_{t-i} \left( \boldsymbol{\lambda}_{t+1-i}, \boldsymbol{\beta}_{t-i}^{*}(\boldsymbol{\lambda}_{t+1-i}), \epsilon \right) \right] \quad (23)$$

For equation 22, with linearity of expectation, we have

$$\frac{1}{w} \sum_{i=0}^{w-1} \mathbb{E}_{\epsilon} \left[ f_{t-i} \left( \boldsymbol{\lambda}_{t-i}, \boldsymbol{\beta}_{t-i}^{*}(\boldsymbol{\lambda}_{t-i}), \epsilon \right) - f_{t+1-i} \left( \boldsymbol{\lambda}_{t+1-i}, \boldsymbol{\beta}_{t+1-i}^{*}(\boldsymbol{\lambda}_{t+1-i}), \epsilon \right) \right]$$

$$= \frac{1}{w} \mathbb{E}_{\epsilon} \left[ f_{t} \left( \boldsymbol{\lambda}_{t}, \boldsymbol{\beta}_{t}^{*}(\boldsymbol{\lambda}_{t}), \epsilon \right) + \ldots + f_{t+1-w} \left( \boldsymbol{\lambda}_{t+1-w}, \boldsymbol{\beta}_{t+1-w}^{*}(\boldsymbol{\lambda}_{t+1-w}), \epsilon \right) \right]$$

$$- \frac{1}{w} \mathbb{E}_{\epsilon} \left[ f_{t+1} \left( \boldsymbol{\lambda}_{t+1}, \boldsymbol{\beta}_{t+1}^{*}(\boldsymbol{\lambda}_{t+1}) \right) + \ldots + f_{t+2-w} \left( \boldsymbol{\lambda}_{t+2-w}, \boldsymbol{\beta}_{t+2-w}^{*}(\boldsymbol{\lambda}_{t+2-w}), \epsilon \right) \right]$$

$$= \frac{1}{w} \mathbb{E}_{\epsilon} \left[ f_{t+1-w} \left( \boldsymbol{\lambda}_{t+1-w}, \boldsymbol{\beta}_{t+1-w}^{*}(\boldsymbol{\lambda}_{t+1-w}), \epsilon \right) - f_{t+1} \left( \boldsymbol{\lambda}_{t+1}, \boldsymbol{\beta}_{t+1}^{*}(\boldsymbol{\lambda}_{t+1}), \epsilon \right) \right]$$

$$= \frac{1}{w} \left( F_{t+1-w}(\boldsymbol{\lambda}_{t+1-w}) - F_{t+1}(\boldsymbol{\lambda}_{t+1}) \right) \leq \frac{2Q}{w}, \quad (24)$$

where the last inequality comes from Assumption E. Note equation 23 can be bounded through

$$\sum_{t=1}^{T} \frac{1}{w} \sum_{i=0}^{w-1} \mathbb{E}_{\epsilon} \left[ f_{t+1-i} \left( \boldsymbol{\lambda}_{t+1-i}, \boldsymbol{\beta}_{t+1-i}^{*}(\boldsymbol{\lambda}_{t+1-i}), \epsilon \right) - f_{t-i} \left( \boldsymbol{\lambda}_{t+1-i}, \boldsymbol{\beta}_{t-i}^{*}(\boldsymbol{\lambda}_{t+1-i}), \epsilon \right) \right]$$

$$\leq \sum_{t=1}^{T} \frac{1}{w} \sum_{i=0}^{w-1} \sup_{\boldsymbol{\lambda}} \mathbb{E}_{\epsilon} \left[ f_{t+1-i} \left( \boldsymbol{\lambda}, \boldsymbol{\beta}_{t+1-i}^{*}(\boldsymbol{\lambda}), \epsilon \right) - f_{t-i} \left( \boldsymbol{\lambda}, \boldsymbol{\beta}_{t-i}^{*}(\boldsymbol{\lambda}), \epsilon \right) \right]$$

$$= \sum_{t=1}^{T} \sup_{\boldsymbol{\lambda} \in \mathcal{X}} \left[ F_{t+1} \left( \boldsymbol{\lambda} \right) - F_{t} \left( \boldsymbol{\lambda} \right) \right] := V_{1,T} \quad (25)$$

Combining equation 23 and equation 25 results in the upper bound of

$$\sum_{t=1}^{T} \left( F_{t,w}(\boldsymbol{\lambda}_{t}) - F_{t,w}(\boldsymbol{\lambda}_{t+1}) \right) \leq \frac{2TQ}{w} + V_{1,T}.$$

$\square$

The next Lemma provides an upper bound on the expected error of $\mathbb{E}_{\bar{\zeta}_{t,K+1}} \left[ \|\boldsymbol{\beta}_{t} - \boldsymbol{\beta}_{t}^{*}(\boldsymbol{\lambda}_{t})\|^{2} \right]$ for all $t \in [1, T]$ in terms of an expected initial error, the expected cumulative differences of the outer level variable, the expected cumulative differences of the optimal inner level variables, and a variance term arising from the stochasticity of $g_{t}(\boldsymbol{\lambda}, \boldsymbol{\beta}, \zeta)$.

**Lemma 14.** *Suppose Assumptions A, B, and C. Choose the inner step size of $\eta = \Omega(1/\mu_{g})$ with the inner iteration count $K$ as*

$$0 < \eta \leq \frac{2}{\ell_{g,1} + \mu_{g}}, \quad \text{and} \quad K \geq 1,$$

*and define the decay parameter $\nu$, the inner level variable error constant $C_{\mu_{g}}$, the initial error $\Delta_{\boldsymbol{\beta}}$, and the inner level variable error variance $C_{K}$ respectively as*

$$\nu := \left( 1 - \frac{\eta \ell_{g,1} \mu_{g}}{\ell_{g,1} + \mu_{g}} \right) \left( 1 - \frac{2\eta \ell_{g,1} \mu_{g}}{\ell_{g,1} + \mu_{g}} \right)^{K-1}, \quad C_{\mu_{g}} := \left( 1 + \frac{\ell_{g,1} + \mu_{g}}{\eta \ell_{g,1} \mu_{g}} \right),$$

$$\Delta_{\boldsymbol{\beta}} := \|\boldsymbol{\beta}_{2} - \boldsymbol{\beta}_{1}^{*}(\boldsymbol{\lambda}_{1})\|^{2} = O(1), \quad \text{and} \quad C_{K} := \sum_{k=1}^{K} \left( 1 - \frac{2\eta \ell_{g,1} \mu_{g}}{\ell_{g,1} + \mu_{g}} \right)^{k}. \quad (26)$$

*Then we have $C_{\mu_g} = O(1)$, and $\forall t \in [1, T]$, t*

$$\mathbb{E}_{\bar{\zeta}_{t,K+1}} \left[ \|\boldsymbol{\beta}_{t+1} - \boldsymbol{\beta}_t^*(\boldsymbol{\lambda}_t)\|^2 \right] \leq \nu^{t-1} \Delta_{\boldsymbol{\beta}} + 2C_{\mu_g} \kappa_g^2 \sum_{j=0}^{t-2} \nu^{j+1} \left[ \|\boldsymbol{\lambda}_{t-1-j} - \boldsymbol{\lambda}_{t-j}\|^2 \right]$$

$$+ 2C_{\mu_g} \sum_{j=0}^{t-2} \nu^{j+1} \left[ \left\| \boldsymbol{\beta}_{t-j}^*(\boldsymbol{\lambda}_{t-1-j}) - \boldsymbol{\beta}_{t-1-j}^*(\boldsymbol{\lambda}_{t-1-j}) \right\|^2 \right] + \frac{C_K \eta^2 \sigma_g^2}{s} \sum_{j=0}^{t-2} \nu^j.$$

*Proof.* Note $\forall k \in [1, K]$ the following expansion holds

$$\left\| \boldsymbol{\omega}_t^k - \boldsymbol{\beta}_t^*(\boldsymbol{\lambda}_t) \right\|^2$$

$$= \left\| \boldsymbol{\omega}_t^k - \boldsymbol{\omega}_t^{k-1} \right\|^2 + 2 \left\langle \boldsymbol{\omega}_t^k - \boldsymbol{\omega}_t^{k-1}, \boldsymbol{\omega}_t^{k-1} - \boldsymbol{\beta}_t^*(\boldsymbol{\lambda}_t) \right\rangle + \left\| \boldsymbol{\omega}_t^{k-1} - \boldsymbol{\beta}_t^*(\boldsymbol{\lambda}_t) \right\|^2$$

$$= \eta^2 \left\| \nabla_{\boldsymbol{\omega}} g_t(\boldsymbol{\lambda}_t, \boldsymbol{\omega}_t^{k-1}, \bar{\zeta}_{t,k}) \right\|^2 - 2\eta \left\langle \nabla_{\boldsymbol{\omega}} g_t(\boldsymbol{\lambda}_t, \boldsymbol{\omega}_t^{k-1}, \bar{\zeta}_{t,k}), \boldsymbol{\omega}_t^{k-1} - \boldsymbol{\beta}_t^*(\boldsymbol{\lambda}_t) \right\rangle$$

$$+ \left\| \boldsymbol{\omega}_t^{k-1} - \boldsymbol{\beta}_t^*(\boldsymbol{\lambda}_t) \right\|^2.$$

Using the definition of variance of

$$VAR_{\bar{\zeta}_{t,k}} \left[ \left\| \nabla_{\boldsymbol{\omega}} g_t(\boldsymbol{\lambda}_t, \boldsymbol{\omega}_t^{k-1}, \bar{\zeta}_{t,k}) \right\| \right]$$

$$= \mathbb{E}_{\bar{\zeta}_{t,k}} \left[ \left\| \nabla_{\boldsymbol{\omega}} g_t(\boldsymbol{\lambda}_t, \boldsymbol{\omega}_t^{k-1}, \bar{\zeta}_{t,k}) \right\|^2 \right] - \mathbb{E}_{\bar{\zeta}_{t,k}} \left[ \left\| \nabla_{\boldsymbol{\omega}} g_t(\boldsymbol{\lambda}_t, \boldsymbol{\omega}_t^{k-1}, \bar{\zeta}_{t,k}) \right\| \right]^2,$$

and conditioning on $\boldsymbol{\omega}_t^{k-1}$, we take expectation (Assumption C) to provide the upper bound of

$$\mathbb{E}_{\bar{\zeta}_{t,k}} \left[ \left\| \boldsymbol{\omega}_t^k - \boldsymbol{\beta}_t^*(\boldsymbol{\lambda}_t) \right\|^2 \right] \leq \eta^2 \left( \frac{\sigma_g^2}{s} + \left\| \nabla_{\boldsymbol{\omega}} g_t(\boldsymbol{\lambda}_t, \boldsymbol{\omega}_t^{k-1}) \right\|^2 \right)$$

$$- 2\eta \left\langle \nabla_{\boldsymbol{\omega}} g_t(\boldsymbol{\lambda}_t, \boldsymbol{\omega}_t^{k-1}), \boldsymbol{\omega}_t^{k-1} - \boldsymbol{\beta}_t^*(\boldsymbol{\lambda}_t) \right\rangle + \left\| \boldsymbol{\omega}_t^{k-1} - \boldsymbol{\beta}_t^*(\boldsymbol{\lambda}_t) \right\|^2. \tag{27}$$

The above upper bound is deterministic, and as such we can utilize the $\mu_g$-strong convexity of $g_t$ to bound

$$-2\eta \left\langle \nabla_{\boldsymbol{\omega}} g_t(\boldsymbol{\lambda}_t, \boldsymbol{\omega}_t^{k-1}), \boldsymbol{\omega}_t^{k-1} - \boldsymbol{\beta}_t^*(\boldsymbol{\lambda}_t) \right\rangle$$

$$\leq -2\eta \left( \frac{\ell_{g,1}\mu_g}{\ell_{g,1} + \mu_g} \left\| \boldsymbol{\omega}_t^{k-1} - \boldsymbol{\beta}_t^*(\boldsymbol{\lambda}_t) \right\|^2 + \frac{1}{\ell_{g,1} + \mu_g} \left\| \nabla_{\boldsymbol{\omega}} g_t(\boldsymbol{\lambda}_t, \boldsymbol{\omega}_t^{k-1}) \right\|^2 \right),$$

which we can substitute in equation 27 to get

$$\mathbb{E}_{\bar{\zeta}_{t,k}} \left[ \left\| \boldsymbol{\omega}_t^k - \boldsymbol{\beta}_t^*(\boldsymbol{\lambda}_t) \right\|^2 \right] \leq \frac{\eta^2 \sigma_g^2}{s} - \eta \left( \frac{2}{\ell_{g,1} + \mu_g} - \eta \right) \left\| \nabla_{\boldsymbol{\omega}} g_t(\boldsymbol{\lambda}_t, \boldsymbol{\omega}_t^{k-1}) \right\|^2$$

$$+ \left( 1 - \frac{2\eta \ell_{g,1}\mu_g}{\ell_{g,1} + \mu_g} \right) \left\| \boldsymbol{\omega}_t^{k-1} - \boldsymbol{\beta}_t^*(\boldsymbol{\lambda}_t) \right\|^2.$$

As $\eta \leq \frac{2}{\ell_{g,1} + \mu_g}$ this provides the upper bound to equation 27 of

$$\mathbb{E}_{\bar{\zeta}_{t,k}} \left[ \left\| \boldsymbol{\omega}_t^k - \boldsymbol{\beta}_t^*(\boldsymbol{\lambda}_t) \right\|^2 \right] \leq \left( 1 - \frac{2\eta \ell_{g,1}\mu_g}{\ell_{g,1} + \mu_g} \right) \left\| \boldsymbol{\omega}_t^{k-1} - \boldsymbol{\beta}_t^*(\boldsymbol{\lambda}_t) \right\|^2 + \frac{\eta^2 \sigma_g^2}{s}.$$

This can be unrolled, through iterative conditioning, from $k = K, \dots, 1$

$$\mathbb{E}_{\bar{\zeta}_{t,K+1}} \left[ \left\| \boldsymbol{\omega}_t^K - \boldsymbol{\beta}_t^*(\boldsymbol{\lambda}_t) \right\|^2 \right] \leq \left( 1 - \frac{2\eta \ell_{g,1}\mu_g}{\ell_{g,1} + \mu_g} \right)^K \mathbb{E}_{\bar{\zeta}_{t,1}} \left\| \boldsymbol{\omega}_t^0 - \boldsymbol{\beta}_t^*(\boldsymbol{\lambda}_t) \right\|^2 + \frac{C_K \eta^2 \sigma_g^2}{s},$$

for $C_K := \sum_{k=1}^{K} \left( 1 - \frac{2\eta \ell_{g,1}\mu_g}{\ell_{g,1} + \mu_g} \right)^k$. By definition of $\boldsymbol{\beta}_{t+1} = \boldsymbol{\omega}_t^K$ and $\boldsymbol{\omega}_t^0 = \boldsymbol{\beta}_t$ gives us

$$\mathbb{E}_{\bar{\zeta}_{t,K+1}} \left[ \left\| \boldsymbol{\beta}_{t+1} - \boldsymbol{\beta}_t^*(\boldsymbol{\lambda}_t) \right\|^2 \right] \leq \left( 1 - \frac{2\eta \ell_{g,1}\mu_g}{\ell_{g,1} + \mu_g} \right)^K \mathbb{E}_{\bar{\zeta}_{t-1,K+1}} \left\| \boldsymbol{\beta}_t - \boldsymbol{\beta}_t^*(\boldsymbol{\lambda}_t) \right\|^2 + \frac{C_K \eta^2 \sigma_g^2}{s}.$$

Note we can decompose

$$\mathbb{E}_{\bar{\zeta}_{t-1,K+1}} \left\| \boldsymbol{\beta}_t - \boldsymbol{\beta}_t^*(\boldsymbol{\lambda}_t) \right\|^2 = \mathbb{E}_{\bar{\zeta}_{t-1,K+1}} \left\| \boldsymbol{\beta}_t - \boldsymbol{\beta}_{t-1}^*(\boldsymbol{\lambda}_{t-1}) + \boldsymbol{\beta}_{t-1}^*(\boldsymbol{\lambda}_{t-1}) - \boldsymbol{\beta}_t^*(\boldsymbol{\lambda}_t) \right\|^2,$$

which can be expanded based on Young's Inequality and the linearity of expectation for any $\delta > 0$ as

$$\mathbb{E}_{\bar{\zeta}_{t-1,K+1}} \left\| \boldsymbol{\beta}_t - \boldsymbol{\beta}_{t-1}^*(\boldsymbol{\lambda}_{t-1}) + \boldsymbol{\beta}_{t-1}^*(\boldsymbol{\lambda}_{t-1}) - \boldsymbol{\beta}_t^*(\boldsymbol{\lambda}_t) \right\|^2$$

$$\leq (1+\delta)\mathbb{E}_{\bar{\zeta}_{t-1,K+1}} \left\| \boldsymbol{\beta}_t - \boldsymbol{\beta}_{t-1}^*(\boldsymbol{\lambda}_{t-1}) \right\|^2$$

$$+ \left( 1 + \frac{1}{\delta} \right) \mathbb{E}_{\bar{\zeta}_{t-1,K+1}} \left\| \boldsymbol{\beta}_{t-1}^*(\boldsymbol{\lambda}_{t-1}) - \boldsymbol{\beta}_t^*(\boldsymbol{\lambda}_t) \right\|^2. \tag{28}$$

Now it holds through linearity of expectation that

$$\mathbb{E}_{\bar{\zeta}_{t-1,K+1}} \left\| \boldsymbol{\beta}_{t-1}^*(\boldsymbol{\lambda}_{t-1}) - \boldsymbol{\beta}_t^*(\boldsymbol{\lambda}_t) \right\|^2 \leq 2\mathbb{E}_{\bar{\zeta}_{t-1,K+1}} \left\| \boldsymbol{\beta}_t^*(\boldsymbol{\lambda}_{t-1}) - \boldsymbol{\beta}_t^*(\boldsymbol{\lambda}_t) \right\|^2$$

$$+ 2\mathbb{E}_{\bar{\zeta}_{t-1,K+1}} \left\| \boldsymbol{\beta}_t^*(\boldsymbol{\lambda}_{t-1}) - \boldsymbol{\beta}_{t-1}^*(\boldsymbol{\lambda}_{t-1}) \right\|^2 \tag{29}$$

which through Lemma 11 can be further upper bounded with the Lipschitz constant of $\kappa_g$ as

$$\mathbb{E}_{\bar{\zeta}_{t-1,K+1}} \left\| \boldsymbol{\beta}_{t-1}^*(\boldsymbol{\lambda}_{t-1}) - \boldsymbol{\beta}_t^*(\boldsymbol{\lambda}_t) \right\|^2$$

$$\leq 2\kappa_g^2 \mathbb{E}_{\bar{\zeta}_{t-1,K+1}} \left\| \boldsymbol{\lambda}_{t-1} - \boldsymbol{\lambda}_t \right\|^2 + 2\mathbb{E}_{\bar{\zeta}_{t-1,K+1}} \left\| \boldsymbol{\beta}_t^*(\boldsymbol{\lambda}_{t-1}) - \boldsymbol{\beta}_{t-1}^*(\boldsymbol{\lambda}_{t-1}) \right\|^2$$

$$= 2\kappa_g^2 \left\| \boldsymbol{\lambda}_{t-1} - \boldsymbol{\lambda}_t \right\|^2 + 2 \left\| \boldsymbol{\beta}_t^*(\boldsymbol{\lambda}_{t-1}) - \boldsymbol{\beta}_{t-1}^*(\boldsymbol{\lambda}_{t-1}) \right\|^2$$

$$\tag{30}$$

where the last line comes from the non-randomness of $\left\| \boldsymbol{\lambda}_{t-1} - \boldsymbol{\lambda}_t \right\|^2$ and $\left\| \boldsymbol{\beta}_t^*(\boldsymbol{\lambda}_{t-1}) - \boldsymbol{\beta}_{t-1}^*(\boldsymbol{\lambda}_{t-1}) \right\|^2$ with respect to $\bar{\zeta}_{t,k}$. Combining equation 29 and equation 28, we have $\forall \delta > 0$

$$\mathbb{E}_{\bar{\zeta}_{t,K+1}} \left[ \left\| \boldsymbol{\beta}_{t+1} - \boldsymbol{\beta}_t^*(\boldsymbol{\lambda}_t) \right\|^2 \right] \leq \left( 1 - \frac{2\eta\ell_{g,1}\mu_g}{\ell_{g,1}+\mu_g} \right)^K (1+\delta)\mathbb{E}_{\bar{\zeta}_{t-1,K+1}} \left[ \left\| \boldsymbol{\beta}_t - \boldsymbol{\beta}_{t-1}^*(\boldsymbol{\lambda}_{t-1}) \right\|^2 \right]$$

$$+ 2 \left( 1 - \frac{2\eta\ell_{g,1}\mu_g}{\ell_{g,1}+\mu_g} \right)^K \left( 1 + \frac{1}{\delta} \right) \kappa_g^2 \left\| \boldsymbol{\lambda}_{t-1} - \boldsymbol{\lambda}_t \right\|^2$$

$$+ 2 \left( 1 - \frac{2\eta\ell_{g,1}\mu_g}{\ell_{g,1}+\mu_g} \right)^K \left( 1 + \frac{1}{\delta} \right) \left\| \boldsymbol{\beta}_t^*(\boldsymbol{\lambda}_{t-1}) - \boldsymbol{\beta}_{t-1}^*(\boldsymbol{\lambda}_{t-1}) \right\|^2 + \frac{C_K \eta^2 \sigma_g^2}{s}.$$

Now setting $\delta = \frac{\eta\ell_{g,1}\mu_g}{\ell_{g,1}+\mu_g} > 0$ implies the upper bound of

$$(1+\delta) \left( 1 - \frac{2\eta\ell_{g,1}\mu_g}{\ell_{g,1}+\mu_g} \right)^K < \left( 1 - \frac{\eta\ell_{g,1}\mu_g}{\ell_{g,1}+\mu_g} \right) \left( 1 - \frac{2\eta\ell_{g,1}\mu_g}{\ell_{g,1}+\mu_g} \right)^{K-1} < 1,$$

which defining $\nu := \left( 1 - \frac{\eta\mu_g\ell_{g,1}}{\ell_{g,1}+\mu_g} \right) \left( 1 - \frac{2\eta\ell_{g,1}\mu_g}{\ell_{g,1}+\mu_g} \right)^{K-1}$ and $\delta > 0$ implies

$$\left( 1 - \frac{2\eta\ell_{g,1}\mu_g}{\ell_{g,1}+\mu_g} \right)^K < \nu,$$

Using the definition of $\nu$, we get

$$\nu\mathbb{E}_{\bar{\zeta}_{t,K+1}} \left[ \left\| \boldsymbol{\beta}_{t+1} - \boldsymbol{\beta}_t^*(\boldsymbol{\lambda}_t) \right\|^2 \right] \leq \nu^2 \mathbb{E}_{\bar{\zeta}_{t-1,K+1}} \left[ \left\| \boldsymbol{\beta}_t - \boldsymbol{\beta}_{t-1}^*(\boldsymbol{\lambda}_{t-1}) \right\|^2 \right]$$

$$+ 2C_{\mu_g}\nu^2\kappa_g^2 \left\| \boldsymbol{\lambda}_{t-1} - \boldsymbol{\lambda}_t \right\|^2 + 2C_{\mu_g}\nu^2 \left\| \boldsymbol{\beta}_t^*(\boldsymbol{\lambda}_{t-1}) - \boldsymbol{\beta}_{t-1}^*(\boldsymbol{\lambda}_{t-1}) \right\|^2 + \frac{\nu C_K \eta^2 \sigma_g^2}{s},$$

where $C_{\mu_g} = \left( 1 + \frac{\ell_{g,1}+\mu_g}{\eta\ell_{g,1}\mu_g} \right)$. Starting at $t = T$, and unrolling to $t = 1$, we can write

$$\mathbb{E}_{\bar{\zeta}_{t,K+1}} \left[ \left\| \boldsymbol{\beta}_{t+1} - \boldsymbol{\beta}_t^*(\boldsymbol{\lambda}_t) \right\|^2 \right] \leq \nu^{t-1}\Delta_{\boldsymbol{\beta}} + 2C_{\mu_g}\kappa_g^2 \sum_{j=0}^{t-2} \nu^{j+1} \left[ \left\| \boldsymbol{\lambda}_{t-1-j} - \boldsymbol{\lambda}_{t-j} \right\|^2 \right]$$

$$+ 2C_{\mu_g} \sum_{j=0}^{t-2} \nu^{j+1} \left[ \left\| \boldsymbol{\beta}_{t-j}^*(\boldsymbol{\lambda}_{t-1-j}) - \boldsymbol{\beta}_{t-1-j}^*(\boldsymbol{\lambda}_{t-1-j}) \right\|^2 \right] + \frac{C_K \eta^2 \sigma_g^2}{s} \sum_{j=0}^{t-2} \nu^j.$$

$$\square$$

The next Lemma utilizes Lemma 11 and Lemma 14 to derive an upper bound on the expected hypergradient error $\forall t \in [1, T]$ with respect to $\bar{\zeta}_{t,k}$ in terms of discounted variations of the (i) cumulative time-smoothed hypergradient error; (ii) bilevel local regret; and (iii) cumulative difference between optimal inner-level variables. There is a term composed of a discounted initial error and smoothness term of the inner objective, as well as an additional term arising from the variance of the stochastic gradients of $g_t(\boldsymbol{\lambda}, \boldsymbol{\beta}, \zeta)$.

**Lemma 15.** *Suppose Assumptions A, B, C, D, and F. Choose the inner step size of $\eta = \Omega(1/\mu_g)$ and inner iteration count $K$ as*

$$0 < \eta \leq \frac{2}{\ell_{g,1} + \mu_g}, \quad and \quad K \geq 1.$$

*With the definitions of $\nu$, $C_{\mu_g}$, $\Delta_{\boldsymbol{\beta}}$, and $C_K$ from Lemma 14, the expected hypergradient error can be bounded as*

$$\mathbb{E}_{\bar{\zeta}_{t,K+1}} \left[ \left\| \widetilde{\nabla} f_t(\boldsymbol{\lambda}_t, \boldsymbol{\beta}_{t+1}) - \nabla F_t(\boldsymbol{\lambda}_t) \right\|^2 \right] \leq \delta_t + A \sum_{j=0}^{t-2} \nu^{j+1} \left\| \mathcal{G}_{\mathcal{X}}(\boldsymbol{\lambda}_{t-1-j}, \nabla F_{t-1-j,w}(\boldsymbol{\lambda}_{t-1-j}), \alpha) \right\|^2$$

$$+ B \sum_{j=0}^{t-2} \nu^{j+1} \left\| \widetilde{\nabla} f_{t-1-j,w}(\boldsymbol{\lambda}_{t-1-j}, \boldsymbol{\beta}_{t-j}, \mathcal{Z}_{t-1-j,w}) - \nabla F_{t-1-j,w}(\boldsymbol{\lambda}_{t-1-j}) \right\|^2$$

$$+ C \sum_{j=0}^{t-2} \nu^{j+1} \left[ \left\| \boldsymbol{\beta}_{t-j}^*(\boldsymbol{\lambda}_{t-1-j}) - \boldsymbol{\beta}_{t-1-j}^*(\boldsymbol{\lambda}_{t-1-j}) \right\|^2 \right] + \frac{D\sigma_g^2}{s}.$$

*where $\delta_t = \kappa_g^2 \nu^{t-1} \Delta_{\boldsymbol{\beta}}$ and $A = 4 C_{\mu_g} \kappa_g^4 \alpha^2, B = \frac{4 C_{\mu_g} \kappa_g^4 \alpha^2}{\rho^2}, C = 2 C_{\mu_g} \kappa_g^2$, and $D = C_K \kappa_g^2 \eta^2 \sum_{j=0}^{t-2} \nu^j$.*

*Proof.* First, from Lemma 11 (Assumption A,B) we have that $\forall \boldsymbol{\lambda} \in \mathcal{X}$ and $\boldsymbol{\beta} \in \mathbb{R}^{d_2}$

$$\left\| \widetilde{\nabla} f_t(\boldsymbol{\lambda}_t, \boldsymbol{\beta}_{t+1}) - \nabla F_t(\boldsymbol{\lambda}_t) \right\|^2 \leq \kappa_g^2 \left\| \boldsymbol{\beta}_{t+1} - \boldsymbol{\beta}_t^*(\boldsymbol{\lambda}_t) \right\|^2. \tag{31}$$

Taking expectation of equation 31 with respect to $\bar{\zeta}_{t,K+1}$ (Assumption C) and substituting the upper bound of Lemma 14, note

$$\mathbb{E}_{\bar{\zeta}_{t,K+1}} \left[ \left\| \widetilde{\nabla} f_t(\boldsymbol{\lambda}_t, \boldsymbol{\beta}_{t+1}) - \nabla F_t(\boldsymbol{\lambda}_t) \right\|^2 \right]$$

$$\leq \kappa_g^2 \left( \nu^{t-1} \Delta_{\boldsymbol{\beta}} + 2 C_{\mu_g} \kappa_g^2 \sum_{j=0}^{t-2} \nu^{j+1} \left\| \boldsymbol{\lambda}_{t-1-j} - \boldsymbol{\lambda}_{t-j} \right\|^2 \right) \tag{32}$$

$$+ \kappa_g^2 \left( 2 C_{\mu_g} \sum_{j=0}^{t-2} \nu^{j+1} \left\| \boldsymbol{\beta}_{t-j}^*(\boldsymbol{\lambda}_{t-1-j}) - \boldsymbol{\beta}_{t-1-j}^*(\boldsymbol{\lambda}_{t-1-j}) \right\|^2 + \frac{C_K \eta^2 \sigma_g^2}{s} \sum_{j=0}^{t-2} \nu^j \right). \tag{33}$$

Focusing on the second term of equation 32 we see by definition

$$\sum_{j=0}^{t-2} \nu^{j+1} \left\| \boldsymbol{\lambda}_{t-1-j} - \boldsymbol{\lambda}_{t-j} \right\|^2$$

$$= \sum_{j=0}^{t-2} \nu^{j+1} \alpha^2 \left\| \mathcal{G}_{\mathcal{X}}(\boldsymbol{\lambda}_{t-1-j}, \widetilde{\nabla} f_{t-1-j,\boldsymbol{w}}(\boldsymbol{\lambda}_{t-1-j}, \boldsymbol{\beta}_{t-j}, \mathcal{Z}_{t-1-j,w}), \alpha) \right\|^2. \tag{34}$$

Using Lemma 10 (Assumption D,F) we have $\forall j \in [0, t-2]$

$$\left\| \mathcal{G}_{\mathcal{X}}(\boldsymbol{\lambda}_{t-1-j}, \widetilde{\nabla} f_{t-1-j,\boldsymbol{w}}(\boldsymbol{\lambda}_{t-1-j}, \boldsymbol{\beta}_{t-j}, \mathcal{Z}_{t-1-j,w}), \alpha) \right\|^2 \leq 2 \left\| \mathcal{G}_{\mathcal{X}}(\boldsymbol{\lambda}_{t-1-j}, \nabla F_{t-1-j,w}(\boldsymbol{\lambda}_{t-1-j}), \alpha) \right\|^2$$

$$+ 2 \left\| \mathcal{G}_{\mathcal{X}}(\boldsymbol{\lambda}_{t-1-j}, \nabla F_{t-1-j,w}(\boldsymbol{\lambda}_{t-1-j}), \alpha) - \mathcal{G}_{\mathcal{X}}(\boldsymbol{\lambda}_{t-1-j}, \widetilde{\nabla} f_{t-1-j,\boldsymbol{w}}(\boldsymbol{\lambda}_{t-1-j}, \boldsymbol{\beta}_{t-j}, \mathcal{Z}_{t-1-j,w}), \alpha) \right\|^2$$

$$\leq 2 \left\| \mathcal{G}_{\mathcal{X}}(\boldsymbol{\lambda}_{t-1-j}, \nabla F_{t-1-j,w}(\boldsymbol{\lambda}_{t-1-j}), \alpha) \right\|^2$$

$$+ \frac{2}{\rho^2} \left\| \widetilde{\nabla} f_{t-1-j,w}(\boldsymbol{\lambda}_{t-1-j}, \boldsymbol{\beta}_{t-j}, \mathcal{Z}_{t-1-j,w}) - \nabla F_{t-1-j,w}(\boldsymbol{\lambda}_{t-1-j}) \right\|^2.$$

We can write an upper bound to equation 34 as

$$\sum_{j=0}^{t-2} \nu^{j+1} \left\| \boldsymbol{\lambda}_{t-1-j} - \boldsymbol{\lambda}_{t-j} \right\|^2 \leq 2\alpha^2 \sum_{j=0}^{t-2} \nu^{j+1} \left\| \mathcal{G}_{\mathcal{X}}(\boldsymbol{\lambda}_{t-1-j}, \nabla F_{t-1-j,w}(\boldsymbol{\lambda}_{t-1-j}), \alpha) \right\|^2$$

$$+ \frac{2\alpha^2}{\rho^2} \sum_{j=0}^{t-2} \nu^{j+1} \left( \left\| \widetilde{\nabla} f_{t-1-j,w}(\boldsymbol{\lambda}_{t-1-j}, \boldsymbol{\beta}_{t-j}, \mathcal{Z}_{t-1-j,w}) - \nabla F_{t-1-j,w}(\boldsymbol{\lambda}_{t-1-j}) \right\|^2 \right). \quad (35)$$

Using equation 35, we get

$$\mathbb{E}_{\bar{\zeta}_{t,K+1}} \left[ \left\| \widetilde{\nabla} f_t(\boldsymbol{\lambda}_t, \boldsymbol{\beta}_{t+1}) - \nabla F_t(\boldsymbol{\lambda}_t) \right\|^2 \right] \leq \delta_t + A \sum_{j=0}^{t-2} \nu^{j+1} \left\| \mathcal{G}_{\mathcal{X}}(\boldsymbol{\lambda}_{t-1-j}, \nabla F_{t-1-j,w}(\boldsymbol{\lambda}_{t-1-j}), \alpha) \right\|^2$$

$$+ B \sum_{j=0}^{t-2} \nu^{j+1} \left\| \widetilde{\nabla} f_{t-1-j,w}(\boldsymbol{\lambda}_{t-1-j}, \boldsymbol{\beta}_{t-j}, \mathcal{Z}_{t-1-j,w}) - \nabla F_{t-1-j,w}(\boldsymbol{\lambda}_{t-1-j}) \right\|^2$$

$$+ C \sum_{j=0}^{t-2} \nu^{j+1} \left[ \left\| \boldsymbol{\beta}_{t-j}^*(\boldsymbol{\lambda}_{t-1-j}) - \boldsymbol{\beta}_{t-1-j}^*(\boldsymbol{\lambda}_{t-1-j}) \right\|^2 \right] + \frac{D\sigma_g^2}{s}.$$

where $\delta_t = \kappa_g^2 \nu^{t-1} \Delta_{\boldsymbol{\beta}}$ and $A = 4C_{\mu_g} \kappa_g^4 \alpha^2, B = \frac{4C_{\mu_g} \kappa_g^4 \alpha^2}{\rho^2}$, $C = 2C_{\mu_g} \kappa_g^2$, and $D = C_K \kappa_g^2 \eta^2 \sum_{j=0}^{t-2} \nu^j$.

$\square$

Lemma 16 provides an upper bound on the expected cumulative time-smoothed hypergradient error in terms of an initial error, expected bilevel local regret, expected cumulative differences of optimal inner level variables, as well as variance terms from the stochastic approximated gradients.

**Lemma 16.** *Suppose Assumptions A, B, C, D, and F. Choose the inner step size of $\eta = \Omega(1/\mu_g)$, the inner iteration count $K$, and the outer step size $\alpha$ respectively as*

$$0 < \eta \leq \frac{2}{\ell_{g,1} + \mu_g}, \quad K \geq 1, \ and \quad \alpha < \frac{\rho\sqrt{(1-\nu)}}{\kappa_g^2 \sqrt{72C_{\mu_g}}}.$$

*Then $\forall t \in [1, T]$ the expected cumulative time-smoothed hypergradient error with respect to independent samples $Z_{t,w}$ satisfies*

$$\mathbb{E}_{Z_{t,w}} \left[ \sum_{t=1}^{T} \left\| \widetilde{\nabla} f_{t,w}(\boldsymbol{\lambda}_t, \boldsymbol{\beta}_{t+1}, \mathcal{Z}_{t,w}) - \nabla F_{t,w}(\boldsymbol{\lambda}_t) \right\|^2 \right] \leq \frac{AT}{w} + BT \left( 1 - \frac{\mu_g}{\ell_{g,1}} \right)^{2m}$$

$$+ C \sum_{t=1}^{T} \mathbb{E} \left\| \mathcal{G}_{\mathcal{X}}(\boldsymbol{\lambda}_t, \nabla F_{t,w}(\boldsymbol{\lambda}_t), \alpha) \right\|^2 + D \sum_{t=2}^{T} \mathbb{E} \left[ \left\| \boldsymbol{\beta}_t^*(\boldsymbol{\lambda}_{t-1}) - \boldsymbol{\beta}_{t-1}^*(\boldsymbol{\lambda}_{t-1}) \right\|^2 \right] + \frac{TE\eta^2 \sigma_g^2}{s}.$$

*where the initial term $\delta_t := \frac{9\kappa_g^2 \Delta_{\boldsymbol{\beta}}}{4(1-\nu)}$ and constants are defined as $A := \frac{9\sigma_f^2}{2}$, $B := \frac{9\ell_{f,1}^2 \kappa_g^2}{2}$, $C := \frac{\rho^2}{8}$, $D := \frac{9C_{\mu_g} \kappa_g^2}{2(1-\nu)}$, and $E := \frac{9C_K \kappa_g^2}{4(1-\nu)}$. Note $\Delta_{\boldsymbol{\beta}}, \nu, C_{\mu_g}, C_K$ are defined in Lemma 14, $\sigma_f^2, \sigma_g^2$ are from Assumption C, $\kappa_g$ denotes the condition number and $\rho$ specifies Bregman Divergence in equation 4.*

*Proof.* With the linearity of expectation we have

$$\mathbb{E}_{Z_{t,w}} \left[ \sum_{t=1}^{T} \left\| \widetilde{\nabla} f_{t,w}(\boldsymbol{\lambda}_t, \boldsymbol{\beta}_{t+1}, \mathcal{Z}_{t,w}) - \nabla F_{t,w}(\boldsymbol{\lambda}_t) \right\|^2 \right]$$

$$= \sum_{t=1}^{T} \mathbb{E}_{Z_{t,w}} \left[ \left\| \widetilde{\nabla} f_{t,w}(\boldsymbol{\lambda}_t, \boldsymbol{\beta}_{t+1}, \mathcal{Z}_{t,w}) - \nabla F_{t,w}(\boldsymbol{\lambda}_t) \right\|^2 \right]$$

$$= \frac{1}{w^2} \sum_{t=1}^{T} \mathbb{E}_{\mathcal{Z}_{t,w}} \left[ \left\| \sum_{i=0}^{w-1} \left[ \widetilde{\nabla} f_{t-i}(\boldsymbol{\lambda}_{t-i}, \boldsymbol{\beta}_{t+1-i}, \mathcal{E}_{t-i}) - \nabla F_{t-i}(\boldsymbol{\lambda}_{t-i}) \right] \right\|^2 \right]. \quad (36)$$

Note that we can upper bound equation 36 as

$$\frac{1}{w^2} \sum_{t=1}^{T} \mathbb{E}_{\mathcal{Z}_{t,w}} \left[ \left\| \sum_{i=0}^{w-1} \left[ \widetilde{\nabla} f_{t-i}(\boldsymbol{\lambda}_{t-i}, \boldsymbol{\beta}_{t+1-i}, \mathcal{E}_{t-i}) - \nabla F_{t-i}(\boldsymbol{\lambda}_{t-i}) \right] \right\|^2 \right]$$

$$\leq \frac{2}{w^2} \sum_{t=1}^{T} \mathbb{E}_{\mathcal{Z}_{t,w}} \left[ \left\| \sum_{i=0}^{w-1} \left[ \widetilde{\nabla} f_{t-i}(\boldsymbol{\lambda}_{t-i}, \boldsymbol{\beta}_{t+1-i}, \mathcal{E}_{t-i}) - \mathbb{E}_{\mathcal{E}_{t-i}} \left[ \widetilde{\nabla} f_{t-i}(\boldsymbol{\lambda}_{t-i}, \boldsymbol{\beta}_{t+1-i}, \mathcal{E}_{t-i}) \right] \right] \right\|^2 \right]$$

$$\tag{37}$$

$$+ \frac{2}{w^2} \sum_{t=1}^{T} \mathbb{E}_{\mathcal{Z}_{t,w}} \left[ \left\| \sum_{i=0}^{w-1} \left[ \mathbb{E}_{\mathcal{E}_{t-i}} \left[ \widetilde{\nabla} f_{t-i}(\boldsymbol{\lambda}_{t-i}, \boldsymbol{\beta}_{t+1-i}, \mathcal{E}_{t-i}) \right] - \nabla F_{t-i}(\boldsymbol{\lambda}_{t-i}) \right] \right\|^2 \right].$$

$$\tag{38}$$

The linearity of expectation, definition of variance, and independence of $Z_{t,w} := \{\mathcal{E}_{t-i}\}_{i=0}^{w-1}$ $\forall t \in [1, T]$ implies for $y_i = \widetilde{\nabla} f_{t-i}(\boldsymbol{\lambda}_{t-i}, \boldsymbol{\beta}_{t+1-i}, \mathcal{E}_{t-i})$ with finite variance $\sigma_f^2$, we have

$$\mathbb{E}_{\mathcal{Z}_{t,w}} \left[ \left\| \sum_{i=0}^{w-1} \widetilde{\nabla} f_{t-i}(\boldsymbol{\lambda}_{t-i}, \boldsymbol{\beta}_{t+1-i}, \mathcal{E}_{t-i}) - \mathbb{E}_{\mathcal{E}_{t-i}} \left[ \sum_{i=0}^{w-1} \widetilde{\nabla} f_{t-i}(\boldsymbol{\lambda}_{t-i}, \boldsymbol{\beta}_{t+1-i}, \mathcal{E}_{t-i}) \right] \right\|^2 \right]$$

$$\leq \sum_{i=0}^{w-1} \sigma_f^2 = w \sigma_f^2. \quad (39)$$

Expanding equation 38 we have

$$\frac{2}{w^2} \sum_{t=1}^{T} \mathbb{E}_{\mathcal{Z}_{t,w}} \left[ \left\| \sum_{i=0}^{w-1} \left[ \mathbb{E}_{\mathcal{E}_{t-i}} \left[ \widetilde{\nabla} f_{t-i}(\boldsymbol{\lambda}_{t-i}, \boldsymbol{\beta}_{t+1-i}, \mathcal{E}_{t-i}) \right] - \nabla F_{t-i}(\boldsymbol{\lambda}_{t-i}) \right] \right\|^2 \right]$$

$$\leq \frac{4}{w^2} \sum_{t=1}^{T} \mathbb{E}_{\mathcal{Z}_{t,w}} \left[ \left\| \sum_{i=0}^{w-1} \left[ \mathbb{E}_{\mathcal{E}_{t-i}} \left[ \widetilde{\nabla} f_{t-i}(\boldsymbol{\lambda}_{t-i}, \boldsymbol{\beta}_{t+1-i}, \mathcal{E}_{t-i}) \right] - \widetilde{\nabla} f_{t-i}(\boldsymbol{\lambda}_{t-i}, \boldsymbol{\beta}_{t+1-i}) \right] \right\|^2 \right] \quad (40)$$

$$+ \frac{4}{w^2} \sum_{t=1}^{T} \left[ \left\| \sum_{i=0}^{w-1} \left[ \widetilde{\nabla} f_{t-i}(\boldsymbol{\lambda}_{t-i}, \boldsymbol{\beta}_{t+1-i}) - \nabla F_{t-i}(\boldsymbol{\lambda}_{t-i}) \right] \right\|^2 \right] \quad (41)$$

Utilizing Lemmas 5 and 7 for equation 40 gives us the expected stochastic gradient bias

$$\frac{4}{w^2} \sum_{t=1}^{T} \mathbb{E}_{\mathcal{Z}_{t,w}} \left[ \left\| \sum_{i=0}^{w-1} \left[ \mathbb{E}_{\mathcal{E}_{t-i}} \left[ \widetilde{\nabla} f_{t-i}(\boldsymbol{\lambda}_{t-i}, \boldsymbol{\beta}_{t+1-i}, \mathcal{E}_{t-i}) \right] - \widetilde{\nabla} f_{t-i}(\boldsymbol{\lambda}_{t-i}, \boldsymbol{\beta}_{t+1-i}) \right] \right\|^2 \right]$$

$$\leq \frac{4}{w^2} \sum_{t=1}^{T} \mathbb{E}_{\mathcal{Z}_{t,w}} \left[ w \sum_{i=0}^{w-1} \left\| \mathbb{E}_{\mathcal{E}_{t-i}} \left[ \widetilde{\nabla} f_{t-i}(\boldsymbol{\lambda}_{t-i}, \boldsymbol{\beta}_{t+1-i}, \mathcal{E}_{t-i}) \right] - \widetilde{\nabla} f_{t-i}(\boldsymbol{\lambda}_{t-i}, \boldsymbol{\beta}_{t+1-i}) \right\|^2 \right]$$

$$\leq \frac{4}{w^2} \sum_{t=1}^{T} \left( w^2 \ell_{f,1}^2 \kappa_g^2 \left( 1 - \frac{\mu_g}{\ell_{g,1}} \right)^{2m} \right) = 4 T \ell_{f,1}^2 \kappa_g^2 \left( 1 - \frac{\mu_g}{\ell_{g,1}} \right)^{2m} \quad (42)$$

Applying Lemma 7 with linearity of expectation to equation 41 results in

$$\frac{4}{w^2} \sum_{t=1}^{T} \left[ \left\| \sum_{i=0}^{w-1} \left[ \widetilde{\nabla} f_{t-i}(\boldsymbol{\lambda}_{t-i}, \boldsymbol{\beta}_{t+1-i}) - \nabla F_{t-i}(\boldsymbol{\lambda}_{t-i}) \right] \right\|^2 \right]$$

$$\leq \frac{4}{w^2} \sum_{t=1}^{T} w \sum_{i=0}^{w-1} \left[ \left\| \widetilde{\nabla} f_{t-i}(\boldsymbol{\lambda}_{t-i}, \boldsymbol{\beta}_{t+1-i}) - \nabla F_{t-i}(\boldsymbol{\lambda}_{t-i}) \right\|^2 \right]$$

$$= \frac{4}{w} \sum_{t=1}^{T} \sum_{i=0}^{w-1} \left[ \left\| \widetilde{\nabla} f_{t-i}(\boldsymbol{\lambda}_{t-i}, \boldsymbol{\beta}_{t+1-i}) - \nabla F_{t-i}(\boldsymbol{\lambda}_{t-i}) \right\|^2 \right] \quad (43)$$

Combining equation 37, equation 38, equation 39, and equation 43, we have the upper bound of

$$\mathbb{E}_{\mathcal{Z}_{t,w}}\left[\sum_{t=1}^{T}\left\|\widetilde{\nabla}f_{t,w}(\boldsymbol{\lambda}_t,\boldsymbol{\beta}_{t+1},\mathcal{Z}_{t,w})-\nabla F_{t,w}\left(\boldsymbol{\lambda}_t\right)\right\|^2\right]\le\frac{4T\sigma_f^2}{w}$$

$$+4T\ell_{f,1}^2\kappa_g^2\left(1-\frac{\mu_g}{\ell_{g,1}}\right)^{2m}+\frac{4}{w}\sum_{t=1}^{T}\sum_{i=0}^{w-1}\left[\left\|\widetilde{\nabla}f_{t-i}(\boldsymbol{\lambda}_{t-i},\boldsymbol{\beta}_{t+1-i})-\nabla F_{t-i}\left(\boldsymbol{\lambda}_{t-i}\right)\right\|^2\right],\quad(44)$$

Taking expectation with respect to $\bar{\zeta}_{t,K+1}$, we utilize the upper bound from Lemma 15. By iterative conditioning and re-indexing the expected cumulative hypergradient error as well as dropping expectation for non-random quantities, we derive an upper bound on equation 44 as

$$\mathbb{E}_{\bar{\zeta}_{t,K+1}}\left[\mathbb{E}_{\mathcal{Z}_{t,w}}\left[\sum_{t=1}^{T}\left\|\widetilde{\nabla}f_{t,w}(\boldsymbol{\lambda}_t,\boldsymbol{\beta}_{t+1},\mathcal{Z}_{t,w})-\nabla F_{t,w}\left(\boldsymbol{\lambda}_t\right)\right\|^2\right]\right]$$

$$\le\frac{4T\sigma_f^2}{w}+4T\ell_{f,1}^2\kappa_g^2\left(1-\frac{\mu_g}{\ell_{g,1}}\right)^{2m}+\frac{2}{w}\sum_{t=1}^{T}\sum_{i=0}^{w-1}\left(\kappa_g^2\nu^{t-i-1}\Delta_{\boldsymbol{\beta}}\right)$$

$$+\frac{2}{w}\sum_{t=1}^{T}\sum_{i=0}^{w-1}\left(4C_{\mu_g}\kappa_g^4\alpha^2\sum_{j=0}^{t-i-2}\nu^{j+1}\left\|\mathcal{G}_{\mathcal{X}}(\boldsymbol{\lambda}_{t-i-j},\nabla F_{t-i-j,w}(\boldsymbol{\lambda}_{t-i-j}),\alpha)\right\|^2\right)$$

$$+\frac{2}{w}\sum_{t=1}^{T}\sum_{i=0}^{w-1}\left(\frac{4C_{\mu_g}\kappa_g^4\alpha^2}{\rho^2}\sum_{j=0}^{t-i-2}\nu^{j+1}A_{t-i,j}\right)$$

$$+\frac{2}{w}\sum_{t=2}^{T}\sum_{i=0}^{w-1}\left(2C_{\mu_g}\kappa_g^2\sum_{j=0}^{t-i-2}\nu^{j+1}B_{t-i,j}\right)+\frac{2}{w}\sum_{t=1}^{T}\sum_{i=0}^{w-1}\left(\frac{C_K\kappa_g^2\eta^2\sigma_g^2}{s}\sum_{j=0}^{t-i-2}\nu^j\right),\quad(45)$$

where

$$A_{t,j}:=\mathbb{E}_{\bar{\zeta}_{t-j,K+1}}\left[\mathbb{E}_{\mathcal{Z}_{t-j,w}}\left[\left\|\widetilde{\nabla}f_{t-j,w}(\boldsymbol{\lambda}_{t-j},\boldsymbol{\beta}_{t+1-j},\mathcal{Z}_{t-j,w})-\nabla F_{t-j,w}(\boldsymbol{\lambda}_{t-j})\right\|^2\right]\right]$$

$$B_{t,j}:=\left\|\boldsymbol{\beta}_{t-j}^*(\boldsymbol{\lambda}_{t-1-j})-\boldsymbol{\beta}_{t-1-j}^*(\boldsymbol{\lambda}_{t-1-j})\right\|^2.$$

Given $\nu<1$, it holds that $\sum_{j=0}^{t-2}\nu^j<\sum_{j=0}^{\infty}\nu^j=\frac{1}{1-\nu}$, which lets us upper bound equation 45 as

$$\mathbb{E}_{\bar{\zeta}_{t,K+1}}\left[\mathbb{E}_{\mathcal{Z}_{t,w}}\left[\sum_{t=1}^{T}\left\|\widetilde{\nabla}f_{t,w}(\boldsymbol{\lambda}_t,\boldsymbol{\beta}_{t+1},\mathcal{Z}_{t,w})-\nabla F_{t,w}\left(\boldsymbol{\lambda}_t\right)\right\|^2\right]\right]\le\frac{4T\sigma_f^2}{w}+4T\ell_{f,1}^2\kappa_g^2\left(1-\frac{\mu_g}{\ell_{g,1}}\right)^{2m}$$

$$+\frac{2}{w}\sum_{t=1}^{T}\sum_{i=0}^{w-1}\left(\kappa_g^2\nu^{t-i-1}\Delta_{\boldsymbol{\beta}}+\frac{4C_{\mu_g}\kappa_g^4\alpha^2}{1-\nu}\left\|\mathcal{G}_{\mathcal{X}}(\boldsymbol{\lambda}_{t-i},\nabla F_{t-i,w}(\boldsymbol{\lambda}_{t-i}),\alpha)\right\|^2\right)$$

$$+\frac{2}{w}\sum_{t=1}^{T}\sum_{i=0}^{w-1}\left(\frac{4C_{\mu_g}\kappa_g^4\alpha^2}{(1-\nu)\rho^2}\mathbb{E}_{\bar{\zeta}_{t-i,K+1}}\left[\mathbb{E}_{\mathcal{Z}_{t-i,w}}\left[\left\|\widetilde{\nabla}f_{t-i,w}(\boldsymbol{\lambda}_{t-i},\boldsymbol{\beta}_{t+1-i},\mathcal{Z}_{t-i,w})-\nabla F_{t-i,w}(\boldsymbol{\lambda}_{t-i})\right\|^2\right]\right]\right)$$

$$+\frac{2}{w}\sum_{t=2}^{T}\sum_{i=0}^{w-1}\left(\frac{2C_{\mu_g}\kappa_g^2}{1-\nu}\left\|\boldsymbol{\beta}_{t-i}^*(\boldsymbol{\lambda}_{t-1-i})-\boldsymbol{\beta}_{t-1-i}^*(\boldsymbol{\lambda}_{t-1-i})\right\|^2\right)+\frac{2}{w}\sum_{t=1}^{T}\sum_{i=0}^{w-1}\left(\frac{C_K\kappa_g^2\eta^2\sigma_g^2}{(1-\nu)s}\right).$$
$$(46)$$

Next we derive the upper bound of equation 46 as

$$\mathbb{E}_{\bar{\zeta}_{t,K+1}}\left[\mathbb{E}_{\mathcal{Z}_{t,w}}\left[\sum_{t=1}^{T}\left\|\widetilde{\nabla}f_{t,w}(\boldsymbol{\lambda}_t,\boldsymbol{\beta}_{t+1},\mathcal{Z}_{t,w})-\nabla F_{t,w}\left(\boldsymbol{\lambda}_t\right)\right\|^2\right]\right]\leq\frac{4T\sigma_f^2}{w}+4T\ell_{f,1}^2\kappa_g^2\left(1-\frac{\mu_g}{\ell_{g,1}}\right)^{2m}$$

$$+\frac{2\kappa_g^2\Delta_{\boldsymbol{\beta}}}{(1-\nu)}+\frac{8C_{\mu_g}\kappa_g^4\alpha^2}{(1-\nu)}\sum_{t=1}^{T}\|\mathcal{G}_{\mathcal{X}}(\boldsymbol{\lambda}_t,\nabla F_{t,w}(\boldsymbol{\lambda}_t),\alpha)\|^2$$

$$+\frac{8C_{\mu_g}\kappa_g^4\alpha^2}{\rho^2(1-\nu)}\mathbb{E}_{\bar{\zeta}_{t,K+1}}\left[\mathbb{E}_{\mathcal{Z}_{t,w}}\left[\sum_{t=1}^{T}\left\|\widetilde{\nabla}f_{t,w}\left(\boldsymbol{\lambda}_t,\boldsymbol{\beta}_{t+1},\mathcal{Z}_{t,w}\right)-\nabla F_{t,w}(\boldsymbol{\lambda}_t)\right\|^2\right]\right]$$

$$+\frac{4C_{\mu_g}\kappa_g^2}{(1-\nu)}\sum_{t=2}^{T}\left\|\boldsymbol{\beta}_t^*(\boldsymbol{\lambda}_{t-1})-\boldsymbol{\beta}_{t-1}^*(\boldsymbol{\lambda}_{t-1})\right\|^2+\frac{2TC_K\kappa_g^2\eta^2\sigma_g^2}{s(1-\nu)}.$$

which implies through linearity of expectation that

$$\left(1-\frac{8C_{\mu_g}\kappa_g^4\alpha^2}{\rho^2(1-\nu)}\right)\mathbb{E}_{\bar{\zeta}_{t,K+1}}\left[\mathbb{E}_{\mathcal{Z}_{t,w}}\left[\sum_{t=1}^{T}\left\|\widetilde{\nabla}f_{t,w}(\boldsymbol{\lambda}_t,\boldsymbol{\beta}_{t+1},\mathcal{Z}_{t,w})-\nabla F_{t,w}\left(\boldsymbol{\lambda}_t\right)\right\|^2\right]\right]$$

$$\leq\frac{4T\sigma_f^2}{w}+4T\ell_{f,1}^2\kappa_g^2\left(1-\frac{\mu_g}{\ell_{g,1}}\right)^{2m}+\frac{2\kappa_g^2\Delta_{\boldsymbol{\beta}}}{(1-\nu)}+\frac{8C_{\mu_g}\kappa_g^4\alpha^2}{(1-\nu)}\sum_{t=1}^{T}\|\mathcal{G}_{\mathcal{X}}(\boldsymbol{\lambda}_t,\nabla F_{t,w}(\boldsymbol{\lambda}_t),\alpha)\|^2$$

$$+\frac{4C_{\mu_g}\kappa_g^2}{(1-\nu)}\sum_{t=2}^{T}\left\|\boldsymbol{\beta}_t^*(\boldsymbol{\lambda}_{t-1})-\boldsymbol{\beta}_{t-1}^*(\boldsymbol{\lambda}_{t-1})\right\|^2+\frac{2TC_K\kappa_g^2\eta^2\sigma_g^2}{s(1-\nu)}.$$

As $0<\alpha\leq\frac{\rho\sqrt{(1-\nu)}}{\kappa_g^2\sqrt{72C_{\mu_g}}}$

$$\left(1-\frac{8C_{\mu_g}\kappa_g^4}{\rho^2(1-\nu)}\right)\geq\frac{8}{9},$$

we have the upper bound of

$$\mathbb{E}_{\bar{\zeta}_{t,K+1}}\left[\mathbb{E}_{\mathcal{Z}_{t,w}}\left[\sum_{t=1}^{T}\left\|\widetilde{\nabla}f_{t,w}(\boldsymbol{\lambda}_t,\boldsymbol{\beta}_{t+1},\mathcal{Z}_{t,w})-\nabla F_{t,w}\left(\boldsymbol{\lambda}_t\right)\right\|^2\right]\right]$$

$$\leq\frac{9T\sigma_f^2}{2w}+\frac{9T\ell_{f,1}^2\kappa_g^2}{2}\left(1-\frac{\mu_g}{\ell_{g,1}}\right)^{2m}+\frac{9\kappa_g^2\Delta_{\boldsymbol{\beta}}}{4(1-\nu)}+\frac{\rho^2}{8}\sum_{t=1}^{T}\|\mathcal{G}_{\mathcal{X}}(\boldsymbol{\lambda}_t,\nabla F_{t,w}(\boldsymbol{\lambda}_t),\alpha)\|^2$$

$$+\frac{9C_{\mu_g}\kappa_g^2}{2(1-\nu)}\sum_{t=2}^{T}\left[\left\|\boldsymbol{\beta}_t^*(\boldsymbol{\lambda}_{t-1})-\boldsymbol{\beta}_{t-1}^*(\boldsymbol{\lambda}_{t-1})\right\|^2\right]+\frac{9TC_K\kappa_g^2\eta^2\sigma_g^2}{4s(1-\nu)}.$$

$\square$

The following theorem presents the bilevel local regret of Algorithm 3.

**Theorem 17.** *Suppose Assumptions A-F. Let the inner step be $\eta=\Omega(1/\mu_g)$ such that $\eta\leq\min\{\frac{2}{\ell_{g,1}+\mu_g},\frac{1}{w}\}$, outer step be $\alpha\leq\min\{\frac{3\rho}{4\ell_{F,1}},\frac{\rho\sqrt{(1-\nu)}}{\kappa_g^2\sqrt{72C_{\mu_g}}}\}$, and batch size for stochastic inverse Hessian approximation $m=\log\left(w\right)/\log\left(1-\frac{\mu_g}{\ell_{g,1}}\right)+1$. Then the bilevel local regret of the single-loop ($K=1$) and sample-efficient ($s=O(1)$) instance of Algorithm 3 satisfies*

$$BLR_w(T)\leq O\left(\frac{T\kappa_g^3}{w\rho}\left(1+\frac{\kappa_g^2+\sigma_f^2+\kappa_g^2\sigma_g^2}{\rho}\right)+\frac{V_{1,T}}{\rho}+\frac{\kappa_g^2H_{2,T}}{\rho^2}\right) \tag{47}$$

*where comparator sequences $V_{1,T}$ and $H_{2,T}$, $\sigma_g^2,\sigma_f^2$ are finite variances from Assumption C, and $\kappa_g>1$ is the condition number of the inner level objective.*

*Proof.* Note, Lemma H.2 of Lin et al. (2024) shows how with Assumption A, we have

$$F_{t,w}\left(\boldsymbol{\lambda}_{t+1}\right) - F_{t,w}\left(\boldsymbol{\lambda}_t\right) \leq \langle \nabla F_{t,w}\left(\boldsymbol{\lambda}_t\right), \boldsymbol{\lambda}_{t+1} - \boldsymbol{\lambda}_t \rangle + \frac{\ell_{F,1}}{2}\left\|\boldsymbol{\lambda}_{t+1} - \boldsymbol{\lambda}_t\right\|^2.$$

then by substituting the step of $\mathcal{G}_{\mathcal{X}}\left(\boldsymbol{\lambda}_t, \widetilde{\nabla} f_{t,w}(\boldsymbol{\lambda}_t, \boldsymbol{\beta}_{t+1}, \mathcal{Z}_{t,w}), \alpha\right) := \frac{1}{\alpha}\left(\boldsymbol{\lambda}_t - \boldsymbol{\lambda}_{t+1}\right)$, we have

$$F_{t,w}\left(\boldsymbol{\lambda}_{t+1}\right) - F_{t,w}\left(\boldsymbol{\lambda}_t\right) \leq \langle \nabla F_{t,w}\left(\boldsymbol{\lambda}_t\right), \boldsymbol{\lambda}_{t+1} - \boldsymbol{\lambda}_t \rangle + \frac{\ell_{F,1}}{2}\left\|\boldsymbol{\lambda}_{t+1} - \boldsymbol{\lambda}_t\right\|^2$$

$$= -\alpha\left\langle \nabla F_{t,w}\left(\boldsymbol{\lambda}_t\right), \mathcal{G}_{\mathcal{X}}\left(\boldsymbol{\lambda}_t, \widetilde{\nabla} f_{t,w}(\boldsymbol{\lambda}_t, \boldsymbol{\beta}_{t+1}, \mathcal{Z}_{t,w}), \alpha\right)\right\rangle$$

$$+ \frac{\alpha^2 \ell_{F,1}}{2}\left\|\mathcal{G}_{\mathcal{X}}\left(\boldsymbol{\lambda}_t, \widetilde{\nabla} f_{t,w}(\boldsymbol{\lambda}_t, \boldsymbol{\beta}_{t+1}, \mathcal{Z}_{t,w}), \alpha\right)\right\|^2,$$

$$= -\alpha\left\langle \widetilde{\nabla} f_{t,w}(\boldsymbol{\lambda}_t, \boldsymbol{\beta}_{t+1}, \mathcal{Z}_{t,w}), \mathcal{G}_{\mathcal{X}}\left(\boldsymbol{\lambda}_t, \widetilde{\nabla} f_{t,w}(\boldsymbol{\lambda}_t, \boldsymbol{\beta}_{t+1}, \mathcal{Z}_{t,w}), \alpha\right)\right\rangle$$

$$+ \alpha\left\langle \widetilde{\nabla} f_{t,w}(\boldsymbol{\lambda}_t, \boldsymbol{\beta}_{t+1}, \mathcal{Z}_{t,w}) - \nabla F_{t,w}\left(\boldsymbol{\lambda}_t\right), \mathcal{G}_{\mathcal{X}}\left(\boldsymbol{\lambda}_t, \widetilde{\nabla} f_{t,w}(\boldsymbol{\lambda}_t, \boldsymbol{\beta}_{t+1}, \mathcal{Z}_{t,w}), \alpha\right)\right\rangle$$

$$+ \frac{\alpha^2 \ell_{F,1}}{2}\left\|\mathcal{G}_{\mathcal{X}}\left(\boldsymbol{\lambda}_t, \widetilde{\nabla} f_{t,w}(\boldsymbol{\lambda}_t, \boldsymbol{\beta}_{t+1}, \mathcal{Z}_{t,w}), \alpha\right)\right\|^2. \quad (48)$$

With Lemma 9, note for $\boldsymbol{q} = \widetilde{\nabla} f_{t,w}(\boldsymbol{\lambda}_t, \boldsymbol{\beta}_{t+1}, \mathcal{Z}_{t,w})$

$$\alpha\left\langle \widetilde{\nabla} f_{t,w}(\boldsymbol{\lambda}_t, \boldsymbol{\beta}_{t+1}, \mathcal{Z}_{t,w}), \mathcal{G}_{\mathcal{X}}\left(\boldsymbol{\lambda}_t, \widetilde{\nabla} f_{t,w}(\boldsymbol{\lambda}_t, \boldsymbol{\beta}_{t+1}, \mathcal{Z}_{t,w}), \alpha\right)\right\rangle$$

$$\geq \alpha\rho\left\|\mathcal{G}_{\mathcal{X}}\left(\boldsymbol{\lambda}_t, \widetilde{\nabla} f_{t,w}(\boldsymbol{\lambda}_t, \boldsymbol{\beta}_{t+1}, \mathcal{Z}_{t,w}), \alpha\right)\right\|^2 + h(\boldsymbol{\lambda}_{t+1}) - h(\boldsymbol{\lambda}_t) \quad (49)$$

and further we get the following based on a variation of Young's Inequality

$$\left\langle \widetilde{\nabla} f_{t,w}(\boldsymbol{\lambda}_t, \boldsymbol{\beta}_{t+1}, \mathcal{Z}_{t,w}) - \nabla F_{t,w}\left(\boldsymbol{\lambda}_t\right), \mathcal{G}_{\mathcal{X}}\left(\boldsymbol{\lambda}_t, \widetilde{\nabla} f_{t,w}(\boldsymbol{\lambda}_t, \boldsymbol{\beta}_{t+1}, \mathcal{Z}_{t,w}), \alpha\right)\right\rangle$$

$$\leq \frac{1}{\rho}\left\|\widetilde{\nabla} f_{t,w}(\boldsymbol{\lambda}_t, \boldsymbol{\beta}_{t+1}, \mathcal{Z}_{t,w}) - \nabla F_{t,w}\left(\boldsymbol{\lambda}_t\right)\right\|^2$$

$$+ \frac{\rho}{4}\left\|\mathcal{G}_{\mathcal{X}}\left(\boldsymbol{\lambda}_t, \widetilde{\nabla} f_{t,w}(\boldsymbol{\lambda}_t, \boldsymbol{\beta}_{t+1}, \mathcal{Z}_{t,w}), \alpha\right)\right\|^2. \quad (50)$$

Combining equation 49 and equation 50 in equation 48 we get

$$F_{t,w}\left(\boldsymbol{\lambda}_{t+1}\right) - F_{t,w}\left(\boldsymbol{\lambda}_t\right) \leq \left(\frac{\alpha^2 \ell_{F,1}}{2} - \frac{3\alpha\rho}{4}\right)\left\|\mathcal{G}_{\mathcal{X}}\left(\boldsymbol{\lambda}_t, \widetilde{\nabla} f_{t,w}(\boldsymbol{\lambda}_t, \boldsymbol{\beta}_{t+1}, \mathcal{Z}_{t,w}), \alpha\right)\right\|^2$$

$$+ \frac{\alpha}{\rho}\left\|\widetilde{\nabla} f_{t,w}(\boldsymbol{\lambda}_t, \boldsymbol{\beta}_{t+1}, \mathcal{Z}_{t,w}) - \nabla F_{t,w}\left(\boldsymbol{\lambda}_t\right)\right\|^2 + h(\boldsymbol{\lambda}_t) - h(\boldsymbol{\lambda}_{t+1}),$$

which as $0 < \alpha \leq \frac{3\rho}{4\ell_{F,1}}$ results in the further upper bound of

$$F_{t,w}\left(\boldsymbol{\lambda}_{t+1}\right) - F_{t,w}\left(\boldsymbol{\lambda}_t\right) \leq \frac{3\alpha\rho}{8}\left\|\mathcal{G}_{\mathcal{X}}\left(\boldsymbol{\lambda}_t, \widetilde{\nabla} f_{t,w}(\boldsymbol{\lambda}_t, \boldsymbol{\beta}_{t+1}, \mathcal{Z}_{t,w}), \alpha\right)\right\|^2$$

$$+ \frac{\alpha}{\rho}\left\|\widetilde{\nabla} f_{t,w}(\boldsymbol{\lambda}_t, \boldsymbol{\beta}_{t+1}, \mathcal{Z}_{t,w}) - \nabla F_{t,w}\left(\boldsymbol{\lambda}_t\right)\right\|^2 + h(\boldsymbol{\lambda}_t) - h(\boldsymbol{\lambda}_{t+1}). \quad (51)$$

Further, we have

$$\left\|\mathcal{G}_{\mathcal{X}}(\boldsymbol{\lambda}_t, \nabla F_{t,w}(\boldsymbol{\lambda}_t), \alpha)\right\|^2 \leq 2\left\|\mathcal{G}_{\mathcal{X}}\left(\boldsymbol{\lambda}_t, \widetilde{\nabla} f_{t,w}(\boldsymbol{\lambda}_t, \boldsymbol{\beta}_{t+1}, \mathcal{Z}_{t,w}), \alpha\right)\right\|^2$$

$$+ 2\left\|\mathcal{G}_{\mathcal{X}}\left(\boldsymbol{\lambda}_t, \widetilde{\nabla} f_{t,w}(\boldsymbol{\lambda}_t, \boldsymbol{\beta}_{t+1}, \mathcal{Z}_{t,w}), \alpha\right) - \mathcal{G}_{\mathcal{X}}(\boldsymbol{\lambda}_t, \nabla F_{t,w}(\boldsymbol{\lambda}_t), \alpha)\right\|^2$$

$$\leq 2\left\|\mathcal{G}_{\mathcal{X}}\left(\boldsymbol{\lambda}_t, \widetilde{\nabla} f_{t,w}(\boldsymbol{\lambda}_t, \boldsymbol{\beta}_{t+1}, \mathcal{Z}_{t,w}), \alpha\right)\right\|^2 + \frac{2}{\rho^2}\left\|\widetilde{\nabla} f_{t,w}(\boldsymbol{\lambda}_t, \boldsymbol{\beta}_{t+1}, \mathcal{Z}_{t,w}) - \nabla F_{t,w}\left(\boldsymbol{\lambda}_t\right)\right\|^2,$$

where the last inequality comes from through Lemma 10. Then we have

$$-\left\|\mathcal{G}_{\mathcal{X}}\left(\boldsymbol{\lambda}_t, \widetilde{\nabla} f_{t,w}(\boldsymbol{\lambda}_t, \boldsymbol{\beta}_{t+1}, \mathcal{Z}_{t,w}), \alpha\right)\right\|^2 \leq -\frac{1}{2}\left\|\mathcal{G}_{\mathcal{X}}(\boldsymbol{\lambda}_t, \nabla F_{t,w}(\boldsymbol{\lambda}_t), \alpha)\right\|^2$$

$$+ \frac{1}{\rho^2}\left\|\widetilde{\nabla} f_{t,w}(\boldsymbol{\lambda}_t, \boldsymbol{\beta}_{t+1}, \mathcal{Z}_{t,w}) - \nabla F_{t,w}\left(\boldsymbol{\lambda}_t\right)\right\|^2. \quad (52)$$

Substituting equation 52 in equation 51

$$F_{t,w}(\boldsymbol{\lambda}_{t+1}) - F_{t,w}(\boldsymbol{\lambda}_t) \leq -\frac{3\alpha\rho}{16}\|\mathcal{G}_{\mathcal{X}}(\boldsymbol{\lambda}_t, \nabla F_{t,w}(\boldsymbol{\lambda}_t), \alpha)\|^2$$

$$+\left(\frac{\alpha}{\rho} + \frac{3\alpha}{8\rho}\right)\left\|\widetilde{\nabla}f_{t,w}(\boldsymbol{\lambda}_t, \boldsymbol{\beta}_{t+1}, \mathcal{Z}_{t,w}) - \nabla F_{t,w}(\boldsymbol{\lambda}_t)\right\|^2 + h(\boldsymbol{\lambda}_t) - h(\boldsymbol{\lambda}_{t+1})$$

Telescoping $t = 1, \ldots, T$ and taking expectation with respect to $\bar{\zeta}_{t,k}$ and $\mathcal{Z}_{t,w}$ gives us

$$\frac{3\alpha\rho}{16}\sum_{t=1}^{T}\|\mathcal{G}_{\mathcal{X}}(\boldsymbol{\lambda}_t, \nabla F_{t,w}(\boldsymbol{\lambda}_t), \alpha)\|^2 \leq \sum_{t=1}^{T}(F_{t,w}(\boldsymbol{\lambda}_t) - F_{t,w}(\boldsymbol{\lambda}_{t+1}))$$

$$+\frac{11\alpha}{8\rho}\mathbb{E}_{\bar{\zeta}_{t,K+1}}\left[\mathbb{E}_{\mathcal{Z}_{t,w}}\left[\sum_{t=1}^{T}\left\|\widetilde{\nabla}f_{t,w}(\boldsymbol{\lambda}_t, \boldsymbol{\beta}_{t+1}, \mathcal{Z}_{t,w}) - \nabla F_{t,w}(\boldsymbol{\lambda}_t)\right\|^2\right]\right] + \Delta_h, \quad (53)$$

where $\Delta_h := h(\boldsymbol{\lambda}_1) - h(\boldsymbol{\lambda}_{T+1})$. Substituting the result of Lemma 16 in equation 53

$$\frac{3\alpha\rho}{16}\sum_{t=1}^{T}\|\mathcal{G}_{\mathcal{X}}(\boldsymbol{\lambda}_t, \nabla F_{t,w}(\boldsymbol{\lambda}_t), \alpha)\|^2 \leq \sum_{t=1}^{T}(F_{t,w}(\boldsymbol{\lambda}_t) - F_{t,w}(\boldsymbol{\lambda}_{t+1})) + \Delta_h$$

$$+\frac{11\alpha}{8\rho}\left(\frac{9T\sigma_f^2}{2w} + \frac{9T\ell_{f,1}^2\kappa_g^2}{2}\left(1 - \frac{\mu_g}{\ell_{g,1}}\right)^{2m} + \frac{9\kappa_g^2\Delta_{\boldsymbol{\beta}}}{4(1-\nu)} + \frac{9TC_K\kappa_g^2\eta^2\sigma_g^2}{4S(1-\nu)}\right)$$

$$+\frac{11\alpha}{8\rho}\left(\frac{\rho^2}{8}\sum_{t=1}^{T}\|\mathcal{G}_{\mathcal{X}}(\boldsymbol{\lambda}_t, \nabla F_{t,w}(\boldsymbol{\lambda}_t), \alpha)\|^2 + \frac{9C_{\mu_g}\kappa_g^2}{2(1-\nu)}\sum_{t=2}^{T}\left[\|\boldsymbol{\beta}_t^*(\boldsymbol{\lambda}_{t-1}) - \boldsymbol{\beta}_{t-1}^*(\boldsymbol{\lambda}_{t-1})\|^2\right]\right)$$

$$(54)$$

we have to rearrange

$$\frac{\alpha\rho}{64}\sum_{t=1}^{T}\|\mathcal{G}_{\mathcal{X}}(\boldsymbol{\lambda}_t, \nabla F_{t,w}(\boldsymbol{\lambda}_t), \alpha)\|^2 \leq \sum_{t=1}^{T}(F_{t,w}(\boldsymbol{\lambda}_t) - F_{t,w}(\boldsymbol{\lambda}_{t+1})) + \Delta_h$$

$$+\frac{99\alpha}{32\rho}\left(\frac{2T\sigma_f^2}{w} + 2T\ell_{f,1}^2\kappa_g^2\left(1 - \frac{\mu_g}{\ell_{g,1}}\right)^{2m} + \frac{\kappa_g^2\Delta_{\boldsymbol{\beta}}}{(1-\nu)} + \frac{TC_K\kappa_g^2\eta^2\sigma_g^2}{s(1-\nu)}\right)$$

$$+\frac{99\alpha C_{\mu_g}\kappa_g^2}{16\rho(1-\nu)}\sum_{t=2}^{T}\|\boldsymbol{\beta}_t^*(\boldsymbol{\lambda}_{t-1}) - \boldsymbol{\beta}_{t-1}^*(\boldsymbol{\lambda}_{t-1})\|^2 \quad (55)$$

or more succinctly with the choice of $s = O(1)$, $m = \log(w)/\log\left(1 - \frac{\mu_g}{\ell_{g,1}}\right) + 1$, and inner step size of $\eta \leq \frac{1}{w}$ we have

$$\sum_{t=1}^{T}\|\mathcal{G}_{\mathcal{X}}(\boldsymbol{\lambda}_t, \nabla F_{t,w}(\boldsymbol{\lambda}_t), \alpha)\|^2 \leq \frac{64}{\alpha\rho}\left(\sum_{t=1}^{T}(F_{t,w}(\boldsymbol{\lambda}_t) - F_{t,w}(\boldsymbol{\lambda}_{t+1})) + \Delta_h\right)$$

$$+\frac{198}{\rho^2}\frac{T}{w}\left(2\sigma_f^2 + 2\ell_{f,1}^2\kappa_g^2 + \frac{C_K\kappa_g^2\sigma_g^2}{(1-\nu)}\right) + \frac{198}{\rho^2}\frac{\kappa_g^2\Delta_{\boldsymbol{\beta}}}{(1-\nu)}$$

$$+\frac{396C_{\mu_g}\kappa_g^2}{\rho^2(1-\nu)}\sum_{t=2}^{T}\|\boldsymbol{\beta}_t^*(\boldsymbol{\lambda}_{t-1}) - \boldsymbol{\beta}_{t-1}^*(\boldsymbol{\lambda}_{t-1})\|^2 \quad (56)$$

Using the result of Lemma 13, we have

$$\sum_{t=1}^{T}(F_{t,w}(\boldsymbol{\lambda}_t) - F_{t,w}(\boldsymbol{\lambda}_{t+1})) \leq \frac{2TQ}{w} + V_{1,T} \quad (57)$$

or all together

$$\sum_{t=1}^{T} \|\mathcal{G}_{\mathcal{X}}(\boldsymbol{\lambda}_t, \nabla F_{t,w}(\boldsymbol{\lambda}_t), \alpha)\|^2 \leq \frac{64}{\alpha\rho} \left( \frac{2TQ}{w} + V_{1,T} + \Delta_h \right)$$

$$+ \frac{198}{\rho^2} \frac{T}{w} \left( 2\sigma_f^2 + 2\ell_{f,1}^2 \kappa_g^2 + \frac{C_K \kappa_g^2 \sigma_g^2}{(1-\nu)} \right) + \frac{198}{\rho^2} \frac{\kappa_g^2 \Delta_{\boldsymbol{\beta}}}{(1-\nu)}$$

$$+ \frac{396 C_{\mu_g} \kappa_g^2}{\rho^2(1-\nu)} \sum_{t=2}^{T} \left\| \boldsymbol{\beta}_t^*(\boldsymbol{\lambda}_{t-1}) - \boldsymbol{\beta}_{t-1}^*(\boldsymbol{\lambda}_{t-1}) \right\|^2 \tag{58}$$

which by recalling definition of $H_{2,T}$, and as $\ell_{F,1} = O(\kappa_g^3)$ this implies $\alpha \leq \min\{\frac{3\rho}{4\ell_{F,1}}, \frac{\rho\sqrt{(1-\nu)}}{\kappa_g^2\sqrt{72C_{\mu_g}}}\} = O\left(\frac{1}{\kappa_g^3}\right)$ this implies the bilevel local regret of Algorithm 3 is

$$BLR_w(T) \leq O\left( \frac{T\kappa_g^3}{w\rho} \left( 1 + \frac{\kappa_g^2 + \sigma_f^2 + \kappa_g^2\sigma_g^2}{\rho} \right) + \frac{V_{1,T}}{\rho} + \frac{\kappa_g^2 H_{2,T}}{\rho^2} \right) \tag{59}$$

$\square$

## C  PROOF IN DETERMINISTIC SETTING

First, we introduce some required lemmas. Lemma 18 provides an analytical form to compute the hypergradient via iterative differentiation.

**Lemma 18.** *(Proposition 2 in Ji et al. (2021)) The partial $\frac{\partial f_t(\boldsymbol{\lambda}_t, \boldsymbol{\omega}_t^K)}{\partial \boldsymbol{\lambda}}$ takes an analytical form of $\frac{\partial f_t(\boldsymbol{\lambda}_t, \boldsymbol{\omega}_t^K)}{\partial \boldsymbol{\lambda}} =$*

$$\nabla_{\boldsymbol{\lambda}} f_t\left(\boldsymbol{\lambda}_t, \boldsymbol{\omega}_t^K\right) - \eta \sum_{k=0}^{K-1} \nabla_{\boldsymbol{\lambda},\boldsymbol{\omega}}^2 g_t\left(\boldsymbol{\lambda}_t, \boldsymbol{\omega}_t^k\right) H_{\boldsymbol{\omega},\boldsymbol{\omega}} \nabla_{\boldsymbol{\omega}} f_t\left(\boldsymbol{\lambda}_t, \boldsymbol{\omega}_t^K\right), \tag{60}$$

*where $H_{\boldsymbol{\omega},\boldsymbol{\omega}} := \prod_{j=k+1}^{K-1} \left( I_{d_2} - \eta\nabla_{\boldsymbol{\omega},\boldsymbol{\omega}}^2 g_t\left(\boldsymbol{\lambda}_t, \boldsymbol{\omega}_t^j\right) \right)$, the $d_2$-identity matrix is denoted $I_{d_2}$, with $\eta > 0$ and $K$ as the step size and number of iterations for the inner loop.*

Lemma 19 provides an upper bound on the hypergradient error when utilizing an iterative differentiation approach for estimation.

**Lemma 19.** *(Lemma 6 in Ji et al. (2021)) Suppose Assumptions A and B are satisfied with $\eta < \frac{1}{\ell_{g,1}}$ and $K \geq 1$. Then we have $\forall t \in [1, T]$*

$$\left\| \frac{\partial f_t(\boldsymbol{\lambda}, \boldsymbol{\omega}_t^K)}{\partial \boldsymbol{\lambda}} - \nabla F_t(\boldsymbol{\lambda}) \right\|$$

$$\leq \left( L_1(1 - \eta\mu_g)^{\frac{K}{2}} + L_2(1 - \eta\mu_g)^{\frac{K-1}{2}} \right) \|\boldsymbol{\beta}_t - \boldsymbol{\beta}_t^*(\boldsymbol{\lambda})\| + L_3(1 - \eta\mu_g)^K, \tag{61}$$

*where $L_1 = \kappa_g(\ell_{g,1} + \mu_g)$, $L_2 = \frac{2\ell_{f,0}\ell_{g,2}}{\mu_g}(1 + \kappa_g)$, and $L_3 = \ell_{f,0}\kappa_g$.*

Similar to the deterministic setting, we apply time-smoothing as specified by the estimator $\widetilde{\nabla} f_{t,w}(\boldsymbol{\lambda}_t, \boldsymbol{\beta}_{t+1})$ defined for all $t \in [1, T]$, window size $w \geq 1$ as

$$\widetilde{\nabla} f_{t,w}(\boldsymbol{\lambda}_t, \boldsymbol{\beta}_{t+1}) := \frac{1}{w} \sum_{i=0}^{w-1} \widetilde{\nabla} f_{t-i}(\boldsymbol{\lambda}_{t-i}, \boldsymbol{\beta}_{t+1-i}), \quad f_t = 0 \,\forall t \leq 0 \tag{62}$$

Next we state our Bregman-bilevel optimizer in the deterministic setting.

---

**Algorithm 4** OBBO: Deterministic Online Bregman Bilevel Optimizer

---

**Require:** Horizon $T$; inner steps $K$; step sizes $\alpha, \eta > 0$; Bregman reference $\phi$; window $w \geq 1$

1: Initialize $\boldsymbol{\beta}_1 \in \mathbb{R}^{d_2}$, $\boldsymbol{\lambda}_1 \in \mathcal{X}$
2: **for** $t = 1\,T$ **do**
3: $\quad \boldsymbol{\omega}_t^0 \leftarrow \boldsymbol{\beta}_t$
4: $\quad$ **for** $k = 1\,K$ **do**
5: $\quad\quad \boldsymbol{\omega}_t^k \leftarrow \boldsymbol{\omega}_t^{k-1} - \eta \, \nabla_{\boldsymbol{\omega}} g_t(\boldsymbol{\lambda}_t, \boldsymbol{\omega}_t^{k-1})$
6: $\quad$ **end for**
7: $\quad \boldsymbol{\beta}_{t+1} \leftarrow \boldsymbol{\omega}_t^K$
8: $\quad \widetilde{\nabla} f_t(\boldsymbol{\lambda}_t, \boldsymbol{\beta}_{t+1}) \leftarrow \partial f_t(\boldsymbol{\lambda}_t, \boldsymbol{\beta}_{t+1})/\partial \boldsymbol{\lambda}$ $\qquad\qquad\qquad$ ▷ from equation 60
9: $\quad$ Store $\widetilde{\nabla} f_t(\boldsymbol{\lambda}_t, \boldsymbol{\beta}_{t+1})$ in memory
10: $\quad \boldsymbol{q} \leftarrow \widetilde{\nabla} f_{t,w}(\boldsymbol{\lambda}_t, \boldsymbol{\beta}_{t+1})$ $\qquad\qquad\qquad$ ▷ from equation 62 with window $w$
11: $\quad \boldsymbol{u} \leftarrow \boldsymbol{\lambda}_t$
12: $\quad \boldsymbol{\lambda}_{t+1} \leftarrow \textsc{GeneralGradStep}(\boldsymbol{u}, \boldsymbol{q}, \alpha, \phi)$ $\qquad\qquad\qquad$ ▷ Alg. 2
13: **end for**
14: **return** $\boldsymbol{\lambda}_{T+1}, \boldsymbol{\beta}_{T+1}$

---

The following Lemma provides an upper bound on the cumulative difference between the time-smoothed outer level objective $F_{t,w}(\boldsymbol{\lambda})$ evaluated at $\boldsymbol{\lambda}_t$ and $\boldsymbol{\lambda}_{t+1}$ in terms of the outer level objective upper bound $Q$, window size $w$, and the comparator sequence $V_{1,T}$.

**Lemma 20.** *Suppose Assumption E. If our Algorithm 4 is applied with window size $w \geq 1$ to generate the sequence $\{\boldsymbol{\lambda}_t\}_{t=1}^T$, then we have*

$$\sum_{t=1}^T \left( F_{t,w}(\boldsymbol{\lambda}_t) - F_{t,w}(\boldsymbol{\lambda}_{t+1}) \right) \leq \frac{2TQ}{w} + V_{1,T}.$$

*where $V_{1,T} := \sum_{t=1}^T \sup_{\boldsymbol{\lambda} \in \mathcal{X}} \left[ F_{t+1}(\boldsymbol{\lambda}) - F_t(\boldsymbol{\lambda}) \right]$*

*Proof.* By definition, in the deterministic setting, we have $F_t(\boldsymbol{\lambda}) \triangleq f_t(\boldsymbol{\lambda}, \boldsymbol{\beta}_t^*(\boldsymbol{\lambda}))$. Then it holds

$$\sum_{t=1}^T \left( F_{t,w}(\boldsymbol{\lambda}_t) - F_{t,w}(\boldsymbol{\lambda}_{t+1}) \right)$$

$$= \sum_{t=1}^T \frac{1}{w} \sum_{i=0}^{w-1} \left( f_{t-i}\left(\boldsymbol{\lambda}_{t-i}, \boldsymbol{\beta}_{t-i}^*(\boldsymbol{\lambda}_{t-i})\right) - f_{t-i}\left(\boldsymbol{\lambda}_{t+1-i}, \boldsymbol{\beta}_{t-i}^*(\boldsymbol{\lambda}_{t+1-i})\right) \right)$$

Which is equivalent to

$$\sum_{t=1}^T \frac{1}{w} \sum_{i=0}^{w-1} \left( f_{t-i}\left(\boldsymbol{\lambda}_{t-i}, \boldsymbol{\beta}_{t-i}^*(\boldsymbol{\lambda}_{t-i})\right) - f_{t-i}\left(\boldsymbol{\lambda}_{t+1-i}, \boldsymbol{\beta}_{t-i}^*(\boldsymbol{\lambda}_{t+1-i})\right) \right)$$

$$= \sum_{t=1}^T \frac{1}{w} \sum_{i=0}^{w-1} \left( f_{t-i}\left(\boldsymbol{\lambda}_{t-i}, \boldsymbol{\beta}_{t-i}^*(\boldsymbol{\lambda}_{t-i})\right) - f_{t+1-i}\left(\boldsymbol{\lambda}_{t+1-i}, \boldsymbol{\beta}_{t+1-i}^*(\boldsymbol{\lambda}_{t+1-i})\right) \right) \quad (63)$$

$$+ \sum_{t=1}^T \frac{1}{w} \sum_{i=0}^{w-1} \left( f_{t+1-i}\left(\boldsymbol{\lambda}_{t+1-i}, \boldsymbol{\beta}_{t+1-i}^*(\boldsymbol{\lambda}_{t+1-i})\right) - f_{t-i}\left(\boldsymbol{\lambda}_{t+1-i}, \boldsymbol{\beta}_{t-i}^*(\boldsymbol{\lambda}_{t+1-i})\right) \right) \quad (64)$$

For equation 63, we can write

$$\frac{1}{w} \sum_{i=0}^{w-1} \left( f_{t-i} \left( \boldsymbol{\lambda}_{t-i}, \boldsymbol{\beta}_{t-i}^*(\boldsymbol{\lambda}_{t-i}) \right) - f_{t+1-i} \left( \boldsymbol{\lambda}_{t+1-i}, \boldsymbol{\beta}_{t+1-i}^*(\boldsymbol{\lambda}_{t+1-i}) \right) \right)$$

$$= \frac{1}{w} \left[ f_t \left( \boldsymbol{\lambda}_t, \boldsymbol{\beta}_t^*(\boldsymbol{\lambda}_t) \right) + \ldots + f_{t+1-w} \left( \boldsymbol{\lambda}_{t+1-w}, \boldsymbol{\beta}_{t+1-w}^*(\boldsymbol{\lambda}_{t+1-w}) \right) \right]$$

$$- \frac{1}{w} \left[ f_{t+1} \left( \boldsymbol{\lambda}_{t+1}, \boldsymbol{\beta}_{t+1}^*(\boldsymbol{\lambda}_{t+1}) \right) + \ldots + f_{t+2-w} \left( \boldsymbol{\lambda}_{t+2-w}, \boldsymbol{\beta}_{t+2-w}^*(\boldsymbol{\lambda}_{t+2-w}) \right) \right]$$

$$= \frac{1}{w} \left[ f_{t+1-w} \left( \boldsymbol{\lambda}_{t+1-w}, \boldsymbol{\beta}_{t+1-w}^*(\boldsymbol{\lambda}_{t+1-w}) \right) - f_{t+1} \left( \boldsymbol{\lambda}_{t+1}, \boldsymbol{\beta}_{t+1}^*(\boldsymbol{\lambda}_{t+1}) \right) \right]$$

$$= \frac{1}{w} \left( F_{t+1-w}(\boldsymbol{\lambda}_{t+1-w}) - F_{t+1}(\boldsymbol{\lambda}_{t+1}) \right) \leq \frac{2Q}{w}, \qquad (65)$$

where the last inequality comes from Assumption E. Note equation 64 can be bounded through

$$\sum_{t=1}^{T} \frac{1}{w} \sum_{i=0}^{w-1} \left( f_{t+1-i} \left( \boldsymbol{\lambda}_{t+1-i}, \boldsymbol{\beta}_{t+1-i}^*(\boldsymbol{\lambda}_{t+1-i}) \right) - f_{t-i} \left( \boldsymbol{\lambda}_{t+1-i}, \boldsymbol{\beta}_{t-i}^*(\boldsymbol{\lambda}_{t+1-i}) \right) \right)$$

$$\leq \sum_{t=1}^{T} \frac{1}{w} \sum_{i=0}^{w-1} \sup_{\boldsymbol{\lambda} \in \mathcal{X}} \left[ f_{t+1-i} \left( \boldsymbol{\lambda}, \boldsymbol{\beta}_{t+1-i}^*(\boldsymbol{\lambda}) \right) - f_{t-i} \left( \boldsymbol{\lambda}, \boldsymbol{\beta}_{t-i}^*(\boldsymbol{\lambda}) \right) \right] \leq V_{1,T} \qquad (66)$$

Combining equation 65 and equation 66 results in the upper bound of

$$\sum_{t=1}^{T} \left( F_{t,w}(\boldsymbol{\lambda}_t) - F_{t,w}(\boldsymbol{\lambda}_{t+1}) \right) \leq \frac{2TQ}{w} + V_{1,T}.$$

$\square$

The next Lemma provides an upper bound on the error of $\|\boldsymbol{\beta}_t - \boldsymbol{\beta}_t^*(\boldsymbol{\lambda}_t)\|^2$ for all $t \in [1, T]$ in terms of an initial error, the cumulative differences of the outer level variable, and the cumulative differences of the optimal inner level variables.

**Lemma 21.** *Suppose Assumptions A and B. Choose the inner step size of $\eta$ and inner iteration count of $K$ to satisfy*

$$\eta < \min \left( \frac{1}{\ell_{g,1}}, \frac{1}{\mu_g} \right), \; and \quad K \geq 1,$$

*and define the decay parameter $\nu$, inner level variable error constant $C_{\mu_g}$, and initial error $\Delta_{\boldsymbol{\beta}}$ respectively as*

$$\nu := \left( 1 - \frac{\eta \mu_g}{2} \right) (1 - \eta \mu_g)^{K-1}, \; and \quad C_{\mu_g} := \left( 1 + \frac{2}{\eta \mu_g} \right),$$

$$and \quad \Delta_{\boldsymbol{\beta}} := \|\boldsymbol{\beta}_1 - \boldsymbol{\beta}_1^*(\boldsymbol{\lambda}_1)\|^2.$$

*Then our Algorithm 4 guarantees $\forall t \in [1, T]$*

$$\|\boldsymbol{\beta}_t - \boldsymbol{\beta}_t^*(\boldsymbol{\lambda}_t)\|^2 \leq \nu^{t-1} \Delta_{\boldsymbol{\beta}}$$

$$+ 2 C_{\mu_g} \kappa_g^2 \sum_{j=0}^{t-2} \nu^j \|\boldsymbol{\lambda}_{t-1-j} - \boldsymbol{\lambda}_{t-j}\|^2 + 2 C_{\mu_g} \sum_{j=0}^{t-2} \nu^j \left\| \boldsymbol{\beta}_{t-j}^*(\boldsymbol{\lambda}_{t-1-j}) - \boldsymbol{\beta}_{t-1-j}^*(\boldsymbol{\lambda}_{t-1-j}) \right\|^2. \quad (67)$$

*Proof.* By definition for $t = 1$, we have $\|\boldsymbol{\beta}_1 - \boldsymbol{\beta}_1^*(\boldsymbol{\lambda}_1)\|^2 = \Delta_{\boldsymbol{\beta}}$. Then $\forall t \in [2, T]$

$$\|\boldsymbol{\beta}_t - \boldsymbol{\beta}_t^*(\boldsymbol{\lambda}_t)\|^2 = \left\| \boldsymbol{\beta}_t - \boldsymbol{\beta}_{t-1}^*(\boldsymbol{\lambda}_{t-1}) + \boldsymbol{\beta}_{t-1}^*(\boldsymbol{\lambda}_{t-1}) - \boldsymbol{\beta}_t^*(\boldsymbol{\lambda}_t) \right\|^2, \qquad (68)$$

which can be expanded based on the Young's Inequality for any $\delta > 0$ as

$$\left\| \boldsymbol{\beta}_t - \boldsymbol{\beta}_{t-1}^*(\boldsymbol{\lambda}_{t-1}) + \boldsymbol{\beta}_{t-1}^*(\boldsymbol{\lambda}_{t-1}) - \boldsymbol{\beta}_t^*(\boldsymbol{\lambda}_t) \right\|^2$$

$$\leq (1 + \delta) \left\| \boldsymbol{\beta}_t - \boldsymbol{\beta}_{t-1}^*(\boldsymbol{\lambda}_{t-1}) \right\|^2$$

$$+ \left( 1 + \frac{1}{\delta} \right) \left\| \boldsymbol{\beta}_{t-1}^*(\boldsymbol{\lambda}_{t-1}) - \boldsymbol{\beta}_t^*(\boldsymbol{\lambda}_t) \right\|^2.$$

Now it holds that

$$\left\| \boldsymbol{\beta}_{t-1}^*(\boldsymbol{\lambda}_{t-1}) - \boldsymbol{\beta}_t^*(\boldsymbol{\lambda}_t) \right\|^2 \leq 2 \left\| \boldsymbol{\beta}_t^*(\boldsymbol{\lambda}_{t-1}) - \boldsymbol{\beta}_t^*(\boldsymbol{\lambda}_t) \right\|^2 + 2 \left\| \boldsymbol{\beta}_t^*(\boldsymbol{\lambda}_{t-1}) - \boldsymbol{\beta}_{t-1}^*(\boldsymbol{\lambda}_{t-1}) \right\|^2$$

which through Lemma 11 can be further upper bounded with the Lipschitz constant of $\kappa_g$ as

$$\left\| \boldsymbol{\beta}_{t-1}^*(\boldsymbol{\lambda}_{t-1}) - \boldsymbol{\beta}_t^*(\boldsymbol{\lambda}_t) \right\|^2 \leq 2\kappa_g^2 \left\| \boldsymbol{\lambda}_{t-1} - \boldsymbol{\lambda}_t \right\|^2 + 2 \left\| \boldsymbol{\beta}_t^*(\boldsymbol{\lambda}_{t-1}) - \boldsymbol{\beta}_{t-1}^*(\boldsymbol{\lambda}_{t-1}) \right\|^2$$

Combining above, we see that $\forall \delta > 0$, equation 68 is upper bounded as

$$\left\| \boldsymbol{\beta}_t - \boldsymbol{\beta}_t^*(\boldsymbol{\lambda}_t) \right\|^2 \leq (1 + \delta) \left\| \boldsymbol{\beta}_t - \boldsymbol{\beta}_{t-1}^*(\boldsymbol{\lambda}_{t-1}) \right\|^2$$

$$+ 2\left(1 + \frac{1}{\delta}\right) \kappa_g^2 \left\| \boldsymbol{\lambda}_{t-1} - \boldsymbol{\lambda}_t \right\|^2 + 2\left(1 + \frac{1}{\delta}\right) \left\| \boldsymbol{\beta}_t^*(\boldsymbol{\lambda}_{t-1}) - \boldsymbol{\beta}_{t-1}^*(\boldsymbol{\lambda}_{t-1}) \right\|^2. \tag{69}$$

As $\eta < \frac{1}{\ell_{g,1}}$, we apply Lemma 8 to see

$$(1 + \delta) \left\| \boldsymbol{\beta}_t - \boldsymbol{\beta}_{t-1}^*(\boldsymbol{\lambda}_{t-1}) \right\|^2 \leq (1 + \delta)(1 - \eta\mu_g)^K \left\| \boldsymbol{\beta}_{t-1} - \boldsymbol{\beta}_{t-1}^*(\boldsymbol{\lambda}_{t-1}) \right\|^2$$

Now setting $\delta = \frac{\eta\mu_g}{2} > 0$ implies that

$$(1 + \delta)(1 - \eta\mu_g)^K = (1 + \frac{\eta\mu_g}{2})(1 - \eta\mu_g)^K < \left(1 - \frac{\eta\mu_g}{2}\right)(1 - \eta\mu_g)^{K-1} < 1$$

Using $\nu := \left(1 - \frac{\eta\mu_g}{2}\right)(1 - \eta\mu_g)^{K-1}$ in equation 69, we get

$$\nu \left\| \boldsymbol{\beta}_t - \boldsymbol{\beta}_t^*(\boldsymbol{\lambda}_t) \right\|^2 \leq \nu^2 \left\| \boldsymbol{\beta}_{t-1} - \boldsymbol{\beta}_{t-1}^*(\boldsymbol{\lambda}_{t-1}) \right\|^2$$

$$+ 2C_{\mu_g}\nu\kappa_g^2 \left\| \boldsymbol{\lambda}_{t-1} - \boldsymbol{\lambda}_t \right\|^2 + 2C_{\mu_g}\nu \left\| \boldsymbol{\beta}_t^*(\boldsymbol{\lambda}_{t-1}) - \boldsymbol{\beta}_{t-1}^*(\boldsymbol{\lambda}_{t-1}) \right\|^2,$$

where $C_{\mu_g} = \left(1 + \frac{2}{\eta\mu_g}\right)$. Starting at $t = T$ and unrolling backward to $t = 1$, results in the upper bound of

$$\left\| \boldsymbol{\beta}_t - \boldsymbol{\beta}_t^*(\boldsymbol{\lambda}_t) \right\|^2 \leq \nu^{t-1}\Delta_{\boldsymbol{\beta}}$$

$$+ 2C_{\mu_g}\kappa_g^2 \sum_{j=0}^{t-2} \nu^j \left\| \boldsymbol{\lambda}_{t-1-j} - \boldsymbol{\lambda}_{t-j} \right\|^2 + 2C_{\mu_g} \sum_{j=0}^{t-2} \nu^j \left\| \boldsymbol{\beta}_{t-j}^*(\boldsymbol{\lambda}_{t-1-j}) - \boldsymbol{\beta}_{t-1-j}^*(\boldsymbol{\lambda}_{t-1-j}) \right\|^2.$$

$$\square$$

The next Lemma utilizes Lemma 19 and Lemma 21 to derive an upper bound on the hypergradient error $\forall t \in [1, T]$ in terms of discounted variations of the (i) cumulative time-smoothed hypergradient error; (ii) bilevel local regret; and (iii) cumulative difference between optimal inner-level variables. A final term is included, composed of a discounted initial error and smoothness term of the inner objective.

**Lemma 22.** *Suppose Assumptions A, B, D, and F. Choose the inner step size of $\eta$ and inner iteration count of $K$ to satisfy*

$$\eta < \min\left(\frac{1}{\ell_{g,1}}, \frac{1}{\mu_g}\right), \text{ and } K \geq 1.$$

*Using the definitions of $\nu$, $C_{\mu_g}$, and $\Delta_{\boldsymbol{\beta}}$ from Lemma 21 as well as the further definition of*

$$L_{\boldsymbol{\beta}} := L_1^2(1 - \eta\mu_g)^K + L_2^2(1 - \eta\mu_g)^{K-1},$$

*then the hypergradient error from our OBBO algorithm in Algorithm 4 is bounded $\forall t \in [1, T]$ as*

$$\left\| \frac{\partial f_t(\boldsymbol{\lambda}_t, \boldsymbol{\omega}_t^K)}{\partial \boldsymbol{\lambda}} - \nabla F_t(\boldsymbol{\lambda}_t) \right\|^2 \leq \delta_t + A \sum_{j=0}^{t-2} \nu^j \left\| \frac{\partial f_{t-1-j,w}(\boldsymbol{\lambda}_{t-1-j}, \boldsymbol{\omega}_{t-1-j}^K)}{\partial \boldsymbol{\lambda}} - \nabla F_{t-1-j,w}(\boldsymbol{\lambda}_{t-1-j}) \right\|^2$$

$$+ B \sum_{j=0}^{t-2} \nu^j \left\| \mathcal{G}_{\mathcal{X}}\left(\boldsymbol{\lambda}_{t-1-j}, \nabla F_{t-1-j,w}(\boldsymbol{\lambda}_{t-1-j}), \alpha\right) \right\|^2 + C \sum_{j=0}^{t-2} \nu^j \left\| \boldsymbol{\beta}_{t-j}^*(\boldsymbol{\lambda}_{t-1-j}) - \boldsymbol{\beta}_{t-1-j}^*(\boldsymbol{\lambda}_{t-1-j}) \right\|^2,$$

$$\tag{70}$$

*where $\delta_t = 3L_3^2(1 - \eta\mu_g)^{2K} + 3L_{\boldsymbol{\beta}}\nu^{t-1}\Delta_{\boldsymbol{\beta}}$ and $A = \frac{12\alpha^2 C_{\mu_g} L_{\boldsymbol{\beta}} \kappa_g^2}{\rho^2}$, $B = 12\alpha^2 C_{\mu_g} L_{\boldsymbol{\beta}} \kappa_g^2$, and $C = 6L_{\boldsymbol{\beta}} C_{\mu_g}$.*

*Proof.* Note that Lemma 19 implies $\forall t \in [1, T]$

$$\left\| \frac{\partial f_t(\boldsymbol{\lambda}_t, \boldsymbol{\omega}_t^K)}{\partial \boldsymbol{\lambda}} - \nabla F_t(\boldsymbol{\lambda}_t) \right\|^2 \le 3L_{\boldsymbol{\beta}} \left\| \boldsymbol{\beta}_t - \boldsymbol{\beta}_t^*(\boldsymbol{\lambda}_t) \right\|^2 + 3L_3^2 (1 - \eta\mu_g)^{2K}.$$

Substituting the upper bound on $\left\| \boldsymbol{\beta}_t - \boldsymbol{\beta}_t^*(\boldsymbol{\lambda}_t) \right\|^2$ from Lemma 21, we have

$$\left\| \frac{\partial f_t(\boldsymbol{\lambda}_t, \boldsymbol{\omega}_t^K)}{\partial \boldsymbol{\lambda}} - \nabla F_t(\boldsymbol{\lambda}_t) \right\|^2 \le 3L_3^2 (1 - \eta\mu_g)^{2K}$$

$$+ 3L_{\boldsymbol{\beta}} \left( \nu^{t-1} \Delta_{\boldsymbol{\beta}} + 2C_{\mu_g} \kappa_g^2 \sum_{j=0}^{t-2} \nu^j \left\| \boldsymbol{\lambda}_{t-1-j} - \boldsymbol{\lambda}_{t-j} \right\|^2 \right)$$

$$+ 6L_{\boldsymbol{\beta}} C_{\mu_g} \sum_{j=0}^{t-2} \nu^j \left\| \boldsymbol{\beta}_{t-j}^*(\boldsymbol{\lambda}_{t-1-j}) - \boldsymbol{\beta}_{t-1-j}^*(\boldsymbol{\lambda}_{t-1-j}) \right\|^2,$$

By definition, we have $\mathcal{G}_{\mathcal{X}} \left( \boldsymbol{\lambda}_{t-1-j}, \frac{\partial f_{t-1-j,w}(\boldsymbol{\lambda}_{t-1-j}, \boldsymbol{\omega}_{t-1-j}^K)}{\partial \boldsymbol{\lambda}}, \alpha \right) := \frac{1}{\alpha} \left( \boldsymbol{\lambda}_{t-1-j} - \boldsymbol{\lambda}_{t-j} \right)$

$$\sum_{j=0}^{t-2} \nu^j \left\| \boldsymbol{\lambda}_{t-1-j} - \boldsymbol{\lambda}_{t-j} \right\|^2 = \alpha^2 \sum_{j=0}^{t-2} \nu^j \left\| \mathcal{G}_{\mathcal{X}} \left( \boldsymbol{\lambda}_{t-1-j}, \frac{\partial f_{t-1-j,w}(\boldsymbol{\lambda}_{t-1-j}, \boldsymbol{\omega}_{t-1-j}^K)}{\partial \boldsymbol{\lambda}}, \alpha \right) \right\|^2$$

$$\le 2\alpha^2 \sum_{j=0}^{t-2} \nu^j \left( \left\| \mathcal{G}_{\mathcal{X}}(\boldsymbol{\lambda}_{t-1-j}, \nabla F_{t-1-j,w}(\boldsymbol{\lambda}_{t-1-j}), \alpha) \right\|^2 \right)$$

$$+ 2\alpha^2 \sum_{j=0}^{t-2} \nu^j \left( \left\| \mathcal{G}_{\mathcal{X}} \left( \boldsymbol{\lambda}_{t-1-j}, \frac{\partial f_{t-1-j,w}(\boldsymbol{\lambda}_{t-1-j}, \boldsymbol{\omega}_{t-1-j}^K)}{\partial \boldsymbol{\lambda}}, \alpha \right) - \mathcal{G}_{\mathcal{X}}(\boldsymbol{\lambda}_{t-1-j}, \nabla F_{t-1-j,w}(\boldsymbol{\lambda}_{t-1-j}), \alpha) \right\|^2 \right)$$

$$\le 2\alpha^2 \sum_{j=0}^{t-2} \nu^j \left( \left\| \mathcal{G}_{\mathcal{X}}(\boldsymbol{\lambda}_{t-1-j}, \nabla F_{t-1-j,w}(\boldsymbol{\lambda}_{t-1-j}), \alpha) \right\|^2 \right)$$

$$+ 2\alpha^2 \sum_{j=0}^{t-2} \nu^j \left( \frac{1}{\rho^2} \left\| \frac{\partial f_{t-1-j,w}(\boldsymbol{\lambda}_{t-1-j}, \boldsymbol{\omega}_{t-1-j}^K)}{\partial \boldsymbol{\lambda}} - \nabla F_{t-1-j,w}(\boldsymbol{\lambda}_{t-1-j}) \right\|^2 \right)$$

$$(71)$$

such that the last inequality comes from Lemma 10. Rearranging terms, we have decomposed the hypergradient error term at $t$ in terms of the cumulative hypergradient error from $j = 1, \ldots, t-1$

$$\left\| \frac{\partial f_t(\boldsymbol{\lambda}_t, \boldsymbol{\omega}_t^K)}{\partial \boldsymbol{\lambda}} - \nabla F_t(\boldsymbol{\lambda}_t) \right\|^2 \le 3L_3^2 (1 - \eta\mu_g)^{2K} + 3L_{\boldsymbol{\beta}} \nu^{t-1} \Delta_{\boldsymbol{\beta}}$$

$$+ 12\alpha^2 C_{\mu_g} L_{\boldsymbol{\beta}} \kappa_g^2 \sum_{j=0}^{t-2} \nu^j \left\| \mathcal{G}_{\mathcal{X}}(\boldsymbol{\lambda}_{t-1-j}, \nabla F_{t-1-j,w}(\boldsymbol{\lambda}_{t-1-j}), \alpha) \right\|^2$$

$$+ \frac{12\alpha^2 C_{\mu_g} L_{\boldsymbol{\beta}} \kappa_g^2}{\rho^2} \sum_{j=0}^{t-2} \nu^j \left\| \frac{\partial f_{t-1-j,w}(\boldsymbol{\lambda}_{t-1-j}, \boldsymbol{\omega}_{t-1-j}^K)}{\partial \boldsymbol{\lambda}} - \nabla F_{t-1-j,w}(\boldsymbol{\lambda}_{t-1-j}) \right\|^2$$

$$+ 6L_{\boldsymbol{\beta}} C_{\mu_g} \sum_{j=0}^{t-2} \nu^j \left\| \boldsymbol{\beta}_{t-j}^*(\boldsymbol{\lambda}_{t-1-j}) - \boldsymbol{\beta}_{t-1-j}^*(\boldsymbol{\lambda}_{t-1-j}) \right\|^2,$$

$$\square$$

The next Lemma provides an upper bound on the cumulative time-smoothed hypergradient error using the result of Lemma 22.

**Lemma 23.** *Suppose Assumptions A, B, D, and F. Choose the inner step size of $\eta < \min\left(\frac{1}{\ell_{g,1}}, \frac{1}{\mu_g}\right)$, the outer step size $\alpha \leq \frac{\rho\sqrt{(1-\nu)}}{\kappa_g\sqrt{108 C_{\mu_g} L_{\boldsymbol{\beta}}}}$, and inner iteration count $K = \frac{\log(T)}{\log\left((1-\eta\mu_g)^{-1}\right)} + 1$. Then the cumulative time-smoothed hypergradient error from our OBBO algorithm in Algorithm 4 satisfies*

$$\sum_{t=1}^{T} \left\|\frac{\partial f_{t,w}(\boldsymbol{\lambda}_t, \boldsymbol{\omega}_t^K)}{\partial \boldsymbol{\lambda}} - \nabla F_{t,w}(\boldsymbol{\lambda}_t)\right\|^2 \leq \frac{27}{8}\left(\frac{\Delta_{\boldsymbol{\beta}} L_{\boldsymbol{\beta}}}{(1-\nu)} + L_3^2\right)$$

$$+ A\sum_{t=1}^{T}\left\|\mathcal{G}_{\mathcal{X}}(\boldsymbol{\lambda}_{t-1-j}, \nabla F_{t-1-j,w}(\boldsymbol{\lambda}_{t-1-j}), \alpha)\right\|^2 + B\sum_{t=2}^{T}\left\|\boldsymbol{\beta}_t^*(\boldsymbol{\lambda}_{t-1}) - \boldsymbol{\beta}_{t-1}^*(\boldsymbol{\lambda}_{t-1})\right\|^2,$$

*where $A := \frac{\rho^2}{8}$ and $B := \frac{27 L_{\boldsymbol{\beta}} C_{\mu_g}}{2(1-\nu)}$.*

*Proof.* Note by definition of the time-smoothed outer level objective and application of Young's inequality we have

$$\left\|\frac{\partial f_{t,w}(\boldsymbol{\lambda}_t, \boldsymbol{\omega}_t^K)}{\partial \boldsymbol{\lambda}} - \nabla F_{t,w}(\boldsymbol{\lambda}_t)\right\|^2 = \left\|\frac{1}{w}\sum_{i=0}^{w-1}\left[\frac{\partial f_{t-i}(\boldsymbol{\lambda}_{t-i}, \boldsymbol{\omega}_{t-i}^K)}{\partial \boldsymbol{\lambda}} - \nabla F_{t-i}(\boldsymbol{\lambda}_{t-i})\right]\right\|^2$$

$$= \left[\sum_{i=0}^{w-1}\frac{1}{w}\sum_{j=0}^{w-1}\frac{1}{w}\left\langle\frac{\partial f_{t-i}(\boldsymbol{\lambda}_{t-i}, \boldsymbol{\omega}_{t-i}^K)}{\partial \boldsymbol{\lambda}} - \nabla F_{t-i}(\boldsymbol{\lambda}_{t-i}), \frac{\partial f_{t-j}(\boldsymbol{\lambda}_{t-j}, \boldsymbol{\omega}_{t-j}^K)}{\partial \boldsymbol{\lambda}} - \nabla F_{t-j}(\boldsymbol{\lambda}_{t-j})\right\rangle\right]$$

$$\leq \left[\sum_{i=0}^{w-1}\frac{1}{w}\sum_{j=0}^{w-1}\frac{1}{w}\left(\frac{1}{2}\left\|\frac{\partial f_{t-i}(\boldsymbol{\lambda}_{t-i}, \boldsymbol{\omega}_{t-i}^K)}{\partial \boldsymbol{\lambda}} - \nabla F_{t-i}(\boldsymbol{\lambda}_{t-i})\right\|^2\right.\right.$$

$$\left.\left.+ \frac{1}{2}\left\|\frac{\partial f_{t-j}(\boldsymbol{\lambda}_{t-j}, \boldsymbol{\omega}_{t-j}^K)}{\partial \boldsymbol{\lambda}} - \nabla F_{t-j}(\boldsymbol{\lambda}_{t-j})\right\|^2\right)\right]$$

$$= \frac{1}{w}\sum_{i=0}^{w-1}\left\|\frac{\partial f_{t-i}(\boldsymbol{\lambda}_{t-i}, \boldsymbol{\omega}_{t-i}^K)}{\partial \boldsymbol{\lambda}} - \nabla F_{t-i}(\boldsymbol{\lambda}_{t-i})\right\|^2$$

$$\tag{72}$$

Substituting the upper bound on $\left\|\frac{\partial f_t(\boldsymbol{\lambda}_t, \boldsymbol{\omega}_t^K)}{\partial \boldsymbol{\lambda}} - \nabla F_t(\boldsymbol{\lambda}_t)\right\|^2$ from Lemma 22 and re-indexing the bilevel local regret and the cumulative time-smoothed hypergradient error, we construct the upper bound of

$$\sum_{t=1}^{T}\left\|\frac{\partial f_{t,w}(\boldsymbol{\lambda}_t, \boldsymbol{\omega}_t^K)}{\partial \boldsymbol{\lambda}} - \nabla F_{t,w}(\boldsymbol{\lambda}_t)\right\|^2$$

$$\leq \sum_{t=1}^{T}\frac{1}{w}\left[\sum_{i=0}^{w-1}\left(3L_3^2(1-\eta\mu_g)^{2K} + 3L_{\boldsymbol{\beta}}\nu^{t-i-1}\Delta_{\boldsymbol{\beta}}\right)\right]$$

$$+ \sum_{t=1}^{T}\frac{1}{w}\left[\sum_{i=0}^{w-1}\left(12\alpha^2 C_{\mu_g}L_{\boldsymbol{\beta}}\kappa_g^2\sum_{j=0}^{t-i-2}\nu^j\left\|\mathcal{G}_{\mathcal{X}}(\boldsymbol{\lambda}_{t-i-j}, \nabla F_{t-i-j,w}(\boldsymbol{\lambda}_{t-i-j}), \alpha)\right\|^2\right)\right]$$

$$+ \sum_{t=1}^{T}\frac{1}{w}\left[\sum_{i=0}^{w-1}\left(\frac{12\alpha^2 C_{\mu_g}L_{\boldsymbol{\beta}}\kappa_g^2}{\rho^2}\sum_{j=0}^{t-i-2}\nu^j\left\|\frac{\partial f_{t-i-j,w}(\boldsymbol{\lambda}_{t-i-j}, \boldsymbol{\omega}_{t-i-j}^K)}{\partial \boldsymbol{\lambda}} - \nabla F_{t-i-j,w}(\boldsymbol{\lambda}_{t-i-j})\right\|^2\right)\right]$$

$$+ \sum_{t=2}^{T}\frac{1}{w}\left[\sum_{i=0}^{w-1}\left(6L_{\boldsymbol{\beta}}C_{\mu_g}\sum_{j=0}^{t-i-2}\nu^j\left\|\boldsymbol{\beta}_{t-i-j}^*(\boldsymbol{\lambda}_{t-i-1-j}) - \boldsymbol{\beta}_{t-i-1-j}^*(\boldsymbol{\lambda}_{t-i-1-j})\right\|^2\right)\right]$$

$$\tag{73}$$

Given $\nu < 1$, it holds that $\sum_{j=0}^{t-2} \nu^j < \sum_{j=0}^{\infty} \nu^j = \frac{1}{1-\nu}$, which lets us upper bound equation 73 as

$$\sum_{t=1}^{T} \left\| \frac{\partial f_{t,w}(\boldsymbol{\lambda}_t, \boldsymbol{\omega}_t^K)}{\partial \boldsymbol{\lambda}} - \nabla F_{t,w}(\boldsymbol{\lambda}_t) \right\|^2$$

$$\leq \sum_{t=1}^{T} \frac{1}{w} \left[ \sum_{i=0}^{w-1} \left( 3L_3^2(1-\eta\mu_g)^{2K} + 3L_{\boldsymbol{\beta}}\nu^{t-i-1}\Delta_{\boldsymbol{\beta}} \right) \right]$$

$$+ \frac{12\alpha^2 C_{\mu_g} L_{\boldsymbol{\beta}} \kappa_g^2}{(1-\nu)} \sum_{t=1}^{T} \frac{1}{w} \left[ \sum_{i=0}^{w-1} \| \mathcal{G}_{\mathcal{X}}(\boldsymbol{\lambda}_{t-i}, \nabla F_{t-i,w}(\boldsymbol{\lambda}_{t-i}), \alpha) \|^2 \right]$$

$$+ \frac{12\alpha^2 C_{\mu_g} L_{\boldsymbol{\beta}} \kappa_g^2}{\rho^2(1-\nu)} \sum_{t=1}^{T} \frac{1}{w} \left[ \sum_{i=0}^{w-1} \left\| \frac{\partial f_{t-i,w}(\boldsymbol{\lambda}_{t-i}, \boldsymbol{\omega}_{t-i}^K)}{\partial \boldsymbol{\lambda}} - \nabla F_{t-i,w}(\boldsymbol{\lambda}_{t-i}) \right\|^2 \right]$$

$$+ \frac{6L_{\boldsymbol{\beta}} C_{\mu_g}}{(1-\nu)} \sum_{t=2}^{T} \frac{1}{w} \left[ \sum_{i=0}^{w-1} \| \boldsymbol{\beta}_{t-i}^*(\boldsymbol{\lambda}_{t-i-1}) - \boldsymbol{\beta}_{t-i-1}^*(\boldsymbol{\lambda}_{t-i-1}) \|^2 \right].$$

and further

$$\sum_{t=1}^{T} \left\| \frac{\partial f_{t,w}(\boldsymbol{\lambda}_t, \boldsymbol{\omega}_t^K)}{\partial \boldsymbol{\lambda}} - \nabla F_{t,w}(\boldsymbol{\lambda}_t) \right\|^2$$

$$\leq \sum_{t=1}^{T} \left( 3L_3^2(1-\eta\mu_g)^{2K} + 3L_{\boldsymbol{\beta}}\nu^{t-1}\Delta_{\boldsymbol{\beta}} \right) + \frac{12\alpha^2 C_{\mu_g} L_{\boldsymbol{\beta}} \kappa_g^2}{(1-\nu)} \sum_{t=1}^{T} \| \mathcal{G}_{\mathcal{X}}(\boldsymbol{\lambda}_t, \nabla F_{t,w}(\boldsymbol{\lambda}_t), \alpha) \|^2$$

$$+ \frac{12\alpha^2 C_{\mu_g} L_{\boldsymbol{\beta}} \kappa_g^2}{\rho^2(1-\nu)} \sum_{t=1}^{T} \left\| \frac{\partial f_{t,w}(\boldsymbol{\lambda}_t, \boldsymbol{\omega}_t^K)}{\partial \boldsymbol{\lambda}} - \nabla F_{t,w}(\boldsymbol{\lambda}_t) \right\|^2 + \frac{6L_{\boldsymbol{\beta}} C_{\mu_g}}{(1-\nu)} \sum_{t=2}^{T} \| \boldsymbol{\beta}_t^*(\boldsymbol{\lambda}_{t-1}) - \boldsymbol{\beta}_{t-1}^*(\boldsymbol{\lambda}_{t-1}) \|^2$$

which implies that

$$\left( 1 - \frac{12\alpha^2 C_{\mu_g} L_{\boldsymbol{\beta}} \kappa_g^2}{\rho^2(1-\nu)} \right) \sum_{t=1}^{T} \left\| \frac{\partial f_{t,w}(\boldsymbol{\lambda}_t, \boldsymbol{\omega}_t^K)}{\partial \boldsymbol{\lambda}} - \nabla F_{t,w}(\boldsymbol{\lambda}_t) \right\|^2$$

$$\leq \frac{3\Delta_{\boldsymbol{\beta}} L_{\boldsymbol{\beta}}}{1-\nu} + \sum_{t=1}^{T} \left( 3L_3^2(1-\eta\mu_g)^{2K} \right) + \frac{12\alpha^2 C_{\mu_g} L_{\boldsymbol{\beta}} \kappa_g^2}{(1-\nu)} \sum_{t=1}^{T} \| \mathcal{G}_{\mathcal{X}}(\boldsymbol{\lambda}_t, \nabla F_{t,w}(\boldsymbol{\lambda}_t), \alpha) \|^2$$

$$+ \frac{6L_{\boldsymbol{\beta}} C_{\mu_g}}{(1-\nu)} \sum_{t=2}^{T} \| \boldsymbol{\beta}_t^*(\boldsymbol{\lambda}_{t-1}) - \boldsymbol{\beta}_{t-1}^*(\boldsymbol{\lambda}_{t-1}) \|^2,$$

Setting $K = \log(T) / \log\left((1-\eta\mu_g)^{-1}\right) + 1$ and $0 < \alpha \leq \frac{\rho\sqrt{(1-\nu)}}{\kappa_g\sqrt{108 C_{\mu_g} L_{\boldsymbol{\beta}}}}$

$$\left( 1 - \frac{12\alpha^2 C_{\mu_g} L_{\boldsymbol{\beta}} \kappa_g^2}{\rho^2(1-\nu)} \right) \geq \frac{8}{9}$$

implies the upper bound of

$$\sum_{t=1}^{T} \left\| \frac{\partial f_{t,w}(\boldsymbol{\lambda}_t, \boldsymbol{\omega}_t^K)}{\partial \boldsymbol{\lambda}} - \nabla F_{t,w}(\boldsymbol{\lambda}_t) \right\|^2 \leq \frac{27}{8} \left( \frac{\Delta_{\boldsymbol{\beta}} L_{\boldsymbol{\beta}}}{(1-\nu)} + L_3^2 \right)$$

$$+ \frac{\rho^2}{8} \sum_{t=1}^{T} \| \mathcal{G}_{\mathcal{X}}(\boldsymbol{\lambda}_t, \nabla F_{t,w}(\boldsymbol{\lambda}_t), \alpha) \|^2 + \frac{27 L_{\boldsymbol{\beta}} C_{\mu_g}}{2(1-\nu)} \sum_{t=2}^{T} \| \boldsymbol{\beta}_t^*(\boldsymbol{\lambda}_{t-1}) - \boldsymbol{\beta}_{t-1}^*(\boldsymbol{\lambda}_{t-1}) \|^2,$$

$\square$

The next theorem presents the theoretical contribution for Algorithm 4. For suitably chosen step sizes, the sequence of iterates $\{\boldsymbol{\lambda}_t\}_{t=1}^T$ achieves sublinear bilevel local regret.

**Theorem 24.** *Suppose Assumptions A, B, D, E, F. Choose the inner step size of $\eta < \min\left(\frac{1}{\ell_{g,1}}, \frac{1}{\mu_g}\right)$, the outer step size of $\alpha \leq \min\left\{\frac{3\rho}{4\ell_{F,1}}, \frac{\rho\sqrt{(1-\nu)}}{\kappa_g\sqrt{108C_{\mu_g}L_\beta}}\right\}$, and inner iteration count $K = \frac{\log(T)}{\log((1-\eta\mu_g)^{-1})} + 1$. Then the bilevel local regret of Algorithm 4 satisfies the bound of*

$$BLR_w(T) := \sum_{t=1}^{T} \left\|\mathcal{G}_\mathcal{X}(\boldsymbol{\lambda}_t, \nabla F_{t,w}(\boldsymbol{\lambda}_t), \alpha)\right\|^2 \leq O\left(\frac{T}{w} + V_{1,T} + \kappa_g^2 H_{2,T}\right), \tag{74}$$

*Proof.* Note, Lemma H.2 of Lin et al. (2024) shows how with Assumption A, we have

$$F_{t,w}(\boldsymbol{\lambda}_{t+1}) - F_{t,w}(\boldsymbol{\lambda}_t) \leq \langle \nabla F_{t,w}(\boldsymbol{\lambda}_t), \boldsymbol{\lambda}_{t+1} - \boldsymbol{\lambda}_t \rangle + \frac{\ell_{F,1}}{2}\|\boldsymbol{\lambda}_{t+1} - \boldsymbol{\lambda}_t\|^2.$$

then by substituting the step of $\mathcal{G}_\mathcal{X}\left(\boldsymbol{\lambda}_t, \frac{\partial f_{t,w}(\boldsymbol{\lambda}_t, \boldsymbol{\omega}_t^K)}{\partial \boldsymbol{\lambda}}, \alpha\right) := \frac{1}{\alpha}(\boldsymbol{\lambda}_t - \boldsymbol{\lambda}_{t+1})$,

$$F_{t,w}(\boldsymbol{\lambda}_{t+1}) - F_{t,w}(\boldsymbol{\lambda}_t) \leq \langle \nabla F_{t,w}(\boldsymbol{\lambda}_t), \boldsymbol{\lambda}_{t+1} - \boldsymbol{\lambda}_t \rangle + \frac{\ell_{F,1}}{2}\|\boldsymbol{\lambda}_{t+1} - \boldsymbol{\lambda}_t\|^2$$

$$= -\alpha\left\langle \nabla F_{t,w}(\boldsymbol{\lambda}_t), \mathcal{G}_\mathcal{X}\left(\boldsymbol{\lambda}_t, \frac{\partial f_{t,w}(\boldsymbol{\lambda}_t, \boldsymbol{\omega}_t^K)}{\partial \boldsymbol{\lambda}}, \alpha\right)\right\rangle + \frac{\alpha^2 \ell_{F,1}}{2}\left\|\mathcal{G}_\mathcal{X}\left(\boldsymbol{\lambda}_t, \frac{\partial f_{t,w}(\boldsymbol{\lambda}_t, \boldsymbol{\omega}_t^K)}{\partial \boldsymbol{\lambda}}, \alpha\right)\right\|^2,$$

$$= -\alpha\left\langle \frac{\partial f_{t,w}(\boldsymbol{\lambda}_t, \boldsymbol{\omega}_t^K)}{\partial \boldsymbol{\lambda}}, \mathcal{G}_\mathcal{X}\left(\boldsymbol{\lambda}_t, \frac{\partial f_{t,w}(\boldsymbol{\lambda}_t, \boldsymbol{\omega}_t^K)}{\partial \boldsymbol{\lambda}}, \alpha\right)\right\rangle$$

$$+\alpha\left\langle \frac{\partial f_{t,w}(\boldsymbol{\lambda}_t, \boldsymbol{\omega}_t^K)}{\partial \boldsymbol{\lambda}} - \nabla F_{t,w}(\boldsymbol{\lambda}_t), \mathcal{G}_\mathcal{X}\left(\boldsymbol{\lambda}_t, \frac{\partial f_{t,w}(\boldsymbol{\lambda}_t, \boldsymbol{\omega}_t^K)}{\partial \boldsymbol{\lambda}}, \alpha\right)\right\rangle$$

$$+\frac{\alpha^2 \ell_{F,1}}{2}\left\|\mathcal{G}_\mathcal{X}\left(\boldsymbol{\lambda}_t, \frac{\partial f_{t,w}(\boldsymbol{\lambda}_t, \boldsymbol{\omega}_t^K)}{\partial \boldsymbol{\lambda}}, \alpha\right)\right\|^2. \tag{75}$$

Using Lemma 9 with $\boldsymbol{q} = \frac{\partial f_{t,w}(\boldsymbol{\lambda}_t, \boldsymbol{\omega}_t^K)}{\partial \boldsymbol{\lambda}}$, note that

$$\alpha\left\langle \frac{\partial f_{t,w}(\boldsymbol{\lambda}_t, \boldsymbol{\omega}_t^K)}{\partial \boldsymbol{\lambda}}, \mathcal{G}_\mathcal{X}\left(\boldsymbol{\lambda}_t, \frac{\partial f_{t,w}(\boldsymbol{\lambda}_t, \boldsymbol{\omega}_t^K)}{\partial \boldsymbol{\lambda}}, \alpha\right)\right\rangle$$

$$\geq \alpha\rho\left\|\mathcal{G}_\mathcal{X}\left(\boldsymbol{\lambda}_t, \frac{\partial f_{t,w}(\boldsymbol{\lambda}_t, \boldsymbol{\omega}_t^K)}{\partial \boldsymbol{\lambda}}, \alpha\right)\right\|^2 + h(\boldsymbol{\lambda}_{t+1}) - h(\boldsymbol{\lambda}_t) \tag{76}$$

and further we get the following based on a variation of Young's Inequality

$$\left\langle \frac{\partial f_{t,w}(\boldsymbol{\lambda}_t, \boldsymbol{\omega}_t^K)}{\partial \boldsymbol{\lambda}} - \nabla F_{t,w}(\boldsymbol{\lambda}_t), \mathcal{G}_\mathcal{X}\left(\boldsymbol{\lambda}_t, \frac{\partial f_{t,w}(\boldsymbol{\lambda}_t, \boldsymbol{\omega}_t^K)}{\partial \boldsymbol{\lambda}}, \alpha\right)\right\rangle$$

$$\leq \frac{1}{\rho}\left\|\frac{\partial f_{t,w}(\boldsymbol{\lambda}_t, \boldsymbol{\omega}_t^K)}{\partial \boldsymbol{\lambda}} - \nabla F_{t,w}(\boldsymbol{\lambda}_t)\right\|^2 + \frac{\rho}{4}\left\|\mathcal{G}_\mathcal{X}\left(\boldsymbol{\lambda}_t, \frac{\partial f_{t,w}(\boldsymbol{\lambda}_t, \boldsymbol{\omega}_t^K)}{\partial \boldsymbol{\lambda}}, \alpha\right)\right\|^2 \tag{77}$$

Using equation 76 and equation 77 in equation 75 we get

$$F_{t,w}(\boldsymbol{\lambda}_{t+1}) - F_{t,w}(\boldsymbol{\lambda}_t) \leq \left(\frac{\alpha^2 \ell_{F,1}}{2} - \frac{3\alpha\rho}{4}\right)\left\|\mathcal{G}_\mathcal{X}\left(\boldsymbol{\lambda}_t, \frac{\partial f_{t,w}(\boldsymbol{\lambda}_t, \boldsymbol{\omega}_t^K)}{\partial \boldsymbol{\lambda}}, \alpha\right)\right\|^2$$

$$+\frac{\alpha}{\rho}\left\|\frac{\partial f_{t,w}(\boldsymbol{\lambda}_t, \boldsymbol{\omega}_t^K)}{\partial \boldsymbol{\lambda}} - \nabla F_{t,w}(\boldsymbol{\lambda}_t)\right\|^2 + h(\boldsymbol{\lambda}_t) - h(\boldsymbol{\lambda}_{t+1}) \tag{78}$$

which as $0 < \alpha \leq \frac{3\rho}{4\ell_{F,1}}$ results in the further upper bound of

$$F_{t,w}(\boldsymbol{\lambda}_{t+1}) - F_{t,w}(\boldsymbol{\lambda}_t) \leq -\frac{3\alpha\rho}{8}\left\|\mathcal{G}_\mathcal{X}\left(\boldsymbol{\lambda}_t, \frac{\partial f_{t,w}(\boldsymbol{\lambda}_t, \boldsymbol{\omega}_t^K)}{\partial \boldsymbol{\lambda}}, \alpha\right)\right\|^2$$

$$+\frac{\alpha}{\rho}\left\|\frac{\partial f_{t,w}(\boldsymbol{\lambda}_t, \boldsymbol{\omega}_t^K)}{\partial \boldsymbol{\lambda}} - \nabla F_{t,w}(\boldsymbol{\lambda}_t)\right\|^2 + h(\boldsymbol{\lambda}_t) - h(\boldsymbol{\lambda}_{t+1}) \tag{79}$$

Further note we can upper bound the local regret as

$$
\|\mathcal{G}_{\mathcal{X}}(\boldsymbol{\lambda}_t, \nabla F_{t,w}(\boldsymbol{\lambda}_t), \alpha)\|^2 \leq 2 \left\| \mathcal{G}_{\mathcal{X}}\left(\boldsymbol{\lambda}_t, \frac{\partial f_{t,w}(\boldsymbol{\lambda}_t, \boldsymbol{\omega}_t^K)}{\partial \boldsymbol{\lambda}}, \alpha\right) \right\|^2
$$

$$
+ 2 \left\| \mathcal{G}_{\mathcal{X}}\left(\boldsymbol{\lambda}_t, \frac{\partial f_{t,w}(\boldsymbol{\lambda}_t, \boldsymbol{\omega}_t^K)}{\partial \boldsymbol{\lambda}}, \alpha\right) - \mathcal{G}_{\mathcal{X}}(\boldsymbol{\lambda}_t, \nabla F_{t,w}(\boldsymbol{\lambda}_t), \alpha) \right\|^2
$$

$$
\leq 2 \left\| \mathcal{G}_{\mathcal{X}}\left(\boldsymbol{\lambda}_t, \frac{\partial f_{t,w}(\boldsymbol{\lambda}_t, \boldsymbol{\omega}_t^K)}{\partial \boldsymbol{\lambda}}, \alpha\right) \right\|^2 + \frac{2}{\rho^2} \left\| \frac{\partial f_{t,w}(\boldsymbol{\lambda}_t, \boldsymbol{\omega}_t^K)}{\partial \boldsymbol{\lambda}} - \nabla F_{t,w}(\boldsymbol{\lambda}_t) \right\|^2,
$$

where the last inequality comes from Lemma 10. This then implies that

$$
- \left\| \mathcal{G}_{\mathcal{X}}\left(\boldsymbol{\lambda}_t, \frac{\partial f_{t,w}(\boldsymbol{\lambda}_t, \boldsymbol{\omega}_t^K)}{\partial \boldsymbol{\lambda}}, \alpha\right) \right\|^2 \leq -\frac{1}{2} \|\mathcal{G}_{\mathcal{X}}(\boldsymbol{\lambda}_t, \nabla F_{t,w}(\boldsymbol{\lambda}_t), \alpha)\|^2
$$

$$
+ \frac{1}{\rho^2} \left\| \frac{\partial f_{t,w}(\boldsymbol{\lambda}_t, \boldsymbol{\omega}_t^K)}{\partial \boldsymbol{\lambda}} - \nabla F_{t,w}(\boldsymbol{\lambda}_t) \right\|^2 \tag{80}
$$

Substituting equation 80 into equation 79 gives us

$$
F_{t,w}(\boldsymbol{\lambda}_{t+1}) - F_{t,w}(\boldsymbol{\lambda}_t) \leq -\frac{3\alpha\rho}{16} \|\mathcal{G}_{\mathcal{X}}(\boldsymbol{\lambda}_t, \nabla F_{t,w}(\boldsymbol{\lambda}_t), \alpha)\|^2
$$

$$
+ \left(\frac{\alpha}{\rho} + \frac{3\alpha}{8\rho}\right) \left\| \frac{\partial f_{t,w}(\boldsymbol{\lambda}_t, \boldsymbol{\omega}_t^K)}{\partial \boldsymbol{\lambda}} - \nabla F_{t,w}(\boldsymbol{\lambda}_t) \right\|^2 + h(\boldsymbol{\lambda}_t) - h(\boldsymbol{\lambda}_{t+1}). \tag{81}
$$

Rearranging we see

$$
\frac{3\alpha\rho}{16} \|\mathcal{G}_{\mathcal{X}}(\boldsymbol{\lambda}_t, \nabla F_{t,w}(\boldsymbol{\lambda}_t), \alpha)\|^2 \leq F_{t,w}(\boldsymbol{\lambda}_t) - F_{t,w}(\boldsymbol{\lambda}_{t+1})
$$

$$
+ \frac{11\alpha}{8\rho} \left\| \frac{\partial f_{t,w}(\boldsymbol{\lambda}_t, \boldsymbol{\omega}_t^K)}{\partial \boldsymbol{\lambda}} - \nabla F_{t,w}(\boldsymbol{\lambda}_t) \right\|^2 + h(\boldsymbol{\lambda}_t) - h(\boldsymbol{\lambda}_{t+1}). \tag{82}
$$

Summing from $1, \ldots, T$ and telescoping $h(\boldsymbol{\lambda}_t)$

$$
\frac{3\alpha\rho}{16} \sum_{t=1}^{T} \|\mathcal{G}_{\mathcal{X}}(\boldsymbol{\lambda}_t, \nabla F_{t,w}(\boldsymbol{\lambda}_t), \alpha)\|^2 \leq \sum_{t=1}^{T} (F_{t,w}(\boldsymbol{\lambda}_t) - F_{t,w}(\boldsymbol{\lambda}_{t+1}))
$$

$$
+ \frac{11\alpha}{8\rho} \sum_{t=1}^{T} \left( \left\| \frac{\partial f_{t,w}(\boldsymbol{\lambda}_t, \boldsymbol{\omega}_t^K)}{\partial \boldsymbol{\lambda}} - \nabla F_{t,w}(\boldsymbol{\lambda}_t) \right\|^2 \right) + \Delta_h,
$$

where $\Delta_h := h(\boldsymbol{\lambda}_1) - h(\boldsymbol{\lambda}_{T+1})$ Then we can substitute Lemma 23 to get

$$
\frac{3\alpha\rho}{16} \sum_{t=1}^{T} \|\mathcal{G}_{\mathcal{X}}(\boldsymbol{\lambda}_t, \nabla F_{t,w}(\boldsymbol{\lambda}_t), \alpha)\|^2 \leq \sum_{t=1}^{T} (F_{t,w}(\boldsymbol{\lambda}_t) - F_{t,w}(\boldsymbol{\lambda}_{t+1}))
$$

$$
+ \frac{11\alpha}{8\rho} \left( \frac{27}{8} \left( \frac{\Delta_{\boldsymbol{\beta}} L_{\boldsymbol{\beta}}}{(1-\nu)} + L_3^2 \right) + \frac{\rho^2}{8} \sum_{t=1}^{T} \|\mathcal{G}_{\mathcal{X}}(\boldsymbol{\lambda}_t, \nabla F_{t,w}(\boldsymbol{\lambda}_t), \alpha)\|^2 \right)
$$

$$
+ \frac{11\alpha}{8\rho} \left( \frac{27 L_{\boldsymbol{\beta}} C_{\mu_g}}{2(1-\nu)} \sum_{t=2}^{T} \|\boldsymbol{\beta}_t^*(\boldsymbol{\lambda}_{t-1}) - \boldsymbol{\beta}_{t-1}^*(\boldsymbol{\lambda}_{t-1})\|^2 \right) + \Delta_h.
$$

Rearranging we have

$$
\frac{12\alpha\rho}{64} \sum_{t=1}^{T} \|\mathcal{G}_{\mathcal{X}}(\boldsymbol{\lambda}_t, \nabla F_{t,w}(\boldsymbol{\lambda}_t), \alpha)\|^2 \leq \sum_{t=1}^{T} (F_{t,w}(\boldsymbol{\lambda}_t) - F_{t,w}(\boldsymbol{\lambda}_{t+1}))
$$

$$
+ \frac{11\alpha\rho}{64} \sum_{t=1}^{T} \|\mathcal{G}_{\mathcal{X}}(\boldsymbol{\lambda}_t, \nabla F_{t,w}(\boldsymbol{\lambda}_t), \alpha)\|^2 + \frac{11\alpha}{8\rho} \left( \frac{27}{8} \left( \frac{\Delta_{\boldsymbol{\beta}} L_{\boldsymbol{\beta}}}{(1-\nu)} + L_3^2 \right) \right)
$$

$$
+ \frac{11\alpha}{8\rho} \left( \frac{27 L_{\boldsymbol{\beta}} C_{\mu_g}}{2(1-\nu)} \sum_{t=2}^{T} \|\boldsymbol{\beta}_t^*(\boldsymbol{\lambda}_{t-1}) - \boldsymbol{\beta}_{t-1}^*(\boldsymbol{\lambda}_{t-1})\|^2 \right) + \Delta_h,
$$

or more succinctly

$$\sum_{t=1}^{T} \|\mathcal{G}_{\mathcal{X}}(\boldsymbol{\lambda}_t, \nabla F_{t,w}(\boldsymbol{\lambda}_t), \alpha)\|^2 \leq \frac{64}{\alpha\rho} \sum_{t=1}^{T} (F_{t,w}(\boldsymbol{\lambda}_t) - F_{t,w}(\boldsymbol{\lambda}_{t+1}))$$

$$+ \frac{88}{\rho^2} \left( \frac{27}{8} \left( \frac{\Delta_{\boldsymbol{\beta}} L_{\boldsymbol{\beta}}}{(1-\nu)} + L_3^2 \right) \right) + \frac{88}{\rho^2} \frac{27 L_{\boldsymbol{\beta}} C_{\mu_g}}{2(1-\nu)} \sum_{t=2}^{T} \left\| \boldsymbol{\beta}_t^*(\boldsymbol{\lambda}_{t-1}) - \boldsymbol{\beta}_{t-1}^*(\boldsymbol{\lambda}_{t-1}) \right\|^2 + \frac{64\Delta_h}{\alpha\rho}. \quad (83)$$

Applying Lemma 20 we see

$$\sum_{t=1}^{T} (F_{t,w}(\boldsymbol{\lambda}_t) - F_{t,w}(\boldsymbol{\lambda}_{t+1})) \leq \frac{2TQ}{w} + V_{1,T}, \quad (84)$$

which by using equation 84 in equation 83 we get for $L_{\boldsymbol{\beta}} = O(\kappa_g^2)$

$$\sum_{t=1}^{T} \|\mathcal{G}_{\mathcal{X}}(\boldsymbol{\lambda}_t, \nabla F_{t,w}(\boldsymbol{\lambda}_t), \alpha)\|^2 \leq \frac{64}{\alpha\rho} \left( \frac{2TQ}{w} + V_{1,T} \right) + \frac{297}{\rho^2} \left( \frac{\Delta_{\boldsymbol{\beta}} L_{\boldsymbol{\beta}}}{(1-\nu)} + L_3^2 \right)$$

$$+ \frac{64\Delta_h}{\alpha\rho} + \frac{1188 L_{\boldsymbol{\beta}} C_{\mu_g}}{\rho^2(1-\nu)} H_{2,T}, \quad (85)$$

which dividing by $T$ and recalling we imposed regularity constraints of $H_{2,T} = o(T)$, as well as $V_{1,T} = o(T)$, implies the bilevel local regret of Algorithm 4 is sublinear on the order of

$$BLR_w(T) := \sum_{t=1}^{T} \|\mathcal{G}_{\mathcal{X}}(\boldsymbol{\lambda}_t, \nabla F_{t,w}(\boldsymbol{\lambda}_t), \alpha)\|^2 \leq O\left( \frac{T}{w} + V_{1,T} + \kappa_g^2 H_{2,T} \right). \quad (86)$$

$\square$

# D  ADDITIONAL ALGORITHMS

Algorithm 5 (following Ghadimi & Wang (2018)) forms a stochastic hypergradient by replacing the inverse Hessian–vector product with a randomized Neumann-series approximation: a uniformly sampled truncation level $\widetilde{m}$ yields the product operator $\mathbf{B}_t$, which serves as an unbiased estimator of $(\nabla_{\boldsymbol{\beta\beta}}^2 g_t)^{-1}$. This avoids explicit matrix inversion while retaining correctness in expectation and is standard in scalable bilevel optimization.

---

**Algorithm 5** Stochastic Hypergradient Estimation (Ghadimi & Wang (2018))

---

**Require:** Get $\boldsymbol{\lambda} \in \mathcal{X}$, $\boldsymbol{\beta} \in \mathbb{R}^{d_2}$, sample upper bound $m$, learning rate $\tilde{\eta}$

Sample $\widetilde{m} \sim \mathcal{U}(0, 1, \ldots, m-1)$ and $\mathcal{E} = \{\epsilon, \zeta^0, \ldots, \zeta^{\widetilde{m}-1}\}$

Compute : $\mathbf{g}_t \leftarrow \nabla_{\boldsymbol{\lambda}} f_t(\boldsymbol{\lambda}, \boldsymbol{\beta}, \epsilon)$

Compute : $\mathbf{H}_t \leftarrow \nabla_{\boldsymbol{\lambda}, \boldsymbol{\beta}}^2 g_t(\boldsymbol{\lambda}, \boldsymbol{\beta}, \zeta^0)$

Compute approximation: $\mathbf{B}_t \leftarrow \frac{m}{\tilde{\eta}} \prod_{j=1}^{\widetilde{m}} \left( I_{d_2} - \frac{1}{\tilde{\eta}} \nabla_{\boldsymbol{\beta}, \boldsymbol{\beta}}^2 g_t(\boldsymbol{\lambda}, \boldsymbol{\beta}, \zeta^j) \right)$

Get estimate: $\widetilde{\nabla} f_t(\boldsymbol{\lambda}, \boldsymbol{\beta}, \mathcal{E}) \leftarrow \mathbf{g}_t - \mathbf{H}_t \mathbf{B}_t \nabla_{\boldsymbol{\beta}} f_t(\boldsymbol{\lambda}, \boldsymbol{\beta}, \epsilon)$

Return stochastic hypergradient estimate $\widetilde{\nabla} f_t(\boldsymbol{\lambda}, \boldsymbol{\beta}, \mathcal{E})$

---

# E  ADDITIONAL EXPERIMENTAL DETAILS

**Hyperparameter Details**: In both experiments, we employed single-loop updates ($K = 1$) and evaluated a range of window sizes $w \in \{50, 100, 250, 500, 1000, 5000\}$. Step sizes $\alpha, \eta \in$

$\{10^{-4}, 10^{-3}, 10^{-2}\}$ were selected via a grid search, following standard practice Ji et al. (2021); Huang et al. (2022a). We tune $(\alpha, \eta)$ through this grid search and perform ablations over the window size $w$ (Figures 1 and 3). The only new hyperparameter relative to prior work is the curvature parameter $\rho$, which in experiments is implicitly instantiated through the adaptive metric $H_t$ in the Bregman divergence $D_t(\lambda, \lambda') = \frac{1}{2}\|\lambda - \lambda'\|_{H_t}^2$. Thus, no additional hyperparameters require tuning.

**Extended Task Formulation.**

For our first experiment, we consider the task of learning a preconditioner $P(\lambda) \succ 0$ directly from data—a special case of optimizer learning Andrychowicz et al. (2016); Wichrowska et al. (2017). Given the previous iterate $\beta_{t-1}$, the inner problem is the proximal form of a preconditioned gradient step under the metric $P(\lambda)^{-1}$:

$$\beta_t(\lambda) = \arg\min_\beta \left\{ L_{\mathrm{tr},t}(\beta) + \frac{\gamma}{2}\|\beta - \beta_{t-1}\|_{P(\lambda)^{-1}}^2 \right\}.$$

The outer problem selects preconditioner parameters

$$\lambda_t \in \arg\min_\lambda F_t(\lambda), \qquad F_t(\lambda) := L_{\mathrm{val},t}(\beta_t(\lambda)),$$

so that the updated parameters generalize on the validation set. At each round, the bilevel optimization is naturally *online*: both preconditioners and model parameters evolve sequentially over $T$ steps, and the optimal solution varies across the evolving loss landscape.

**Extended Model Details and Results**

- *Quadratic loss.* We use $L_{\mathrm{tr},t}(\beta) = \frac{1}{2}\beta^\top H_{\mathrm{tr}}\beta - b_{\mathrm{tr}}^\top\beta$, with diagonal $P(\lambda) = \mathrm{diag}(\lambda)$. The inner problem admits a closed-form minimizer, enabling us to track the comparator sequence

$$H_{2,T} := \sum_{t=1}^T \sup_{\boldsymbol{\lambda} \in \mathcal{X}} \left\| \beta_{t-1}^*(\boldsymbol{\lambda}) - \beta_t^*(\boldsymbol{\lambda}) \right\|^2.$$

  The validation loss is quadratic as well, $L_{\mathrm{val},t}(\beta) = \frac{1}{2}\beta^\top H_{\mathrm{val}}\beta - b_{\mathrm{val}}^\top\beta$, with $H_{\mathrm{val}}, b_{\mathrm{val}}$ derived from a validation set.

- *SVM loss.* For linear scores $f_\theta(x) = \theta^\top x$ and labels $y \in \{-1, +1\}$, we define

$$L_{\mathrm{svm},t}(\theta) = \frac{1}{n_t} \sum_{i=1}^{n_t} \tau \log\big(1 + \exp\big((1 - y_i\theta^\top x_i)/\tau\big)\big), \quad \tau > 0.$$

  The inner problem becomes

$$g_t^{\mathrm{svm}}(\theta, \lambda) = L_{\mathrm{svm},t}(\theta) + \frac{\gamma}{2}(\theta - \theta_{t-1})^\top \mathrm{diag}(1/\lambda)(\theta - \theta_{t-1}),$$

  which is smooth and convex (indeed strongly convex for $\gamma > 0$), and can be solved efficiently by descent methods.

**Results**:
Bilevel local regret for the SVM model on the GSDC dataset is included in Figure 4.

**Runtime Comparison.** To complement the regret analysis, we report wall-clock running times across all algorithms and window sizes used in our experiments. Table 3 summarizes these running times in seconds. We highlight three key takeaways: *(i)* stochastic algorithms provide substantial speedups over deterministic methods due to mini-batching; *(ii)* increasing the window size $w$ noticeably increases the runtime of **OAGD** due to the cost of averaging hypergradients, whereas the runtime of all other algorithms remains largely insensitive to $w$ due to averaging *hypergradient evaluations*; and *(iii)* the additional computational overhead introduced by the Bregman proximal gradient step in **OBBO** and **SOBBO** is negligible in practice.

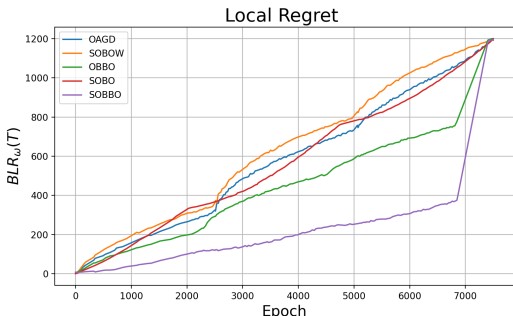

**Figure 4: SOBBO** offers the lowest regret relative to baselines.

**Table 3:** Running times (seconds) for preconditioner learning across algorithms and window sizes. Note that **SOBBO** with $(w=1)$ reduces to the SBio-BreD algorithm of Huang et al. (2022a).

| Algorithm | $w{=}1$ | $w{=}5$ | $w{=}25$ | $w{=}100$ | $w{=}250$ |
|---|---|---|---|---|---|
| OAGD | $29.52 \pm 0.23$ | $30.01 \pm 0.27$ | $31.76 \pm 0.28$ | $42.38 \pm 0.66$ | $60.56 \pm 0.37$ |
| SOBOW | $29.20 \pm 0.15$ | $29.65 \pm 0.29$ | $29.54 \pm 0.11$ | $29.68 \pm 0.17$ | $29.81 \pm 0.22$ |
| OBBO | $30.78 \pm 0.25$ | $31.98 \pm 0.24$ | $31.55 \pm 0.39$ | $29.41 \pm 0.14$ | $28.89 \pm 0.39$ |
| SOBO | $13.34 \pm 0.29$ | $13.47 \pm 0.42$ | $13.41 \pm 0.29$ | $13.92 \pm 0.21$ | $14.02 \pm 0.30$ |
| SOBBO | $13.72 \pm 0.10$ | $13.35 \pm 0.16$ | $13.33 \pm 0.20$ | $14.44 \pm 0.19$ | $14.56 \pm 0.35$ |

### E.1 BILEVEL MATRIX REGRESSION TASK

**Task.** We evaluate a Muon-style Jordan et al. instance of our Bregman bilevel optimizer on a bilevel matrix regression task. The outer variable $U \in \mathbb{R}^{p \times r}$ induces a weighting matrix $W(U) = UU^\top$, which represents a data-dependent weighting. **Models.** The inner variable $X \in \mathbb{R}^{m \times n}$ solves the strongly-convex weighted ridge regression $X^\star(U) = \arg\min_X, \frac{1}{2}|W(U)^{1/2}(A_{\mathrm{tr}} X B_{\mathrm{tr}} - C_{\mathrm{tr}})|_F^2 + \frac{\mu}{2}|X|_F^2$, and the outer objective evaluates the nonconvex validation objective $F(U) = \frac{1}{2}|A_{\mathrm{val}} X^\star(U) B_{\mathrm{val}} - C_{\mathrm{val}}|_F^2$. **Datasets.** The validation set is a strict subset of the training set, introducing distribution shift and forcing the optimizer to learn which samples matter through the adaptive weights $W(U)$. **Baselines** We instantiate **SOBBO** with a Muon-style step as defined by the time-varying quadratic potential $\phi_t(U) = \frac{1}{2}\|U\|_{H_t}^2 = \frac{1}{2}\mathrm{tr}(U^\top H_t U)$, which yields the Bregman divergence $D_{\phi_t}(U, U_t) = \frac{1}{2}\|U - U_t\|_{H_t}^2$. The adaptive metric $H_t$ follows the Muon update rule, using exponential moving averages of gradients: first the momentum $M_t = \beta M_{t-1} + (1-\beta)G_t$, then the second-moment accumulator $V_t = \gamma V_{t-1} + (1-\gamma)M_t^{\odot 2}$, and finally $H_t = \sqrt{V_t} + \varepsilon$, where $G_t = \nabla_U F(U_t)$ is the outer-level gradient. The resulting trust-region–normalized update takes the form $U_{t+1} = U_t - \alpha_t H_t^{-1}\widehat{M_t}$ with $\widehat{M_t} = M_t/\max\{1, |H_t^{-1/2}M_t|_F/\tau\}$, yielding a curvature-adaptive Muon-style update directly on $U$. Baselines include **OAGD**, **SOBOW**, **OBBO**, stochastic **SOBO**, and direct **Adam** and **Muon** applied to outer variable $U$.

**Results.** Figure 5 shows that our Muon-Style **SOBBO** achieves the lowest bilevel local regret, compared to the deterministic Euclidean baselines (**OAGD** and **SOBOW**) as well as direct single-level optimizers (**Adam** and **Muon**). Table 4 further demonstrates that all methods attain comparable mean squared error, indicating that our advantage does not come from overfitting or improved estimation accuracy, but rather from more stable and geometry-aware optimization dynamics. Together, these results show that Muon-Style **SOBBO** provides an effective curvature-adaptive mechanism for online bilevel learning, outperforming both existing bilevel methods (**OAGD**, **SOBOW**) and strong single-level baselines (**Adam, Muon**).

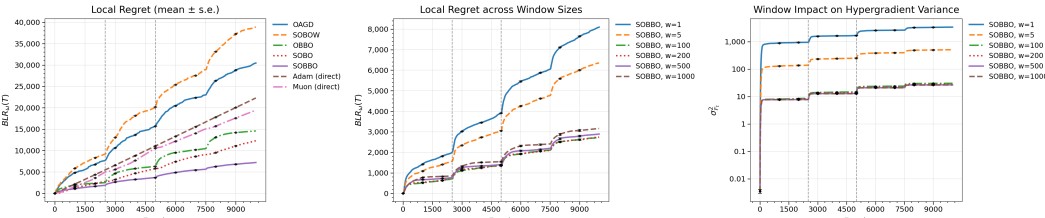

**Figure 5: Left:** Regret for deterministic and stochastic online bilevel optimizers on bilevel matrix reweighted regression; **Middle:** Increasing window size reduces regret incurred by stabilizing updates. **Right:** Larger window size reduces the variance of stochastic hypergradient estimates, as shown theoretically in Corollary 6.2.

| Algorithm | $t{=}2500$ | $t{=}5000$ | $t{=}7500$ | **Mean $\pm$ SE** |
|---|---|---|---|---|
| *Values reported as $(\widehat{X}_t - 1.125) \times 10^3$* | | | | |
| OAGD | $-1.1$ | $+2.3$ | $+4.4$ | $+1.9 \pm 1.6$ |
| SOBOW | $-2.1$ | $+2.7$ | $-0.3$ | $+0.1 \pm 1.4$ |
| OBBO | $-\mathbf{2.4}$ | $+2.0$ | $+0.2$ | $-0.1 \pm 1.3$ |
| SOBO | $-0.6$ | $+\mathbf{1.2}$ | $-1.4$ | $-\mathbf{0.2} \pm 0.8$ |
| SOBBO | $-0.9$ | $+2.3$ | $-\mathbf{2.0}$ | $-\mathbf{0.2} \pm 1.3$ |
| Adam (direct) | $+4.9$ | $+5.3$ | $+4.7$ | $+4.9 \pm 0.2$ |
| Muon (direct) | $+4.9$ | $+5.4$ | $+4.9$ | $+5.1 \pm 0.2$ |

**Table 4:** MSE between $X_{\text{true}}$ and $\widehat{X}_t$ (over 3 seeds), $(\widehat{X}_t - 1.125) \times 10^3$ to highlight relative differences.

