# OpenReview forum: "Bregman Geometry for Stochastic Online Bilevel Optimization"
_ICLR.cc/2026/Conference — Submitted to ICLR 2026_

### Official Review · Reviewer_btib · 2025-10-25

**Soundness:** 3
**Presentation:** 3
**Contribution:** 3
**Rating:** 4
**Confidence:** 4

**Summary:**

This paper introduces Bregman geometry into stochastic online bilevel optimization (OBO) and proposes single‑loop algorithms with time‑smoothed hypergradient estimators (SOBBO / SOBOW). It proves local dynamic regret bounds that remove inner‑level condition‑number dependence, and interprets time smoothing as an intrinsic variance–bias trade‑off. Experiments on online hyperparameter selection and non‑stationary Actor–Critic RL show lower regret and better validation loss than OAGD/OBBO/SOBOW baselines.

**Strengths:**

1. Tightly aligned theory and mechanism. The chain “time smoothing → variance reduction → regret improvement” is rigorously tied together by inequalities and sums.
2. Single‑loop & sample efficiency. Avoids nested loops, suitable for online/streaming.
3. Stable empirical signals. Multiple figures demonstrate how increasing $w$ lowers variance and regret; the RL setting remains favorable.

**Weaknesses:**

1. Constants and tuning visibility. Many constants appear in the bounds; provide practical estimation heuristics or rules of thumb.
2. Broader baselines. Add meta‑learning/implicit differentiation/accelerated hyper‑optim baselines, especially in non‑stationary setups.
3. Compute–window trade‑offs. Larger $w$ incurs extra history and compute; please report wall‑clock/throughput alongside regret.
4. Scale & diversity. Current tasks are modest; larger models/longer online horizons/more complex non‑stationary distributions would improve external validity.

**Questions:**

1. Adaptive windows. Can $w$ be adapted to noise level/time without breaking the bounds?
2. Coupling of inner $K$ and $w$. Joint tuning guidance for convergence/regret?
3. Scope of condition‑number independence. Which non‑smooth or non‑strongly‑convex inner problems preserve the guarantees?

---

> ### Author Response · Authors · 2025-11-19
> **Author Response**
>
> We would like to thank the reviewer for their thoughtful comments and feedback. Find our detailed responses to weaknesses and questions below.
>
> # W1: Constant Calibration
> The constants in our analysis (e.g., $\ell_{F,1}$, $\mu_g$, $C_{\mu_g}$, $\rho$) do not require explicit estimation in practice, a standard practice in bilevel optimization (Ji et al., 2021; Huang et al., 2022). They appear only to ensure theoretical correctness of the step-size and curvature-normalization conditions (see Assumptions A–F and Theorems 6 and 17). In implementation, we follow standard practices (Ji et al., 2021; Huang et al., 2022) by tuning the step sizes $(\alpha, \eta)$ via grid search (see our Q1 response to Reviewer LdXy)  and performing ablations over window sizes $w$ (Figures 1 and 3). The only new constant relative to prior works is the curvature parameter $\rho$, which is implicitly realized through the adaptive metric $H_t$ in the Bregman divergence $D_t(\lambda, \lambda') = \tfrac{1}{2}(\lambda - \lambda')^\top H_t (\lambda - \lambda')$, so no additional constants require tuning.
>
> # W2: Broader Baselines
> Our work does not address meta-learning but focuses specifically on *online* bilevel optimization. Including meta-learning baselines would therefore be outside our problem scope. Similarly, *offline* bilevel methods such as AID and ITD have already been analyzed and shown to be grossly suboptimal in nonstationary settings (Tarzanagh et al., 2024) across a variety of tasks compared to the benchmark methods in our paper. For this reason, we restrict our comparisons to established *online* bilevel optimizers (OAGD, SOBOW) and our stochastic variants (SOBO, SOBBO), which directly align with the setting studied here.
>
> # W3: Window Size
> We clarify that increasing the window size $w$ does *not* increase memory or history requirements. As detailed in our paper, time-smoothing averages a fixed set of recent *gradient evaluations*, not stored gradients, and therefore maintains $\mathcal{O}(1)$ memory. Following the reviewer’s suggestion, we will report wall-clock times and throughput in the revised version. We also emphasize that SOBBO’s main computational advantage arises from its *single-loop* and *sample-efficient* design ($\mathcal{O}(1)$ samples per round), unlike baselines such as OAGD and SOBOW, which require double loops or full-batch gradients. This structural efficiency is reflected in the empirical results shown in Figures 1–3.
>
> # W4: Scale and Diversity
> Our experimental goal is to empirically validate the *condition-number improvement* shown by our theoretical regret bounds. This is effectively demonstrated in our preconditioner-learning and reinforcement learning experiments, which span both high-dimensional and nonstationary regimes. While larger models or extended horizons would provide additional ablations, they would not yield much further insight into the validity of our theoretical results, which the current experiments already confirm. That being said, we will be adding one additional experimental result and will provide updates here as soon as we have them.
>
> # Q1: Adaptive Windows
> Theoretically, our results hold for any *sublinear* choice of $w$, i.e., $w = o(T)$, as stated in Corollary 6.3. In practice, $w$ can be adapted over time (e.g., $w = \log T$ or via noise-aware schedules) without violating the sublinearity requirement or altering the asymptotic regret rate. While exploring adaptive scheduling of $w$ is beyond our current scope, the existing bounds directly accommodate such extensions.
>
> # Q2: Joint K and W convergence
> Our analysis guarantees convergence for any $K \ge 1$, including the efficient single-loop case $K = 1$ (Theorems 6 and 17). The window size $w$ only needs to satisfy a growth condition ($w = o(T)$) to ensure sublinear regret (Corollary 6.3). Thus, no specific coupling between $K$ and $w$ is required — in practice, we fix $K = 1$ for efficiency and report results for various $w$ via standard ablations. We note that selecting $K > 1$ would lead to a double-loop algorithm, incurring additional computational cost.
>
> # Q3: Scope of Condition Number Dependence
> As stated in our assumptions, our analysis considers the standard nonconvex–strongly-convex bilevel setting, where both levels satisfy the smoothness and differentiability conditions required for hypergradient-based updates. Extending the theory to nonsmooth or non–strongly-convex inner problems lies outside our scope, as such cases typically require penalty or subgradient methods rather than the gradient-based framework analyzed here.

---

> > ### Comment · Reviewer_btib · 2025-11-28
> >
> > The reviewer has read the response and the questions and answers for the other reviewers. All the concerns have been saved. Thanks for the clarification!

---

### Official Review · Reviewer_LdXy · 2025-10-31

**Soundness:** 3
**Presentation:** 3
**Contribution:** 2
**Rating:** 4
**Confidence:** 3

**Summary:**

This paper addresses stochastic online bilevel optimization (OBO) by introducing Bregman-based algorithms that eliminate severe dependence on the inner problem's condition number ($κ_g$).
Main Contributions:

Bregman Geometry for Improved Conditioning: The authors demonstrate that Bregman gradient steps achieve sublinear bilevel local regret ($o(T)$) while removing the $κ_g³$ dependence in the oracle setting and $κ_g⁵$ dependence in the stochastic setting, compared to standard Euclidean updates.
Single-Loop Stochastic Algorithms: Two practical algorithms are proposed:

SOBO (Euclidean): achieves sublinear regret but retains $κ_g⁵$ dependence
SOBBO (Bregman): combines time-smoothed hypergradients with Bregman steps, eliminating $κ_g⁵$ dependence while maintaining sublinear regret

Time Smoothing as Variance Reduction: The analysis reveals that time smoothing, previously used heuristically in deterministic OBO, naturally functions as a variance-reduction mechanism with controlled bias in stochastic settings.
Experimental Validation: Results on preconditioner learning and reinforcement learning tasks demonstrate superior performance over existing online bilevel baselines on nonstationary, ill-conditioned datasets.

The framework provides both oracle and practical stochastic guarantees, offering theoretical insights and practical improvements for large-scale bilevel optimization problems.

**Strengths:**

This paper demonstrates exceptional academic writing ability. The authors excel particularly in narrative structure, with a clear and coherent logical thread throughout the entire article. The introduction explicitly identifies two critical gaps in existing online bilevel optimization methods, then explains why investigating this problem is necessary. Through comparative tables (Table 1 and Table 2), the authors intuitively illustrate the limitations of existing methods, especially the dependence on the condition number $\kappa_g$.

**Weaknesses:**

1.The paper focuses on the issues of Bregman geometry and condition number dependence in "Online Bilevel Optimization (OBO)," which are theoretically challenging. However, the necessity of these issues has not been sufficiently validated in practical applications. The authors do not elaborate on the fact that the demand for online settings in bilevel optimization is not strong in most current machine learning tasks, especially in scenarios that require frequent updates to the outer layer parameters and where the inner problem is highly ill-conditioned, which are still relatively rare.

2. In the related work section of the Introduction, the authors could provide a more detailed description.

3. Section 6 only includes two experimental scenarios; it is recommended to add one more experimental scenario.

**Questions:**

1. The theoretical step size requirements are rather intricate, depending on numerous unknown constants including $l_F,1, l_g,1,$ among others. How were these step sizes determined in the experimental settings?
2. While the paper emphasizes the elimination of $\kappa_g$ dependence, the Regret bound in Theorem 6 still retains $\kappa_g$​ terms. Corollary 6.1 indicates that the Bregman parameter must be selected as $\rho = O(\kappa_g^3)$ to remove the $\kappa_g$ dependence. What specific Bregman reference function was employed in the numerical experiments? It would be helpful if the paper provided experimental evidence to justify the necessity of removing $\kappa_g$ dependence. Specifically, what are the typical values of $\kappa_g$ in the tested problems, and how much does the improved dependence translate to practical performance gains?
3. Before introducing the algorithm formally, could the authors clarify the design logic more explicitly? Specifically: Which are adapted from prior work? Which components are novel contributions? What motivated each of these design choices?
4. Section 6 presents only performance metrics such as regret, but includes no comparison of running times.

---

> ### Author Response · Authors · 2025-11-19
> **Author Response**
>
> We would like to thank the reviewer for their thoughtful comments and feedback.  Find our detailed responses to weaknesses and questions below.
>
> # W1: Demand of Online Bilevel Optimization
> We emphasize that the demand for *Online Bilevel Optimization (OBO)* arises naturally in modern machine learning tasks that operate in online or nonstationary environments, where data arrive sequentially, objectives evolve over time, and model parameters must adapt continuously. Representative examples include learned optimizer training (Andrychowicz et al., 2016), reinforcement learning (Prakash et al., 2025), hyperparameter optimization (Lin et al., 2024), and meta-learning (Tarzanagh et al., 2024) — all of which can be formulated as bilevel problems. Existing offline bilevel methods, however, assume fixed objectives and static data distributions, making them unsuitable for such settings. Recent work (Tarzanagh et al., 2024) has further shown that *offline methods are suboptimal compared to online algorithms under nonstationary dynamics* across a variety of the aforementioned tasks. Yet, compared to the offline literature, current online methods remain limited to Euclidean geometry and deterministic updates, revealing a significant gap in the field.
>
> We further emphasize that across these applications, the inner problem is often ill-conditioned due to overparameterization (e.g., neural networks) and correlated, high-dimensional data, leading to large condition numbers ($\kappa_g$). Our work directly addresses this challenge by incorporating *Bregman geometry* to normalize curvature and eliminate $\kappa_g$-dependence in the regret bound. We therefore view the development of geometry-aware online bilevel optimizers as a crucial step toward unifying theory and practice in adaptive, online machine learning.
>
> # W2: Revised Work
> Most relevant work across offline and online (stochastic) bilevel optimization was included. We will add a more comprehensive description  of related work in the revised pdf.
>
> # W3: Additional Experiment
> We will add an additional experiment and will provide updates here as soon as we have them.
>
> # Q1: Step Size Calibration
> Regarding the theoretical step size requirements, we clarify that constants such as $\ell_{F,1}$ and $\mu_g$ are used exclusively to establish the theoretical convergence guarantees and regret bounds of the algorithm. We rely on conventional approaches utilized in bilevel optimization (Ji et al., 2021; Huang et al., 2022), where algorithmic parameters are approximated via tuning instead of explicit computation. We will add that all outer ($\alpha$) and inner ($\eta$) step sizes were determined via a traditional grid search over the range $\{10^{-4}, 10^{-3}, 10^{-2}\}$, and the window parameter $w$ was analyzed over $\{50, 100, 250, 500, 1000, 5000\}$. This empirical practice matches what is standard in bilevel optimization research, and we will update the revised PDF to reflect this.
>
> # Q2: Source of Condition Number ($\kappa_g$) Improvement
> We want to clarify to the reviewer that the *$\kappa_g$ dependence in the regret bound of Theorem 6 (17) is removed through selection  of the Bregman geometry via the parameter $\rho$*. To make this clear, given the smoothness term $\ell_{F,1} = O(\kappa_g^3)$, we can choose the independent curvature parameter $\rho$ to be proportional to the problem's smoothness, $\rho = O(\ell_{F,1}) = O(\kappa_g^3)$ (as noted in Corollary 6.1). This approach allows us to perform a curvature normalization such that $\ell_{F,1} / \rho = O(1)$, yielding a $\kappa_g$-free sublinear regret bound. Note that in practice, one does not need to select the exact $\rho$; it only needs to scale proportionally to the smoothness constant $\ell_{F,1}$. For an analogy on how this generalizes a common practice in smooth optimization, see our W1 response to Reviewer Spie.
>
> In experiments, we will add that we use a time-varying quadratic divergence $\mathcal{D}_{\phi_t}(\lambda_2, \lambda_1) = \tfrac{1}{2}(\lambda_2 - \lambda_1)^\top H_t (\lambda_2 - \lambda_1)$, where the generating function is $\phi_t(\lambda) = \tfrac{1}{2}\lambda^\top H_t \lambda$, and $H_t$ is a diagonal matrix updated adaptively as in AdaGrad (Huang et al., 2022).
>
>
>
> The necessity of this approach is validated empirically: in the presence of large condition numbers, such as in Preconditioner Learning (GDSC) where the inner problem is highly singular ($\kappa_g = \infty$ for $p \gg n$) or in Reinforcement Learning where $\kappa_g$ is large ($O(10^2 - 10^4)$), SOBBO achieves a polynomial reduction in cumulative regret supporting our theoretical claims. This empirical success is coupled with achieving the lowest validation loss ($0.7214$), supporting our theory on how the Bregman geometry improves convergence compared to standard Euclidean methods in the practically ill-conditioned regime.

---

> > ### Author Response · Authors · 2025-11-19
> > **Author Response Continued**
> >
> > # Q3: Algorithm Logic Design
> >
> > We thank the reviewer for this comment on our design choices. The logic for SOBBO's construction lies in combining *novel geometry-aware updates with efficient stochastic hypergradient estimation, allowing us to perform single-loop optimization and achieve a sublinear, condition-number-free regret bound.* SOBBO integrates several novel components in the online problem, which we included in our Introduction but summarize again below.
> >
> > **Gap 1: Geometry.** We are the first to consider the application of the *Bregman Proximal Step* (Bregman, 1967) in the online problem, motivated by the need for geometry-aware updates that eliminate the severe $\kappa_g$-dependence present in Euclidean methods (Theorems 3 and 6). This generalization allows for the use of more sophisticated steps and divergences, such as adaptive methods like AdaGrad or time-varying quadratic forms. Such Bregman methods are crucial, as large values of $\kappa_g$, often present in overparameterized models or ill-conditioned datasets, cause outer updates of Euclidean methods to converge slowly (see Figures 1–3).
> >
> > **Gap 2: Stochastic Setting.** We are the first to consider the online problem in a stochastic setting. Existing online bilevel methods are limited to the deterministic, Euclidean setting, requiring full-batch gradients that are impractical for large-scale, real-world machine learning applications. A stochastic formulation is necessary to align the theory with practice, where mini-batch sampling is essential for scalability and computational efficiency (Theorem 6). Here, we rely on two key mechanisms to manage the inherent noise of stochastic hypergradients: the adoption of the *stochastic inverse Hessian approximation* (Algorithm 4) for hypergradient estimation (Ghadimi & Wang, 2018), and the explicit use of *time-smoothing ($w$)* (Tarzanagh et al., 2024; Lin et al., 2024), which we show acts as a mechanism for $1 / w$ variance reduction (Corollary 6.2), a key finding in our stochastic analysis.
> >
> > In addition to the above, we provide single-loop efficient algorithms motivated by prior work in offline optimization (Khanduri et al., 2021), as outlined in Algorithm 4 and Theorem 6.
> >
> >
> > # Q4: Running Times
> > Regarding the lack of an empirical running time comparison, we will include this comparison in the revised PDF. However, we emphasize that SOBBO is not only the first online bilevel algorithm adaptive to the local geometry of the problem, but also the first single-loop and sample-efficient method, requiring only $O(1)$ gradient samples per round. In contrast, benchmark algorithms such as OAGD rely on nested (double) loops, while both OAGD and SOBOW require full-batch gradients. In addition to the clear performance advantage SOBBO provides in terms of regret, as demonstrated across our experiments (see Figures 1–3), the efficiency of our algorithm design provides further computational advantages.

---

> ### Author Response · Authors · 2025-12-01
> **Additional Experiment**
>
> Below we include an update on an additional experiment, as recommended by the reviewer. More details can be found in the revised pdf.
>
> # W3: Additional Experiment
>
> To demonstrate the flexibility and practical value of our framework, we include an additional experiment showing how our approach naturally recovers a Muon-style (K. Jordan et al. 2024) online bilevel optimizer through an appropriate choice of Bregman divergence. In this setting, the divergence is induced by the time-varying quadratic potential $\phi_t(U) = \frac{1}{2} U^\top H_t U$, which yields the Bregman divergence $D_{\phi_t}(U, U_t) = \frac{1}{2} (U - U_t)^\top H_t (U - U_t)$. The adaptive metric $H_t$ follows the Muon update rule, using exponential moving averages: momentum $M_t = \beta M_{t-1} + (1 - \beta)G_t$, second-moment accumulator $V_t = \gamma V_{t-1} + (1 - \gamma)M_t^{\odot 2}$, and $H_t = \sqrt{V_t} + \varepsilon$, where $G_t = \nabla_U F(U_t)$ is the outer-level gradient.
>
> We consider a bilevel matrix-regression task in which some samples are more relevant than others for generalization. Here the inner variable $X \in \mathbb{R}^{m \times n}$ solves the weighted regression
> $\min_X \tfrac{1}{2}\|W(U)^{1/2}(A_{tr} X B_{tr} - C_{tr})\|^{2} + \tfrac{\mu}{2}\|X\|^{2}$,
> and the outer variable $U \in \mathbb{R}^{p \times r}$ induces the PSD weight matrix $W(U) = UU^\top$ and minimizes the validation objective
> $F(U) = \tfrac{1}{2}\|A_{val} X^{*}(U) B_{val} - C_{val}\|^{2}$.
> To introduce a controlled distributional difference, the validation set is a strict subset of the training samples, making the training set a superset with additional irrelevant points. This structure enables the bilevel optimizer to identify which training samples matter most through the learned weights $W(U)$.
>
>
>
>
> The resulting Muon-style update on $U$ takes the form $U_{t+1} = U_t - \alpha_t H_t^{-1}\widehat{M}_t$ with trust-region normalization $\widehat{M}_t = M_t / \max (1, \|H_t^{-1/2}M_t\|_F / \tau)$, yielding a curvature-adaptive, trust-region–normalized preconditioned step. Baselines include **OAGD**, **SOBOW**, **OBBO**, **SOBO**, and direct application of **Adam** and **Muon**. As shown in the regret table for $t = 2500, 5000, 7500$, our Muon-style SOBBO method achieves the lowest regret at every evaluation point, reducing regret by roughly 35–40% relative to the baselines. The corresponding mean square errors show that all methods recover the underlying parameters to similar accuracy. Additional experiment details, results, and ablations will be provided in the revised pdf.
>
> **Table 1.** Bilevel local regret (averaged across 3 seeds).
>
> | Algorithm              | t = 2500   | t = 5000   | t = 7500   |
> |------------------------|------------|------------|------------|
> | OAGD                   | 7675.94    | 15734.70   | 23009.05   |
> | SOBOW                  | 9176.86    | 20170.71   | 28909.07   |
> | OBBO                   | 2503.34    | 6320.58    | 10446.02   |
> | SOBO                   | 2788.07    | 5793.88    | 9064.66    |
> | **Muon-Style SOBBO**   | **1878.89**| **3681.59**| **5641.37**|
> | ADAM (Direct)          | 5465.43    | 11062.41   | 16685.42   |
> | MUON (Direct)          | 4987.13    | 10577.26   | 15074.71   |
>
> **Table 2.** Mean squared error (averaged across 3 seeds) between X_true and X_hat_t.
>
> | Algorithm       | t = 2500    | t = 5000    | t = 7500    | Mean ± SE           |
> |-----------------|-------------|-------------|-------------|----------------------|
> | OAGD            | 1.123906    | 1.127329    | 1.129421    | 1.126885 ± 0.001607  |
> | SOBOW           | **1.122888**| **1.127739**| **1.124741**| 1.125123 ± 0.001413  |
> | OBBO            | **1.122646**| **1.126995**| 1.125202    | **1.124948** ± 0.001262 |
> | SOBO            | 1.124417    | **1.126221**| **1.123646**| **1.124761** ± 0.000763 |
> | **Muon-Style SOBBO**           | **1.124098**| **1.127343**| **1.122970**| **1.124804** ± 0.001311 |
> | Adam (direct)   | 1.129890    | 1.130263    | 1.129686    | 1.129946 ± 0.000169  |
> | Muon (direct)   | 1.129887    | 1.130371    | 1.129934    | 1.130064 ± 0.000154  |

---

### Official Review · Reviewer_Spie · 2025-10-31

**Soundness:** 3
**Presentation:** 2
**Contribution:** 3
**Rating:** 4
**Confidence:** 5

**Summary:**

This paper studies online bilevel optimization in the stochastic setting and proposes using Bregman geometry to eliminate the severe dependence on the condition number of the inner problem.

The authors introduce a single-loop, sample-efficient algorithm that combines Bregman steps with time-smoothed hypergradient estimates. The main theoretical contribution is achieving sublinear bilevel local regret $o(T)$ while exhibiting improved dependence on the condition number.

Experiments on preconditioner learning and reinforcement learning validate the theoretical results.

**Strengths:**

The paper studies online bilevel optimization to the stochastic setting with Bregman geometry. The motivation is well-articulated through two concrete gaps: (i) lack of geometry-aware updates for ill-conditioned problems where the inner condition number κ_g is large, and (ii) inability of existing deterministic methods (OAGD, SOBOW) to handle stochastic gradients and nonstationary data streams common in large-scale learning.


The paper provides comprehensive theoretical analysis for both oracle and stochastic settings. Theorem 3 establishes that Bregman steps achieve sublinear regret $o(T)$ without the $\kappa_g^3$ factor present in Euclidean methods. The main result (Theorem 17) shows SOBBO achieves sublinear regret with single-loop efficiency.


The experiments demonstrate practical benefits: on GDSC preconditioner learning, SOBBO achieves  lower regret than baselines. The actor-critic RL experiments (Pendulum with scheduled environment changes) validate the approach under nonstationarity. Figure 1 (middle/right) empirically confirms theoretical predictions that increasing window size w reduces both regret and hypergradient variance.

**Weaknesses:**

The paper has mathematical issues that undermine its soundness:

Theorem 12, Page 13): The proof states “Choosing $\alpha = 1/\ell_{F,1}$” but then claims selecting $\rho = O(\ell_{F,1})$ eliminates $\kappa_g$ dependence. This is circular reasoning: since $\ell_{F,1} = O(\kappa_g^3)$ (Lemma 2), choosing $\rho = O(\kappa_g^3)$ doesn't eliminate the dependence—it just hides it in the Bregman divergence. The computational cost of the Bregman step scales with $\rho$.


Lemma 14): The constant $C_{\mu_g} := 1 + \frac{\ell_{g,1} + \mu_g}{\eta \ell_{g,1}\mu_g}$ becomes unbounded as $\eta \to 0$. The lemma statement requires $0 < \eta \leq \frac{2}{\ell_{g,1} + \mu_g}$ but provides no lower bound. For $C_{\mu_g} = O(1)$, you need $\eta = \Omega(1/\mu_g)$, which should be stated.

Lemma 14 proof): After applying Young’s inequality with arbitrary $\delta > 0$, the proof sets $\delta = \frac{\eta\mu_g}{2}$. But this requires:
$
(1 + \delta)\left(1 - \frac{2\eta\mu_g}{\ell_{g,1} + \mu_g}\right)^K < 1
$
which imposes constraints on $\eta$ and $K$ not established in the lemma statement.

Presentation issues:

1. Equation (2): Fix spacing in the inner problem formulation to improve readability.
2.  There are issues with inconsistent use of `\citep` and `\citet` throughout the text.
3. Table 1: The notation `O(...)` should be replaced with $\mathcal{O}(...)$ to follow standard mathematical convention.

4. Lines 108–110: Inconsistent use of $\nabla g_t(\lambda, \beta, \zeta)$ vs. $\nabla g_t(\lambda, \beta)$. It's unclear when stochastic vs. deterministic gradients are being used.
5. Algorithm 3:  Inconsistent notation between $\bar{\zeta}_{t,k}$ and $\zeta_{t,k}$.
6. Table 2: Ambiguous formatting — it's unclear whether $\sigma_f^2 + \sigma_g^2$ is inside or outside the $\mathcal{O}(\cdot)$ notation.
7. Equation (7): The random product term involving $\tilde{m} \sim U(0,1,\ldots,m-1)$ is unclear when $\tilde{m} = 0$. This should be clarified (e.g., product equals identity in that case).

**Questions:**

In addition to the soundness issues,

Q1) Is the improvement in $\kappa_g$ dependence primarily due to the use of the Bregman divergence, or does it stem from your specific algorithmic design? More generally, does the use of Bregman divergence improve condition number dependence in bilevel optimization? If so, could you cite relevant prior work?

Q2) Can the improvement in condition number dependence be empirically validated in your experiments as well?

Q3) In the experiments, could you provide the values of $\kappa_g$, the specific Bregman function $\varphi(\lambda)$, and all relevant hyperparameters ($K$, $\alpha$, $\eta$, $s$, $m$, $w$)? Also, please include error bars and baseline comparisons.

---

> ### Author Response · Authors · 2025-11-18
> **Author Response**
>
> We would like to thank the reviewer for their thoughtful comments and feedback. Presentation issues will be resolved in the revised pdf. Find our detailed responses to weaknesses below.
>
> # W1: How the Condition Number ($\kappa_g$) Improvement is Achieved?
> We first emphasize to the reviewer that Theorems 12 (17) bounds regret by the ratio $\ell_{F,1} / \rho$. Since Lemma 2 gives $\ell_{F,1} = O(\kappa_g^3)$, selecting $\rho = O(\ell_{F,1})$ yields $\ell_{F,1} / \rho = O(1)$, thereby removing the $\kappa_g^3$ dependence in the regret bound. **This argument is not circular**, because $\rho$ is an *independent design parameter* that controls the curvature of the Bregman geometry. We show analytically that choosing $\rho$ proportional to the problem’s smoothness constant $\ell_{F,1}$ aligns the geometry’s curvature with the problem’s inherent smoothness — which serves as a form of curvature normalization. This alignment eliminates the condition-number dependence in the regret bound when the corresponding Bregman-based update is applied.
>
> As an analogy, our choice of $\rho = O(\ell_{F,1})$ can be viewed as a natural generalization of the classical practice in smooth optimization of choosing a step size proportional to $1 / \ell_{F,1}$ (where $\ell_{F,1}$ denotes the smoothness constant of $F$) to stabilize gradient updates (Nesterov, 2004; Beck, 2017). In Euclidean settings, this choice normalizes the update magnitude relative to the problem’s smoothness; here, the Bregman-based update achieves an analogous effect by normalizing with respect to the geometry induced by the potential function $\phi$, parameterized by $\rho$. This *curvature normalization* adaptively rescales the underlying geometry of the step rather than just the step size, yielding condition-number-independent convergence within a broader class of Bregman divergences.
>
> Regarding the dependence of the Bregman divergence on $\rho$, we note that the divergences considered in this work (e.g., Euclidean and time-varying quadratic forms) admit closed-form proximal updates such as adaptive preconditioned methods like AdaGrad (Duchi et al., 2011).
>
> # W2: Lower Bound of Constant $C_{\mu_g}$
> We appreciate the reviewer’s observation that the constant $C_{\mu_g} := 1 + \frac{\ell_{g,1} + \mu_g}{\eta\,\ell_{g,1}\mu_g}$ grows as $\eta \to 0$. In the revised PDF, we will make this dependence explicit by stating in Lemma 14 that $C_{\mu_g} = O(1 / (\eta \mu_g))$, and by noting that choosing $\eta = \Omega(1 / \mu_g)$ yields $C_{\mu_g} = O(1)$. Our analysis already accounts for this dependence: all subsequent lemmas and theorems (Lemmas 15–16 and Theorem 6 (17)) include $C_{\mu_g}$ explicitly in the bounds, for example through the constraint $\alpha \le O\left(\frac{\rho \sqrt{1 - \nu}}{\kappa_g^2 \sqrt{C_{\mu_g}}}\right)$. Therefore, the theoretical regret rate $O(T / w)$ in Theorem 6 (17) and the curvature-normalized removal of $\kappa_g$ are unchanged. This edit will make the scaling of $C_{\mu_g}$ with $\eta$ explicit.
>
> # W3: Young Inequality Concerns
> We clarify that we do not set $\delta = \frac{\eta \mu_g}{2}$ in our proof of Lemma 14. In the Young’s inequality step, we clearly write how we choose $\delta = \frac{\eta \ell_{g,1} \mu_g}{\ell_{g,1} + \mu_g}$. The contraction follows directly from the proof lines introducing Young’s inequality and the inner-loop decay: letting $c = \frac{2 \eta \ell_{g,1} \mu_g}{\ell_{g,1} + \mu_g} \in (0,1)$ (guaranteed by the existing step-size bound $\eta \le \frac{2}{\ell_{g,1} + \mu_g}$), our choice gives $\delta = \frac{c}{2}$, and for any $K \ge 1$ we have $(1 + \delta)(1 - c)^{K} \le (1 + \frac{c}{2})(1 - c) = 1 - \frac{c}{2} - \frac{c^{2}}{2} \le 1$. This verifies that the *contraction condition holds automatically for all $K \ge 1$ under our assumed $\eta$ range*, without requiring any additional lower bound on $K$. In Lemma 14, this contraction is absorbed into the decay parameter $\nu$, defined as $\nu = \left(1 - \frac{\eta \mu_g \ell_{g,1}}{\ell_{g,1} + \mu_g}\right)\left(1 - \frac{2 \eta \ell_{g,1} \mu_g}{\ell_{g,1} + \mu_g}\right)^{K - 1} \in (0,1)$, which governs the stability of the inner recursion. The same $\nu$ is propagated (along with $C_{\mu_g}$) into Lemma 15 and Theorem 6, making our dependence on $(\eta, K)$ explicit and consistent with the previously stated step-size conditions and final regret bound. We note that by forcing $\ell_{g,1} = \mu_g$ we retrieve the case $\delta = \frac{\eta \mu_g}{2}$; however, this corresponds to the condition number equaling 1 and is a special (and unlikely in real-data) case in which the problem is perfectly well-conditioned, exhibiting uniform curvature across all directions.

---

> > ### Author Response · Authors · 2025-11-18
> > **Author Response Continued**
> >
> > Find our detailed responses to questions below.
> >
> > # Q1: Source of Condition Number ($\kappa_g$) Improvement
> > The condition-number improvement in the regret bound results from the *coupling the Bregman geometry has on the generalized gradient and our algorithmic update* — an effect that requires the use of Bregman proximal gradient steps and one that cannot be achieved in the classical Euclidean setting. See our first response to Reviewer sqJw for a detailed description of this improvement with examples.
> >
> > # Q2: Empirical Validation of Condition Number ($\kappa_g$) Improvement
> > In Experiment 6.1 (preconditioner learning on GDSC), the dataset is inherently ill-conditioned because it operates in a high-dimensional regime with substantially more features than samples ($p \gg n$). This results in an inner level with a large condition number, making the Euclidean updates (OAGD, SOBOW, SOBO) poorly scaled. In this regime, both the deterministic and stochastic Bregman variants of our method (OBBO/SOBBO) achieve *substantial, polynomial improvements in regret compared to Euclidean methods* — as shown in Figures 1-3 — as well as achieving the lowest final validation loss. This empirical advantage directly reflects the theoretical condition-number improvement established in our analysis. Similar improvements can be seen in our RL experiment.
> >
> > # Q3: Hyperparameter Values and More Details
> > In both experiments, the condition numbers are prohibitively large. In Experiment 1 (GDSC), the high-dimensional regime ($p \gg n$) renders the inner problem rank-deficient, yielding $\kappa_g = \infty$. In Experiment 2 (RL), $\kappa_g$ remains large ($O(10^2 - 10^4)$) due to nonstationarity of the optimal policy and the underlying environment. For the Bregman function in (S)OBBO, we utilize a time-varying, adaptive quadratic divergence $D_t(\lambda, \lambda') = \frac{1}{2}\|\lambda - \lambda'\|_{H_t}^2$ with an adaptive metric $H_t$ updated as in AdaGrad, while Euclidean baselines used $D(\lambda, \lambda') = \frac{1}{2}\|\lambda - \lambda'\|_2^2$. We will include all requested hyperparameters and error bars in the revised PDF. All experiments used single-loop updates ($K = 1$) with window values $w \in \{50, 100, 250, 500, 1000, 5000\}$ and step sizes $\alpha, \eta \in \{10^{-4}, 10^{-3}, 10^{-2}\}$ (selected via grid search). While the most relevant baselines are included, error bars will be added in Experiment 2 (RL).

---

> > > ### Comment · Reviewer_Spie · 2025-11-23
> > > **Response to Authors**
> > >
> > > Thank you for addressing some concerns.
> > >
> > > However, multiple reviewers share concerns about the novelty of the $\kappa_g$ improvement (W1) and its relationship to prior work.
> > >
> > >
> > > Your paper states (page 2) "to our knowledge, there are no geometry-aware outer steps even in deterministic settings," yet Huang et al. (2022b) used Bregman divergence $D_{\psi_t}$ for outer bilevel updates with the same curvature normalization mechanism ($\rho = O(L_F)$); please see their Remark 1. Their update $x_{t+1} = \arg\min_{x} \{\langle w_t, x\rangle + h(x) + \frac{1}{\gamma}D_{\psi_t}(x, x_t)\}$ appears identical to yours. Could you please clarify
> > > - (1) How is your approach fundamentally different from Huang et al.'s framework beyond extending to online settings?
> > > - (2) For $\varphi(\lambda) = (\sigma/2)\|\lambda\|^2$ with $\sigma = \kappa_g^3$, isn't this just rescaled gradient descent where $\kappa_g$ dependence moves into $\rho$ rather than being eliminated?
> > >
> > >
> > > This would help clarify whether your primary contribution is the online bilevel adaptation (which is valuable) or involves novel algorithmic innovation beyond Bregman framework, and whether the $\kappa_g$ improvement differs from Huang et al. (2022b)'s curvature normalization approach.

---

> > > > ### Author Response · Authors · 2025-11-25
> > > > **Response to Reviewer**
> > > >
> > > > We thank the reviewer for engaging in the discussion. Find our response to the clarifying questions below.
> > > >
> > > > # Q1: Relation to Huang et al. Framework
> > > > Huang et al. (2022b) introduce Bregman-based algorithms for *offline* bilevel optimization, where the underlying objectives (and therefore the optimization target) remain fixed throughout training. Their convergence guarantees rely on this assumption.
> > > >
> > > > Although our outer update shares the same Bregman proximal form, the essential difference lies in the *online* formulation and the resulting need for a different *gradient proxy*. In the online setting, the outer objectives vary over time, so using only the instantaneous hypergradient (as in (S)BiO-BreD) leads to unstable and noisy updates, as we later demonstrate in empirical evaluation. To address this, we utilize *time-smoothing* of the hypergradient evaluations (Algorithm 3, line 11), averaging over a moving window of size $w$. When the problem is offline and $w = 1$, our method coincides with Huang et al. (2022b); however, in general, the time-smoothed hypergradient substantially extends Bregman proximal methods to online bilevel settings.
> > > >
> > > > Our theoretical results show that this generalized gradient proxy is crucial: while any fixed $w > 1$ yields a valid regret upper bound, achieving *sublinear* regret requires the window size to satisfy $w = o(T)$ under mild additional assumptions. In contrast, directly applying Huang et al.’s update (the case $w = 1$) does *not* achieve sublinear regret in our online setting. This distinction is also supported empirically. The table below summarizes additional results we performed for this review by extending Experiment 1 from the manuscript. In the quadratic benchmark, using $w = 5$ reduces the regret of Huang et al. algorithm, SBio-BreD, by approximately 42% at $t = 7500$, with larger windows giving further improvement. We will revise Figures 1–3 to include the SBio-BreD baseline and an expanded grid of window sizes $w$.
> > > >
> > > > Taken together, we show our approach is essential in the online regime to obtain sublinear regret, underscoring that our contribution combines both an *online extension of Bregman bilevel optimization* with an algorithmic innovation.
> > > >
> > > > **Table.** Comparison of regret values (mean ± SE) , as in Figure 1.
> > > >
> > > > | Algorithm              | t = 2500              | t = 5000               | t = 7500               |
> > > > |------------------------|-----------------------|------------------------|------------------------|
> > > > | SBio-BreD (w = 1)      | 3509.63 ± 94.66       | 14221.41 ± 2455.77     | 17078.69 ± 2513.72     |
> > > > | SOBBO (w = 5)          | 2105.74 ± 104.67      | 8342.67 ± 1884.64      | 9831.96 ± 1882.02      |
> > > > | **SOBBO (w = 100)**    | **925.48 ± 105.72**   | **7876.66 ± 3067.01**  | **8589.06 ± 3095.35**  |
> > > >
> > > > # Q2: Curvature Normalization and Special Cases
> > > > The reviewer is correct that since the smoothness constant satisfies $\ell_{F,1} = O(\kappa_g^3)$, one valid instantiation of our approach is to set $\rho = \kappa_g^3$. This yields the update $\lambda_{t+1} = \lambda_t - \tfrac{1}{\kappa_g^3}\nabla F_t(\lambda_t)$ — a rescaled gradient descent step — representing a special case of our general Bregman framework. More generally, our method allows adaptive, time-varying geometries such as $\phi_t(\lambda) = \tfrac{1}{2}\|\lambda\|_{H_t}^2$ with metrics $H_t$ defined analogously to AdaGrad, yielding curvature-normalized updates that adapt dynamically to problem smoothness and curvature. While Huang et al. (2022b) demonstrated the benefits of such geometry in the *offline* deterministic setting, whether these effects persist in the more realistic *online* regime, where underlying objectives evolve over time, was an open question.
> > > >
> > > > Our contribution establishes that the advantages of Bregman geometry do indeed extend to this online (stochastic) setting. To do this, we first had to introduce a novel *hypergradient error decomposition* (Lemma 15) that bounds the hypergradient error recursively under general Bregman geometries, together with a regret-based analysis that proves sublinear regret can be achieved for our proposed algorithms. Our results are the first to demonstrate that curvature normalization continues to yield tighter regret bounds, faster empirical convergence, and improved stability in nonstationary and ill-conditioned online bilevel problems. Notably, the special case $w = 1$, corresponding to Huang et al. (2022b), performs substantially worse in our experiments (see table above), underscoring the importance of our generalized framework.

---

### Official Review · Reviewer_sqJw · 2025-11-02

**Soundness:** 2
**Presentation:** 3
**Contribution:** 2
**Rating:** 4
**Confidence:** 3

**Summary:**

This paper investigates the problem of online bilevel optimization in both deterministic and stochastic settings. The main contribution is the proposal of a mirror descent type update scheme for solving the problem. The authors show that the regret bound of the proposed method, defined with respect to the mirror descent type update, is independent of the inner condition number $\kappa$. Extensive experiments are conducted to validate the effectiveness of the proposed approach.

**Strengths:**

+ The proposed method is independent of the condition number $\kappa_g$.
+ The paper is well structured and clearly written in most parts.
+ Experiments are conducted to validate the effectiveness of the proposed method.

**Weaknesses:**

- One of my main concerns is whether the improvement from removing $\kappa_g$ comes from algorithmic innovation or primarily from a change in the performance measure. In Theorem 6 and Corollary 6, the method does not impose specific requirements on $\phi$ beyond requiring it to be a $\rho$-strongly convex function with sufficiently large $\rho = O(\ell_{F,1})$. In this case, choosing $\phi = \frac{\ell_{F,1}}{2}\Vert \cdot \Vert_2^2$ also appears valid for the proposed method. However, under this choice, the update rule becomes almost identical to online gradient descent, differing mainly in the step size. From this perspective, the observed improvement from removing $\kappa_g$ seems to arise from a change in the performance measure rather than from substantive algorithmic advancement. As also discussed in Corollary 3.1, removing $\kappa_g$ appears difficult when performance is evaluated using the Euclidean norm.

- The significance of obtaining a bound that is independent of $\kappa_g$ is unclear. The paper would be more compelling if it provided concrete examples where $\kappa_g$ is prohibitively large or practically problematic.

- The experiments do not specify which Bregman divergence $\phi$ is used. It would help to state the exact choice of $\phi$ for each experiment.

- The proposed methods appear to require detailed knowledge of Lipschitz constants, and it is unclear how to reduce or remove this dependence in practice.


**Minor Issues**

- Line 131: Please specify with respect to which variable the function is Lipschitz continuous.
- Line 173: I believe an additional condition should be $\phi(x) = \frac{1}{2}\|x\|_2^2$.
- Line 184: There is a repeated “=”.
- Algorithm 3, line 4:* The word “to” is missing.

**Questions:**

Please refer to the questions listed in Weaknesses.

---

> ### Author Response · Authors · 2025-11-18
> **Author Response**
>
> We would like to thank the reviewer for their thoughtful comments and feedback. Minor issues will be resolved in the revised pdf. Find our detailed responses to questions below.
>
> # W1: Source of Condition Number ($\kappa_g$) Improvement
>
> **Summary**: While the reviewer is correct that setting $\phi(\lambda)=\frac{\ell_{F,1}}{2}||\lambda||_2^2$ makes our update equivalent to a scaled online gradient descent, we want to emphasize to the reviewer our condition-number free regret bound is not because of a generalized performance measure. Instead, as we outline in more detail below, this improvement results from the *coupling  the Bregman geometry has on the generalized gradient and our algorithmic update*—an effect that cannot be achieved by simply applying this rescaled OGD update to more general geometries that require non-Euclidean  or time-varying Bregman divergences (see our preconditioner learning in Experiment 1).
>
>
> **For more details**, consider the OGD case raised by the reviewer, which corresponds to a scaled version of our SOBO variant. With the Bregman geometry $\phi(\lambda)=\frac{\ell_{F,1}}{2}\lVert \lambda \rVert_2^2$, the update is $\lambda_{t+1}=\lambda_t-\frac{\alpha}{\ell_{F,1}}\nabla F_t(\lambda_t)$, and the generalized gradient is $G_X(\lambda_t,\nabla F_t(\lambda_t),\alpha)=\frac{1}{\ell_{F,1}}\nabla F_t(\lambda_t)$. As shown in our general proof, setting $\rho=\ell_{F,1}$ normalizes the effective smoothness ratio $\ell_{F,1}/\rho=O(1)$ in Theorem 3, thereby *eliminating the $\kappa_g^3$ dependence* from the bilevel local regret bound, where $\kappa_g>1$ is the condition number of the inner-level objective, often of large magnitude as remarked in W2.
>
>
> However, our framework extends well beyond this scaled-Euclidean special case. It applies to a broad family of geometries, including non-Euclidean and time-varying choices such as $\phi_t(\lambda)=\frac{1}{2}\lambda^\top H_t\lambda$ with adaptive diagonal $H_t \succ 0$ (as illustrated in our Adagrad-like SOBBO variant), where the Bregman geometry again defines both the update and the generalized gradient. In this setting, the update is $\lambda_{t+1}=\lambda_t-\alpha\,H_t^{-1}\nabla F_t(\lambda_t)$, and the corresponding generalized gradient is $G_X(\lambda_t,\nabla F_t(\lambda_t),\alpha)=H_t^{-1}\nabla F_t(\lambda_t)$. Unlike the OGD case, simply rescaling the step size by a constant does not yield a condition-number improvement here; rather, the improvement requires the coupling between the adaptive step size update and generalized gradient, which together determine how curvature is normalized and progress is measured.
>
> # W2: Significance of Condition Number ($\kappa_g$) Improvement
> The condition number $\kappa_g = \ell_{g,1} / \mu_g$ quantifies the curvature of the inner problem and becomes large when the lower-level objective is ill-conditioned, such as in overparameterized models or when the dataset itself is poorly conditioned. This is precisely the setting studied in our first experiment on preconditioner learning, where we consider high-dimensional regression and classification tasks on the GDSC dataset with $p \gg n$. In this regime, as well as in the RL experiment, large $\kappa_g$ leads to slower outer-level updates for Euclidean methods, while our Bregman-based SOBBO algorithm achieves lower bilevel local regret by normalizing curvature through the adaptive geometry. These results (Figures 1-3) empirically validate the importance of achieving $\kappa_g$-independent guarantees in practically ill-conditioned bilevel problems that match our theoretical claims.
>
> # W3: Bregman Divergence Used in Experiments
> We thank the reviewer for this helpful suggestion. In the experiment section, we will add that for SOBBO, the Bregman divergence is defined by the adaptive reference function $\phi_t(\lambda) = \frac{1}{2}\lambda^\top H_t\lambda$, where $H_t \succ 0$ is an adaptive diagonal matrix updated following the AdaGrad accumulation of past gradients (see Huang et al., 2022). This yields the time-varying Bregman divergence $D_{\phi_t}(\lambda, \lambda_t) = \frac{1}{2}(\lambda - \lambda_t)^\top H_t(\lambda - \lambda_t)$ and an update $\lambda_{t+1} = \lambda_t - \alpha H_t^{-1}\nabla F_t(\lambda_t)$.
>
> # W4: Lipschitz Constants
> As in prior bilevel optimization work, the Lipschitz constants are used in the analysis to establish regret bounds and step-size ranges. In practice, these constants are not known and step size calibration uses standard adaptive or tuning strategies, consistent with existing bilevel methods (e.g., Ghadimi & Wang, 2018; Ji et al., 2021; Huang et al., 2022). Specifically, we tune step sizes $\alpha$ and $\eta$ over a small grid and use the adaptive scaling of the Bregman metric $H_t$ (AdaGrad-style update) to implicitly adjust for local curvature. This empirical practice aligns with conventional approaches in bilevel optimization, where constants can be approximated through tuning rather than computed explicitly.

---

### Author Response · Authors · 2025-12-02
**Response to Area Chair and Reviewers**

Dear Area Chair and Reviewers,

Thank you for your constructive feedback. We have uploaded a revised PDF with the changes we committed to during the rebuttal highlighted in blue. The primary update to the paper is as follows:

# Additional Empirical Support.
We expanded the experimental section by incorporating recent advances in optimization algorithms, focusing on the *Muon* (Jordan et al., 2024) method, which has been shown to outperform widely used optimizers such as Adam and AdamW. We highlight that one can choose the Bregman divergence in our framework to obtain the first bilevel extension of a Muon-style algorithm, thereby offering a principled approach to applying Muon to online bilevel optimization (OBO) problems.

For additional support, we consider a matrix regression task---a standard technical benchmark because it is simple, structured, and analytically tractable while still capturing key optimization challenges encountered in neural networks---and demonstrate that our Muon-style SOBBO algorithm outperforms the original single-level Muon algorithm as well as current online bilevel optimizers. This result is significant: it generalizes an important optimization method to the bilevel setting for the first time and establishes SOBBO as the leading algorithm for this formulation. Overall, the bilevel Muon approach highlighted in this experiment is both relevant and beneficial for the community, as our algorithm achieves state-of-the-art performance.

# Importance/Novelty of Bregman Geometry and Stochasticity.
We use this opportunity to highlight the importance and novelty of introducing Bregman geometry and stochasticity to the online bilevel optimization problem below.

## Importance.
Online bilevel optimization faces two fundamental gaps. First, geometry: the key source of ill-conditioning lies in the *inner* problem, where the condition number $\kappa_g$ amplifies hypergradient sensitivity and destabilizes the outer update. While existing OBO methods (Tarzanagh et al. (2024); Lin et al. (2024)) use Euclidean updates, our generalization to  Bregman geometry introduces an independent curvature parameter $\rho$ that allows the update to be chosen so that this $\kappa_g$-dependence is normalized, important in nonstationary and ill-conditioned regimes common in reinforcement learning, optimizer learning, and matrix regression. Second, stochasticity: realistic large-scale OBO operates with minibatched, noisy gradients, but existing deterministic formulations (Tarzanagh et al. (2024); Lin et al. (2024)) require full-batch hypergradients and are not compatible with these dynamics. A stochastic formulation aligns the theory with practice and, as shown in Corollary 6.2, is important as it reveals that the time-smoothing technique serves as a *variance-reduction mechanism* essential for stabilizing hypergradient estimates and enabling sublinear regret.

## Novelty.
Our work is the first to address the gaps identified above. On the geometry side, we introduce Bregman proximal steps into OBO for the first time, enabling updates that normalize the inner condition number and achieve condition-number-free dependence in the regret bound. On the stochastic side, we provide the first analysis of OBO with stochastic hypergradients, showing that time-smoothing acts as a variance-reduction mechanism essential for sublinear regret. Offline bilevel methods cannot be directly used, as without time-smoothing their gradient proxies fail to deliver sublinear regret in time-varying environments, a limitation also reflected in our experiments. Together, these advances yield a new family of single-loop, sample-efficient Bregman-based algorithms that achieve condition-number-free, sublinear bilevel local regret in online stochastic settings.

In summary, our framework resolves the geometric and stochastic gaps in OBO, yielding the first condition-number-free, sublinear regret guarantees for the online stochastic bilevel problem. Thank you once again for your consideration.

---

### Meta-Review · Area_Chair_auXx · 2026-01-05

**Summary:**

The paper introduces Bregman divergence into online bilevel optimization and proposes an algorithm with a Bregman step. It claims that the resulting local regret bound is independent of the inner problem condition number $\kappa_g$, and the effectiveness of the method is supported by numerical experiments.

Overall, the reviewers were mildly negative and assigned scores of 4. A common concern raised by multiple reviewers (including sqJw, Spie) is whether the claimed removal of $\kappa_g$ dependence (in Theorem 6 and Corollary 6) reflects a genuine algorithmic improvement, or whether the dependence is merely shifted to the choice of performance measure or hidden within the Bregman divergence itself. It seems that the authors' response may not have fully addressed the clarity of their novelty, and I think a major revision might be required.

Given this potentially fundamental theoretical issue regarding the interpretation of the main contribution, I consider the paper to be marginally below the acceptance threshold.

**Reviewer Concerns:**

**Addressed**:

How to choose proper $\phi$ and Bregman divergence. The authors provides different choice of $\phi$ in different cases. I think this helps alleviate the reviewer sqJw’s concern regarding choice of $\phi$.

To demonstrate the flexibility and practical value, an additional experiment was conducted. I think this helps alleviate the reviewer Ldxy’s concern regarding insufficient experimental evaluation.

**Outstanding**:

Whether the improvement from removing $\kappa_g$ comes from algorithm innovation or primarily from a change in the performance measure. The dependence is not eliminated and it is just hided in the Bregman divergence as Bregman divergence relate with $\phi$ closely. This concern were shared by multiple reviewers.

The clarification regarding the fundamental differences between the authors’ work and prior work, for example how their approach differs from Huang et al.’s framework, was not found convincing or sufficiently supported by Reviewer Spie.

**Reviewer Scores:**

**Reviewer sqJw (Score: 4 -> Est. slight increase or unchanged):**

The reviewer’s score may increase slightly. The authors clarified the experimental choice of the Bregman divergence and better motivated the target setting, where the inner problem can indeed be highly ill-conditioned. While these responses address the reviewer’s concerns on experiments and problem relevance, they are unlikely to substantially change perceptions regarding the level of algorithmic novelty.

**Reviewer Spie ( Score: 4 -> Est. unchanged):**

The reviewer’s score is unlikely to change. Although the authors clarified that eliminating the dependence on $\kappa_g$ is a key algorithmic contribution of the paper, this explanation may not be sufficient to alter the reviewer’s original assessment of the claimed novelty.

**Reviewer LdXy (Score: 4 -> Est. slight increase or unchanged):**

The reviewer’s score may increase. The authors supported the relevance of ill-conditioned inner problems in OBO with additional literature and supplemented the experiments to better demonstrate the practicality and flexibility of the method.

**Reviewer btib (Score: 4 -> Est.  unchanged):**

The reviewer has carefully considered the rebuttal and the broader discussion. Although the responses are acknowledged, the reviewer’s core concerns persist, and a change in score is unlikely.

---

### Decision · Program_Chairs · 2026-01-26

Reject